EMBO
Molecular Medicine

# Jag1 insufficiency alters liver fibrosis via T cell and hepatocyte differentiation defects

Jan Mašek [1,2,3]✉, Iva Filipovic[4,6], Noémi Van Hul [1,6], Lenka Belicová [1,6], Markéta Jiroušková [5,7], Daniel V Oliveira [2,7], Anna Maria Frontino[2,7], Simona Hankeova[1,7], Jingyan He[1,7], Fabio Turetti [2], Afshan Iqbal [1], Igor Červenka[1], Lenka Sarnová[5], Elisabeth Verboven [1], Tomáš Brabec[2], Niklas K Björkström[4], Martin Gregor [5], Jan Dobeš [2] & Emma R Andersson [1,3]✉

## Abstract

Fibrosis contributes to tissue repair, but excessive fibrosis disrupts organ function. Alagille syndrome (ALGS, caused by mutations in *JAGGED1*) results in liver disease and characteristic fibrosis. Here, we show that $Jag1^{Ndr/Ndr}$ mice, a model for ALGS, recapitulate ALGS-like fibrosis. Single-cell RNA-seq and multi-color flow cytometry of the liver revealed immature hepatocytes and paradoxically low intrahepatic T cell infiltration despite cholestasis in $Jag1^{Ndr/Ndr}$ mice. Thymic and splenic regulatory T cells (Tregs) were enriched and $Jag1^{Ndr/Ndr}$ lymphocyte immune and fibrotic capacity was tested with adoptive transfer into $Rag1^{-/-}$ mice, challenged with dextran sulfate sodium (DSS) or bile duct ligation (BDL). Transplanted $Jag1^{Ndr/Ndr}$ lymphocytes were less inflammatory with fewer activated T cells than $Jag1^{+/+}$ lymphocytes in response to DSS. Cholestasis induced by BDL in $Rag1^{-/-}$ mice with $Jag1^{Ndr/Ndr}$ lymphocytes resulted in periportal Treg accumulation and three-fold less periportal fibrosis than in $Rag1^{-/-}$ mice with $Jag1^{+/+}$ lymphocytes. Finally, the $Jag1^{Ndr/Ndr}$ hepatocyte expression profile and Treg overrepresentation were corroborated in patients' liver samples. Jag1-dependent hepatic and immune defects thus interact to determine the fibrotic process in ALGS.

**Keywords** Notch; Jagged1; Alagille syndrome; Fibrosis; Treg
**Subject Categories** Digestive System; Immunology

## Introduction

Although fibrosis is a physiological tissue repair mechanism, excessive fibrosis is pathological and 45% of all deaths in the industrialized world are attributed to fibrosis (Henderson et al, 2020). Pathological liver fibrosis occurs in response to liver damage-induced chronic inflammation (Dobie et al, 2019). Hepatic fibrosis presents distinct fibrotic patterns including periportal, bridging or pericellular fibrosis, depending on the nature of the insult. Intriguingly, although Alagille syndrome (OMIM 118450; ALGS (Alagille et al, 1975)) and biliary atresia are both pediatric cholangiopathies, the associated liver repair and fibrotic mechanisms are distinct, with less bridging fibrosis but more pericellular fibrosis in ALGS than in biliary atresia (Fabris et al, 2007). One-quarter of children with ALGS survive to adulthood without a liver transplant, and in many of these children serum bilirubin, cholesterol, and GGT, as well as pruritus and xanthomas, spontaneously improve (Kamath et al, 2020). Understanding the fibrotic process in ALGS could reveal disease-specific mechanisms and leverageable pathways to modulate specific aspects of liver fibrosis.

Notch signaling controls development and homeostasis, and mutations in the Notch signaling pathway lead to both developmental diseases and cancer (Mašek and Andersson, 2017; Siebel and Lendahl, 2017). ALGS is mainly caused by mutations in the Notch ligand *JAGGED1* (*JAG1*, 94%) (Mašek and Andersson, 2017; Oda et al, 1997), affecting bile duct development and morphogenesis, resulting in bile duct paucity and cholestasis. Immune dysregulation has also been described (Tilib Shamoun et al, 2015), but how this might interact with liver disease in ALGS to affect fibrosis is not known. Cholestasis-induced inflammation results in fibrosis and cirrhosis driven by T cells (Shivakumar et al, 2007; Zhang and Zhang, 2020), natural killer (NK) cells (Shivakumar et al, 2009), and myeloid cells (Henderson et al, 2020; Jin et al, 2019), attracted to the liver by injured liver parenchyma (Allen et al, 2011). Conversely, fibrosis is attenuated by regulatory T cells (Zhang and Zhang, 2020). While Notch regulates T cell lineage specification in liver and thymus (Chen et al, 2019; Herman et al, 2005; Radtke et al, 2004), the function of Jag1 in this process is less clear. Importantly, there is a ~30% increase in infections in patients with ALGS (Tilib Shamoun et al, 2015), although this has been attributed to a Notch receptor-independent mechanism regulating T cell function itself (Le Friec et al, 2012) and it is unknown whether and how immune dysregulation in ALGS impacts liver disease and fibrotic progression.

[1]Department of Cell and Molecular Biology, Karolinska Institute, SE-171 77 Solna, Stockholm, Sweden. [2]Department of Cell Biology, Faculty of Science, Charles University, Viničná 7, 128 00 Prague 2, Czech Republic. [3]Department of Biosciences and Nutrition, Karolinska Institute, Huddinge 14183, Sweden. [4]Center for Infectious Medicine, Department of Medicine Huddinge, Karolinska Institutet, Karolinska University Hospital, Stockholm, Sweden. [5]Laboratory of Integrative Biology, Institute of Molecular Genetics of the Czech Academy of Sciences, Vídeňská, 1083 Prague, Czech Republic. [6]These authors contributed equally to this work as second authors: Iva Filipovic, Noémi Van Hul, Lenka Belicová. [7]These authors contributed equally to this work as third authors: Markéta Jiroušková, Daniel V Oliveira, Anna Maria Frontino, Simona Hankeova, Jingyan He. ✉E-mail: jan.masek@natur.cuni.cz; emma.andersson@ki.se

Here, we investigated liver-immune system interactions in inflammation and fibrosis using $Jag1^{Ndr/Ndr}$ mice, a model of ALGS (Andersson et al, 2018), with orthogonal validation of corresponding transcriptomic signatures in liver biopsies from children with ALGS. Using multiomics across liver, spleen, and thymus, as well as adoptive transfer of lymphocytes into lymphodeficient mice, we assess the lymphocytic capacity for interaction with liver cells. Our data demonstrate a multilayered role of Jag1 in the fibrotic process, affecting both hepatocyte maturation and injury response, as well as T cell differentiation and pro-fibrotic activity, providing a mechanistic framework for therapeutic modulation of Notch signaling in liver disease.

## Results

### $Jag1^{Ndr/Ndr}$ mice recapitulate ALGS-like pericellular fibrosis and signs of portal hypertension

Fabris and colleagues compared fibrotic architecture in ALGS and biliary atresia, demonstrating that patients with ALGS have less extensive bridging fibrosis but more pericellular fibrosis than patients with biliary atresia (Fabris et al, 2007). Liver fibrosis compresses blood vessels and reduces their blood flow, leading to portal hypertension, a serious consequence of liver disease that can manifest as splenomegaly. Therefore, we first asked whether $Jag1^{Ndr/Ndr}$ mice recapitulate ALGS-like pericellular fibrosis ((Fabris et al, 2007), Fig. 1A) and portal hypertension (Kamath et al, 2020).

$Jag1^{Ndr/Ndr}$ mice are jaundiced by postnatal day 3 (P3), with overt bile duct paucity and high levels of bilirubin at P10, which resolve by adulthood in surviving mice (Andersson et al, 2018). Sirius red staining (SR) at P10, P30 and 3 months demonstrated an onset of heterogeneous liver fibrosis at P10, characterized by pericellular "chicken-wire" fibrosis, perisinusoidal fibrosis and denser fibrotic loci with pronounced immune cell infiltrate, which was notable in sections but not significantly enriched when quantified (Fig. 1B–E). In contrast, fibrosis was significantly elevated at P30, with occasional faint bridging fibrosis and pericellular fibrosis (Fig. 1C,E). At 3 months, fibrosis had partially resolved into sparse pericellular collagen strands (Fig. 1D,E).

Splenic size is an indicator of portal hypertension in ALGS (Kamath et al, 2020). At birth, the $Jag1^{Ndr/Ndr}$ spleen was proportional to its body size, but splenomegaly developed by P10 and was further exacerbated by P30 (Fig. 1F,G). $Jag1^{Ndr/Ndr}$ mice thus mimic ALGS-like perisinusoidal, pericellular, and immune-associated fibrosis (Fabris et al, 2007) and develop splenomegaly, like patients with ALGS (Kamath et al, 2020; Vandriel et al, 2022). The atypical fibrotic reaction, despite severe bile duct paucity and cholestasis (Andersson et al, 2018) mimics ALGS and indicates that Jag1 mutation alters the profibrotic mechanisms, which we next investigated.

### Hepatocytes are less differentiated in $Jag1^{Ndr/Ndr}$ mice and in Alagille syndrome, with dampened pro-inflammatory activation

To investigate which liver cell populations are affected by Jag1 loss of function at the onset of cholestasis, we first analyzed $Jag1^{+/+}$, $Jag1^{Ndr/+}$, and $Jag1^{Ndr/Ndr}$ livers with single-cell RNA sequencing (scRNA seq) at embryonic day (E) 16.5 and postnatal day 3 (P3)

(Fig. 2A). 183,542 cells passed the quality control and filtering (Fig. EV1A,B), and could be assigned to 20 parenchymal and non-parenchymal cell clusters including tissue-resident Kupffer cells, and hematopoietic stem cells/erythroid-myeloid progenitors (HSc/EMP) and their derivatives: (i) the myeloid lineage, (ii) the lymphoid lineage, and (iii) the erythroid lineage (Liang et al, 2022; Popescu et al, 2019) (Figs. 2A and EV1C–E).

The adult hepatocyte injury response program depends on hepatic stellate cell (HSC) activation of NOTCH2 on hepatocytes, mediated by JAG1 (Nakano et al, 2017). We therefore subset and re-clustered the 3470 hepatoblasts and hepatocytes to analyze these in detail (Figs. 2B,C and EV2A), identifying $Meg3^+$ and $Dlk1^+$ hepatoblasts, $Pcna^+/Ube2c^+$ proliferative hepatoblasts and more mature $Fabp1^+$ or $Cyp4a14^+$ hepatocytes. While there were few or no significantly dysregulated genes between genotypes (Source Data Tables S1–5), there was an overt shift in cell population distributions between the genotypes (Fig. 2D). Hepatoblast and proliferating hepatoblasts ratios were similar across genotypes at E16.5, but $Jag1^{Ndr/Ndr}$ P3 livers contained 3-fold more Proliferative_Hepatoblasts than $Jag1^{+/+}$, $Jag1^{Ndr/+}$ controls (31.5% vs 9.1%, 10.1%, respectively, by proportion), but 1/3 less bone fide hepatocytes (68.8% vs 90.1%, 89.2%, respectively), (Figs. 2D and EV2B), suggesting a hepatocyte differentiation/maturation defect in $Jag1^{Ndr/Ndr}$ mice.

To determine whether hepatocyte differentiation was similarly delayed at later stages, we deconvoluted P10 $Jag1^{Ndr/Ndr}$ and $Jag1^{+/+}$ liver bulk RNA-seq data (Fig. EV2C, (Andersson et al, 2018)), using a neo- and post-natal liver scRNA-seq reference (Liang et al, 2022). Neonatal hepatocytes were significantly depleted in $Jag1^{Ndr/Ndr}$ livers (Fig. 2E). We then estimated the maturation stage of the hepatocytes in P10 livers using deconvolution with liver time-course scRNAseq datasets ((Liang et al, 2022), Fig. 2F) and ((L Yang et al, 2023), Fig. 2G). Both analyses showed that $Jag1^{+/+}$ P10 livers are, as expected, enriched for stage-matched hepatocytes (P7 for Liang et al, 2022 and P9–P15 for Yang et al, 2023). In contrast, $Jag1^{Ndr/Ndr}$ P10 livers were enriched for less mature hepatocytes, with 68% $^{+/-}$ 6% constituted of P3 hepatocytes using Liang et al, 2022. Deconvolution using Yang et al, 2023, which includes a wider range of time points (E17.5–P60), suggested variability in proportions of different maturation stages between liver samples. Interestingly, signatures of neonatal (E17.5/P3/P6), and mature (P30) hepatocytes were represented in $Jag1^{Ndr/Ndr}$ P10 livers, indicating the differentiation defect possibly overlaps with an injury response (Fig. 2G).

To identify genes driving the immature hepatocyte signature of $Jag1^{Ndr/Ndr}$ P10 livers, we generated a reference gene signature for hepatoblasts and hepatocytes using a published bulk RNA-seq dataset of isolated primary murine hepatoblasts and hepatocytes (top 500 enriched genes, Source Data Tables S6, 7 from (Belicova et al, 2021)), and performed gene set enrichment analysis (GSEA) on P10 liver bulk RNA-seq (Fig. 2H). $Jag1^{Ndr/Ndr}$ P10 livers were enriched for hepatoblast signature genes (Fig. 2H left,I) and depleted for hepatocyte signature genes (Fig. 2H right,I). Corroborating this, protein expression of the hepatoblast marker alpha-fetoprotein (AFP) was 3.1-fold enriched (Fig. 2J,K), while the mature pericentral hepatocyte marker CYP1A2 was 1.7-fold less expressed at the protein level (Fig. 2L,M) and almost undetectable at the RNA level (Fig. 2N). Similarly, the pericentral marker glutamine synthetase (GS) was downregulated at the protein

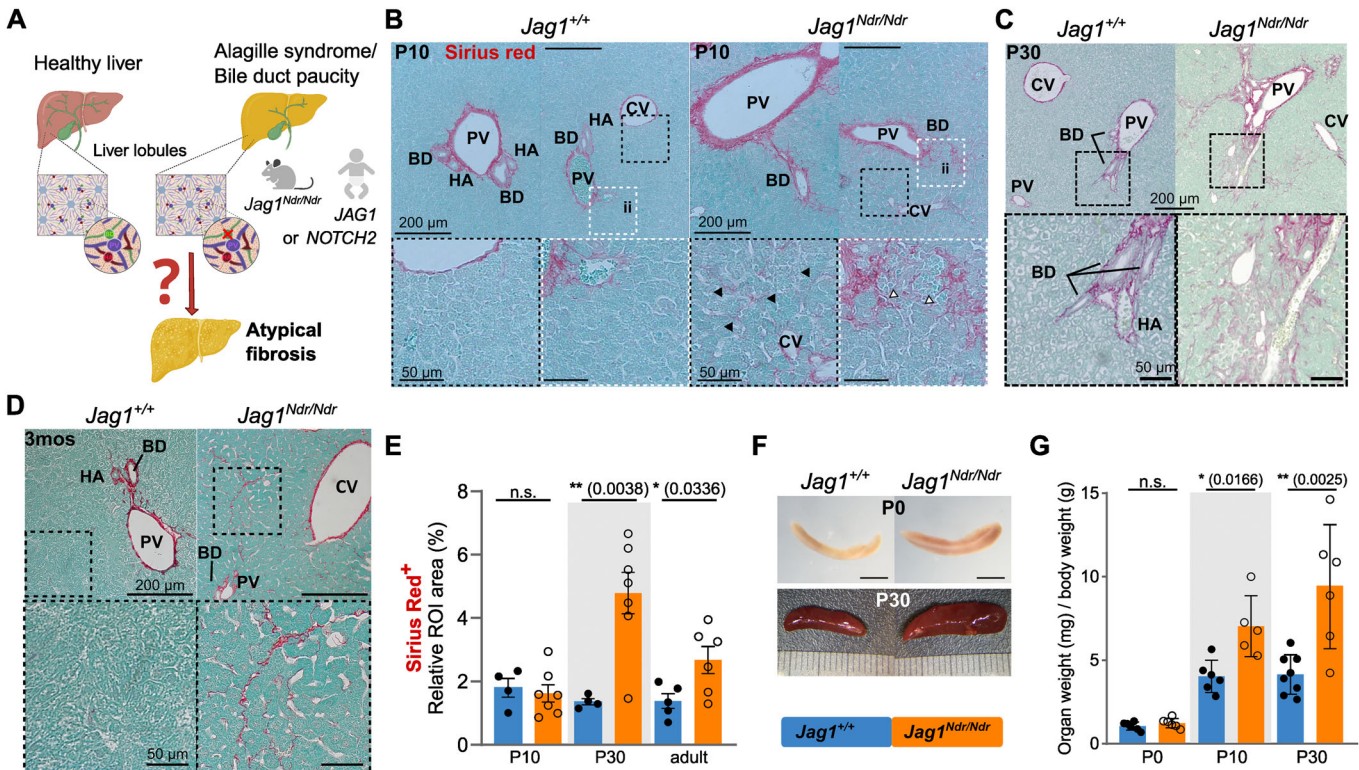

**Figure 1.** *Jag1^Ndr/Ndr* mice display pericellular fibrosis and signs of portal hypertension.

(A) Schematic of healthy liver architecture and liver architecture with bile duct paucity in patients with ALGS or in *Jag1^Ndr/Ndr* mice. Bile duct paucity results in cholestasis and liver disease, which in ALGS results in atypical fibrosis by unknown mechanisms and associated portal hypertension. (B–D) H&E and Sirius red staining at P10 (B), P30 (C) and 3 months (D) in *Jag1^+/+* and *Jag^Ndr/Ndr* livers, with boxed regions magnified. Black arrowheads indicate pericellular and perisinusoidal fibrosis, white arrowheads indicate immune infiltration. (E) Quantification of the Sirius Red⁺ at P10, P30 and 3 months in *Jag1^+/+* and *Jag^Ndr/Ndr* LLL liver sections. (F, G) Images (F) and weights (G) of spleens from *Jag1^+/+* and *Jag^Ndr/Ndr* mice at the indicated ages. The scalebar in (F) is 1 mm for P0, and at P30 a ruler indicates 1 mm increments. (Each data point represents a biological replicate). Mean ± SD, Unpaired, two-tailed Student's t-test, *p ≤ 0.05, **p ≤ 0.01, n.s. not significant. For (B–D), scalebar lengths are specified within each panel, and are identical within panels. LLL left lateral lobe, mos months, ROI region of interest, BD bile duct, PV portal vein, CV central vein, HA hepatic artery, ii immune infiltrate.

(Fig. 2L) and RNA level (*Glul*, Fig. 2N). RNA-seq analysis of liver samples from children with ALGS confirmed an enrichment of hepatoblast signature genes and depletion of hepatocyte-related genes in ALGS (Fig. 2H,I), demonstrating that hepatocytes are immature in both *Jag1^Ndr/Ndr* mice and in ALGS.

Finally, we tested whether the immature *Jag1^Ndr/Ndr* hepatocytes become activated and can initiate a pro-inflammatory Egr1-driven response upon cholestatic injury (Allen et al, 2011). Re-analysis of the P10 bulk RNAseq dataset showed that pro-inflammatory *Egr1* was significantly but only mildly upregulated in *Jag1^Ndr/Ndr* mice (2-fold), as were its targets *Cxcl1* (5.1-fold), *Cxcl2* (4-fold), *Cxcl10* (3.5-fold), *Ccl2* (4.7-fold), *Serpine1* (6.3-fold), *Isg20* (2.3-fold), and *Plaur* (2.9-fold). GSEA revealed a significant enrichment of myeloid leukocyte signatures in *Jag1^Ndr/Ndr* liver but only modest lymphocytic contribution, indicating that the mild hepatocyte activation may fail to attract lymphocytes (Figs. 2O, and EV2D,E, Source Data Table S8). In sum, scRNA seq, bulk RNA seq, and histological analyses revealed that hepatocytes are immature in *Jag1^Ndr/Ndr* mice and in people with ALGS. The mild pro-inflammatory activation of immature hepatocytes could compromise the ability of *Jag1^Ndr/Ndr* liver to attract and/or activate T cells upon cholestatic injury, which we next investigated.

## Intrahepatic T cells are under-represented in *Jag1^Ndr/Ndr* mice

Typically, the injured liver attracts immune cells to repair tissue damage, clear dead cells, and restore homeostasis (Allen et al, 2011; Hammerich and Tacke, 2023). To investigate immune cell composition and liver homing in *Jag1^Ndr/Ndr* mice, we analyzed livers and spleens with 25-color flow cytometry (Fig. 3A). Unsupervised clustering identified 20 cell types in 34 subpopulations, including mesenchymal cells, endothelial cells, and a majority of hematopoietic cells (Fig. 3A,B; Appendix Fig. S1A–C). There was a tendency towards fewer T cells in *Jag1^Ndr/Ndr* spleens at P3, and a significant 61% decrease in the frequency of CD4⁺ T cells in *Jag1^Ndr/Ndr* cholestatic livers compared to healthy *Jag1^+/+* controls of the same stage (Fig. 3C,D; Appendix Fig. S1D). We next stained and quantified lymphocyte-specific protein tyrosine kinase positive (LCK⁺) T cells in periportal areas of *Jag1^Ndr/Ndr* livers at P10, a stage at which *Jag1^Ndr/Ndr* mice are highly cholestatic (Andersson et al, 2018). Despite the ongoing cholestasis that should attract immune cells (Kisseleva and Brenner, 2021), there was no enrichment of LCK⁺ T cells in *Jag1^Ndr/Ndr* portal areas (Fig. 3E,F). The liver and splenic immune cell compositions in *Jag1^Ndr/Ndr* mice thus suggest a

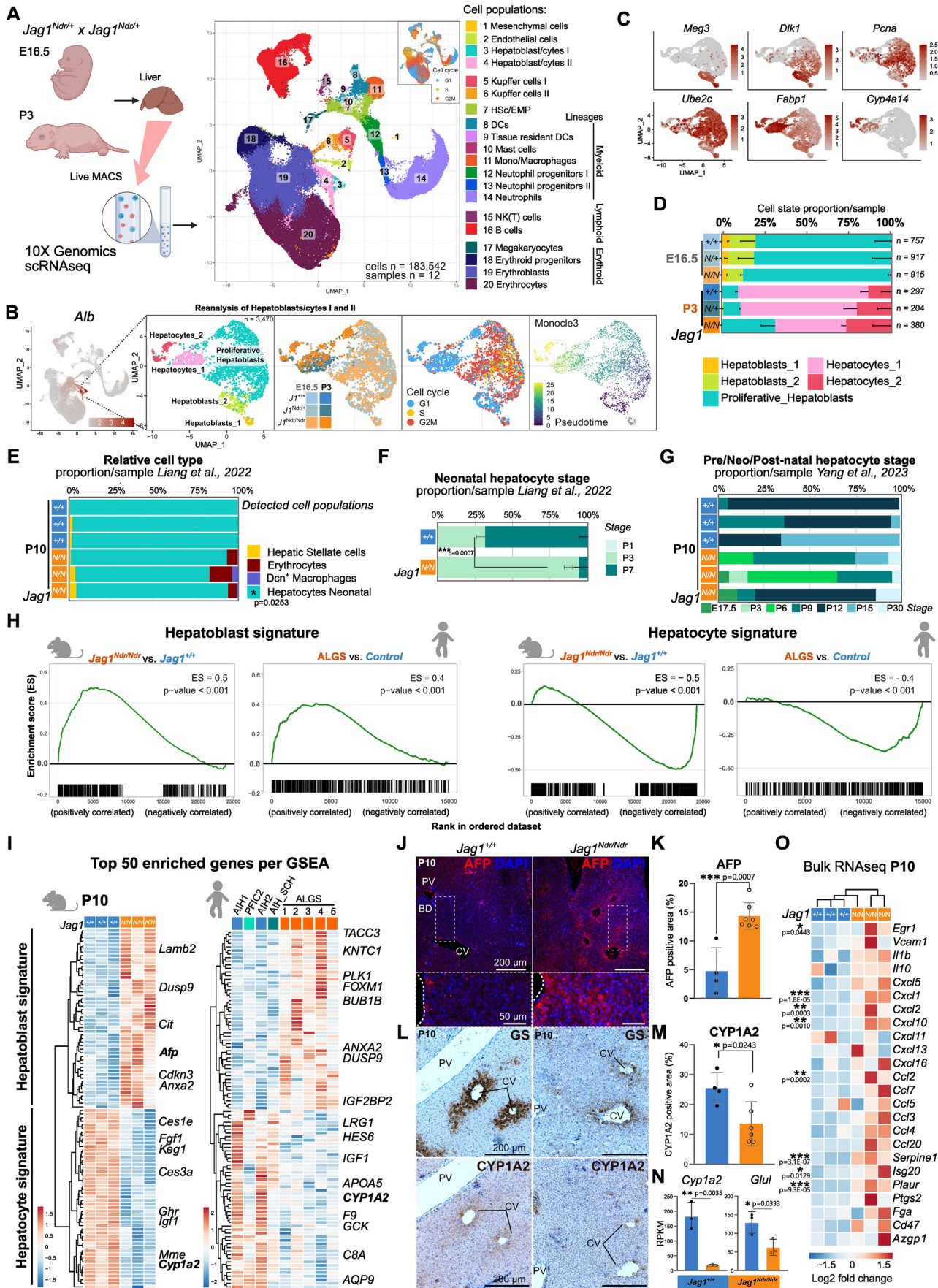

**Figure 2. Hepatocytes are less mature in neonatal liver of *Jag1^Ndr/Ndr* mice and patients with ALGS.**

(A) Livers were collected at E16.5 or P3 and single-cell suspensions were analyzed by 10xGenomics scRNA sequencing. UMAP projection of 183,542 sequenced cells in cell type-annotated clusters, and cell cycle phase. (B) UMAP of the Hepatoblast/cytes I and II subset, reflecting their subclusters, genotypes, developmental stage, cell cycle status, and differentiation trajectory. (C) Feature plots with hepatoblast (*Meg3*, *Dlk1*), proliferation (*Pcna*, *Ube2c*), and hepatocyte (*Fabp1*, *Cyp4a14*) marker mRNA expression. (D) Average proportion of hepatic cell types per stage and genotype, for data in (B, C). (E–G) MuSiC deconvolution of P10 *Jag1^Ndr/Ndr* and Jag1^+/+ whole liver bulk RNA seq using scRNAseq reference datasets (E), and stage-specific hepatocyte expression profile (F, G). (H) GSEA for hepatoblast and hepatocyte signatures in *Jag1^Ndr/Ndr* vs *Jag1^+/+* mice (left of each subpanel) or in patients with ALGS vs other liver diseases (right of each subpanel). Enrichment score (ES) reflects the degree to which a gene set is overrepresented at the top or bottom of a ranked list of genes. *p*-value was calculated using gene set permutation test (1000 permutations), *p*-value < 0.001 indicates the actual *p*-value of less than 1/(number of permutations), see methods for further details. (I) Heatmap of top 50 enriched genes from each GSEA in (H) for mouse (left) and human (right) liver. (J, K) Staining for AFP and DAPI at P10 in *Jag1^+/+* (n = 4) and *Jag1^Ndr/Ndr* (n = 7) livers (J), with quantification (K). Scalebar lengths are specified within the panel and are identical for *Jag1^+/+* and *Jag1^Ndr/Ndr* mice. (L–N) Staining for GS and CYP1A2, with H&E counterstain (L), in consecutive sections at P10 in *Jag1^+/+* (n = 4) and *Jag1^Ndr/Ndr* (n = 6) livers, with quantification of CYP1A2 protein (M), and P10 whole liver *Cyp1a2* and *Glu1* mRNA expression (n = 3 each) (N). Scalebar lengths in (L) are specified within the panel and are identical for *Jag1^+/+* and *Jag1^Ndr/Ndr* mice. (O) Heatmap of Egr1-induced pro-inflammatory genes expressed by hepatocytes in *Jag1^Ndr/Ndr* and *Jag1^+/+* mice at P10. HSc Hematopoietic Stem cells, EMP erythroid-myeloid progenitors, DCs dendritic cells, NK natural killer, AIH autoimmune hepatitis, PFIC2 progressive familial intrahepatic cholestasis type 2, SCH sclerosing cholangitis (mean ± SD; (E) was analyzed with 2-way ANOVA with Sidak multiple comparison). Interaction between cell type proportion and genotype is significant with *p*-value = 0.0139, graph indicates *p*-value of the multiple comparison. Data in (F), (K), (M), and (N) were analyzed by unpaired, two-tailed Student's t-test, *p < 0.05, **p < 0.01, ***p < 0.001, in (O) was significance confirmed via Benjamini–Hochberg *P* value correction (Log2 FC > 0.5).

defect in T cell response to cholestasis which could impact the fibrotic process.

## Disrupted thymic development and altered T cell differentiation in *Jag1^Ndr/Ndr* mice

Our data suggest both that hepatocytes fail to attract T cells to the liver and that T cell development may be disrupted in *Jag1^Ndr/Ndr* mice (Fig. 3). Notch signaling modulates immune system development (Vanderbeck and Maillard, 2021), and the immune system modulates the fibrotic process (Kisseleva and Brenner, 2021), but it is not known whether the immune system is impacted in ALGS and if it can affect fibrosis. We therefore next investigated the immune system in *Jag1^Ndr/Ndr* mice, its competence, and its role in liver fibrosis.

First, we characterized hematopoiesis and T cell development in *Jag1^Ndr/Ndr* mice at embryonic and postnatal stages. Hematopoietic progenitor development is regulated by Jag1-Notch signaling (Robert-Moreno et al, 2008). However, flow cytometry of E9.5 *Jag1^Ndr/Ndr* yolk sac and embryo proper revealed no differences in erythroid, macrophage, megakaryocyte, or erythro-myeloid progenitor populations at this stage (Fig. EV3A–D), suggesting the first wave of definitive immune cell production is unaffected in *Jag1^Ndr/Ndr* mice.

Lymphocyte progenitors differentiate into T cells in the thymus, a primary lymphoid organ in which thymic epithelial cells (TECs) provide instructive cues (Brandstadter and Maillard, 2019). Thymocytes that are double negative (DN) for CD4 and CD8 develop into double-positive thymocytes (DP) which subsequently differentiate into CD8^+ or CD4^+ T cells (Brandstadter and Maillard, 2019). The CD4^+ T cells can further develop into regulatory CD4^+ T cells (Tregs) (Fig. 4A). Several of these steps are Notch-dependent (Brandstadter and Maillard, 2019), but whether they are affected in ALGS is not known. We therefore assessed the thymus and its constituent cell types in *Jag1^Ndr/Ndr* mice.

*Jag1^Ndr/Ndr* thymi were significantly smaller than *Jag1^+/+* thymi at both P0 and P10 but reached a normal size by P30 (Fig. 4B,C). We mapped the expression of Notch components in human and mouse developing thymus using a published scRNA-seq atlas encompassing developmental stages from fetal, pediatric, and adult human individuals, and mouse thymi at 4, 8, and 24 weeks of age

(Park et al, 2020). This analysis confirmed enriched expression of *Jag1* in medullary Thymic Epithelial Cells (mTECs) (Li et al, 2020), and Notch1-mediated activation (high *Notch1/Hes1* expression) in DN thymocytes prior to differentiation into CD4^+ or CD8^+ T cells (Lehar et al, 2005; Radtke et al, 2004) (Fig. EV3E,F). However, although *Jag1* is expressed in TEC subpopulations and Notch signaling is active in these cell types with upregulated *Hes1* and *Nrarp* expression (Fig. EV3E,F), there were no significant differences in *Jag1^Ndr/Ndr* TEC subpopulations including Aire^+ mTECs, postAire mTECs (MHCII^+Ly6D^+Aire^neg/low), mTECs^low (MHCII^low CD80^low) or mTECs^high (MHCII^high CD80^high) (Fig. 4D), suggesting that while *Notch1* regulates TEC specification (Li et al, 2020), Jag1^Ndr does not disrupt this process.

We next analyzed T cell differentiation in the thymus at P30. While the frequency of DN thymocytes was similar in *Jag1^+/+* and *Jag1^Ndr/Ndr* thymi, there was a 25% reduction in DP thymocyte frequency in *Jag1^Ndr/Ndr* thymi, indicating skewed differentiation in *Jag1^Ndr/Ndr* mice (Fig. 4E). In contrast, CD4^+ T cells and Tregs were twice as prevalent in *Jag1^Ndr/Ndr* thymi, while the CD8^+ T cell frequency was unaffected in the *Jag1^Ndr/Ndr* thymus (Fig. 4E). These data suggest a disrupted DN-to-DP differentiation, and an excess production of T cells, particularly Tregs.

Finally, we assessed immune composition in the spleen, to determine how defects established in the thymus continue to develop in this secondary lymphoid organ and to investigate whether *Jag1^Ndr/Ndr* immune cells are correctly activated. While there were no differences in CD4^+ or CD44^+/CD4^+ activated T cell populations in *Jag1^Ndr/Ndr* spleens, there were significantly fewer CD8^+ T cells and CD44^+/CD8^+ activated T cells (Fig. 4F). This decrease in CD8^+ T cells entailed an increased CD4^+/CD8^+ T cell ratio in *Jag1^Ndr/Ndr* spleens. Importantly, similar to in the thymus, Tregs were 39% more frequent in the *Jag1^Ndr/Ndr* spleens (Fig. 4F). Collectively, these data revealed a role for Jag1 in thymic development in a model for ALGS, apparent already at P0, preceding cholestasis. We therefore next aimed to investigate the functional competence of the *Jag1^Ndr/Ndr* lymphocytes.

## *Jag1^Ndr/Ndr* T cells are less inflammatory

To investigate *Jag1^Ndr/Ndr* T cell function, we performed adoptive transfer of splenic lymphocytes into *Rag1^−/−* mice, which lack

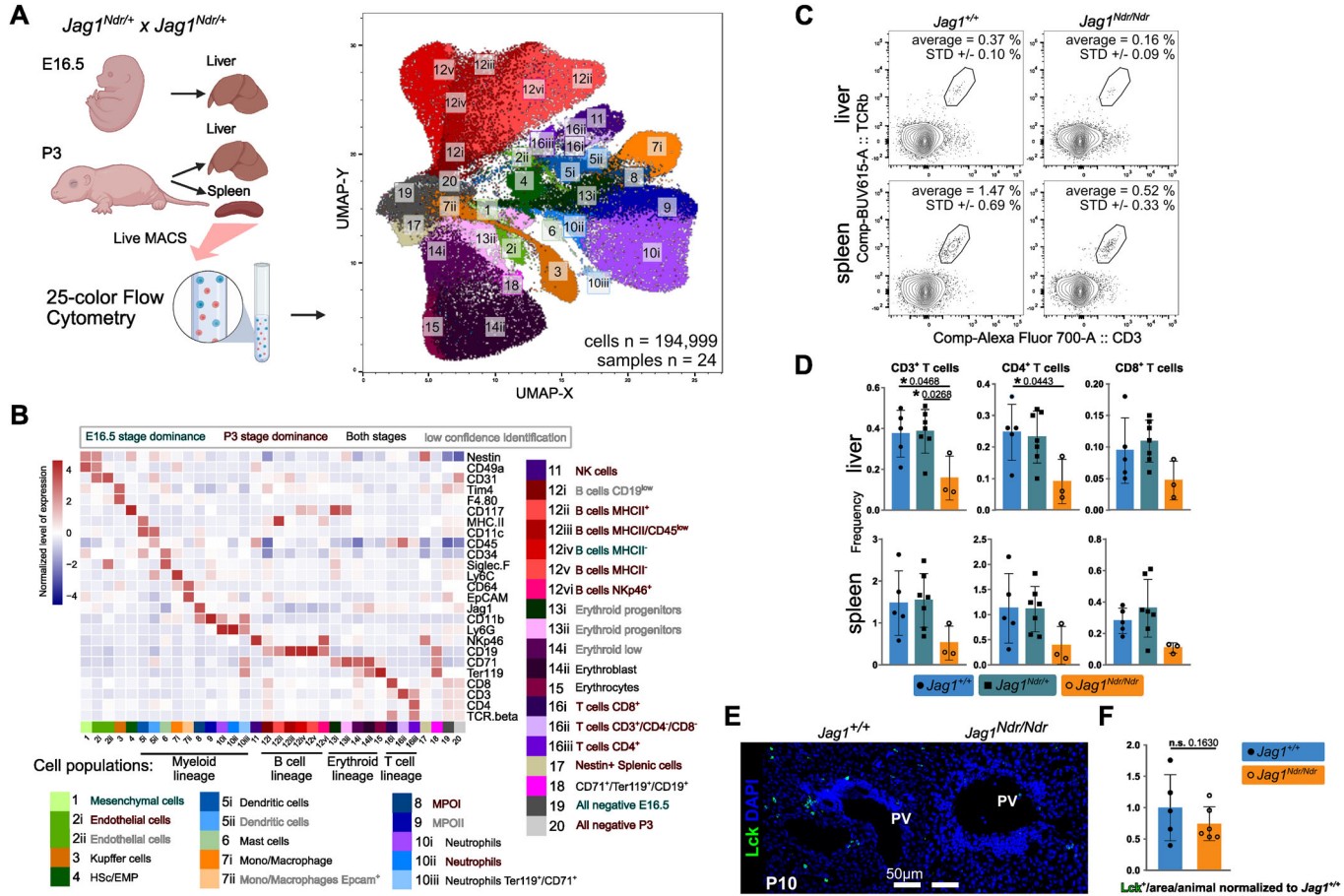

**Figure 3. Flow cytometry analyses reveal a reduction of T cells in *Jag1^{Ndr/Ndr}* liver.**

(A) UMAP projection of 194,999 randomly selected sub-sampled cells from E16.5 and P3 liver and P3 spleens, analyzed by 25-color flow cytometry, with cell type-annotated clusters, matched to (Fig. 2A) insofar as possible. (B) Heatmap showing a row z-score of median protein expression levels of cell type markers in the aggregated 25-color flow cytometry dataset from E16.5 and P3 livers, and P3 spleens of *Jag1^{+/+}* (*n* = 7), *Jag1^{Ndr/+}* (*n* = 11), and *Jag1^{Ndr/Ndr}* (*n* = 6) animals. (C, D) Representative flow cytometry plots (C) and relative frequency of the CD3^+, CD4^+/CD3^+, and CD8^+/CD3^+ T cells in livers and spleens from the *Jag1^{+/+}*, *Jag1^{+/Ndr}* and *Jag1^{Ndr/Ndr}* mice at P3 (D). (E, F) Representative immunofluorescent staining (E) and quantification (F) of LCK^+ T cells in periportal areas in *Jag1^{Ndr/Ndr}* and *Jag1^{+/+}* livers at P10 (*n* = 5 *Jag1^{+/+}*, and 6 *Jag1^{Ndr/Ndr}* animals). PV, portal vein. Scalebar lengths in (E) are specified within the panel and are identical for *Jag1^{+/+}* and *Jag1^{Ndr/Ndr}* mice. HSc hematopoietic stem cells, EMP erythroid-myeloid progenitors, MPO myeloid progenitors, NK natural killer. All graphs represent mean ± SD. Statistical test in (D) is one-way ANOVA with Dunnett's multiple comparison test, (*Jag1^{+/+}* vs. *Jag1^{Ndr/Ndr}*) *Jag1^{+/Ndr}* and *Jag1^{Ndr/Ndr}*, and in (F) is an unpaired, two-tailed Student's t-test, *p < 0.01); Each data point represents a biological replicate.

mature B and T cell populations, but provide a host environment with normal Jag1 (Mombaerts et al, 1992). T cell transfer to lymphodeficient recipients leads to mild gastrointestinal tract inflammation, the degree of which reflects hyper- or hypo-active T cells in homeostatic conditions. *Jag1^{Ndr/Ndr}* or *Jag1^{+/+}* lymphocytes were transferred into *Rag1^{−/−}* mice, their weight was monitored, and after 8 weeks their intestines were analyzed histologically, and their mesenteric glands were analyzed with flow cytometry (Fig. EV4A). There were no significant differences in survival, weight, or intestinal inflammatory status, between the recipients of the *Jag1^{Ndr/Ndr}* and *Jag1^{+/+}* lymphocytes (Fig. EV4B–E), demonstrating that *Jag1^{Ndr/Ndr}* lymphocytes are neither hyper- nor hypo-active in homeostatic conditions.

We next assessed *Jag1^{Ndr/Ndr}* T cell capacity to respond to challenge, using the well-established recovery model of dextran sodium sulfate (DSS)-induced intestinal colitis (Chassaing et al,

2014). Four weeks after *Jag1^{+/+}→Rag1^{−/−}* or *Jag1^{Ndr/Ndr}→Rag1^{−/−}* lymphocyte transfer, mice were challenged with two consecutive 10 day-long 2.5% DSS treatments, inducing bacterial infiltration and inflammation of the intestinal submucosa (Hernández-Chirlaque et al, 2016), followed by 7 days of tissue recovery, after which the intestine was analyzed histologically, and the mesenteric lymph node by flow cytometry (Fig. 5A). The intestines and colons were shorter and heavier in *Jag1^{Ndr/Ndr}* lymphocyte recipients, demonstrating less efficient recovery (Fig. 5B). As expected, histological scoring of intestinal inflammation revealed elevated inflammation in *Jag1^{+/+}→Rag1^{−/−}* mice treated with DSS (Fig. 5C,D) compared to *Jag1^{+/+}→Rag1^{−/−}* mice not treated with DSS (Fig. EV4D,E). However, there was significantly less inflammation in *Jag1^{Ndr/Ndr}→Rag1^{−/−}* mice than in *Jag1^{+/+}→Rag1^{−/−}* mice (Fig. 5C,D). While there was a tendency towards fewer LCK^+ T cells in the colon, the ratio of FOXP3^+ Tregs to LCK^+ T cells was similar between

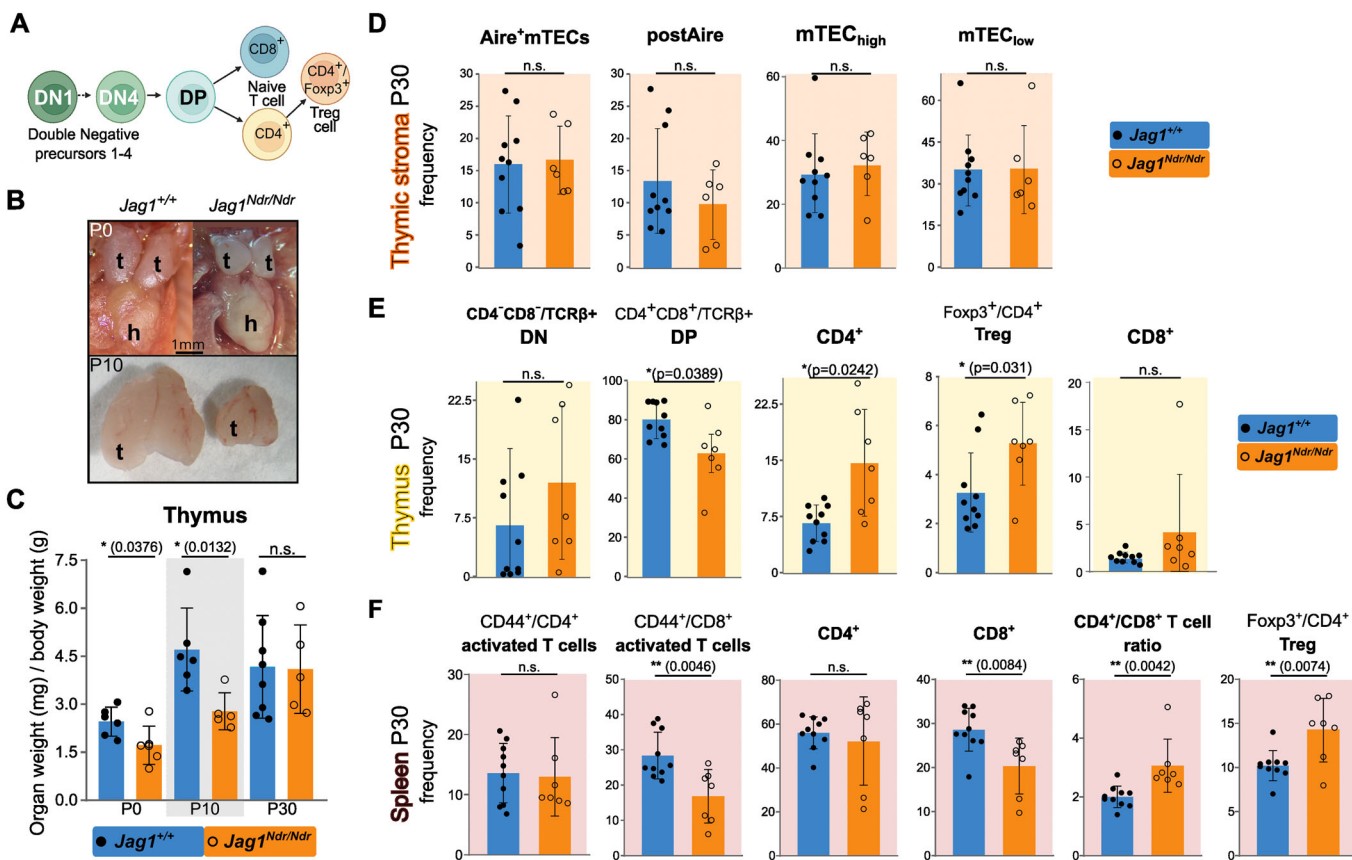

**Figure 4. Thymic developmental delay and excess Tregs in postnatal $Jag1^{Ndr/Ndr}$ mice.**

(A) Schematic of thymocyte development in the thymus. (B) Macroscopic images of $Jag1^{+/+}$ and $Jag1^{Ndr/Ndr}$ thymi at P0 and P10. (C) Thymic weights at P0, P10 and P30, normalized to body weight. (D–F) Flow cytometric analyses of cell frequencies of thymic epithelial cells, TECs (D), thymic thymocytes/T cells (E), and splenic T cells (F) in $Jag1^{+/+}$ and $Jag1^{Ndr/Ndr}$ mice at P30. Each data point represents a biological replicate. Mean ± SD, Unpaired, two-tailed Student's t-test, *p ≤ 0.05, **p ≤ 0.01, n.s. not significant. t thymus, h heart, DN double negative, DP double positive, Mac macrophage, NK natural killer cell, TEC thymic epithelial cell, mTEC medullary TEC, cTEC cortical TEC, Treg regulatory T cell.

the two experimental groups (Fig. EV4F–H). Finally, there were fewer activated CD44$^+$/CD4$^+$ and CD44$^+$/CD8$^+$ T cells in the $Jag1^{Ndr/Ndr} \rightarrow Rag1^{-/-}$ mesenteric lymph node compared to $Jag1^{+/+} \rightarrow Rag1^{-/-}$ controls (Fig. 5E). In sum, these transplantation experiments show that $Jag1^{Ndr/Ndr}$ T cells behave normally in homeostatic conditions (Fig. EV4A–E) but are less activated and less efficient at mediating recovery from intestinal insult (Fig. 5). The altered ability of $Jag1^{Ndr/Ndr}$ lymphocytes to become activated or sustain inflammation could impact liver fibrosis, which we next investigated.

### $Jag1^{Ndr/Ndr}$ T cells attenuate BDL-induced cholestatic fibrosis

Tregs are anti-fibrotic in the context of cholestatic liver injury (Roh et al, 2014; Zhang and Zhang, 2020), and were enriched in $Jag1^{Ndr/Ndr}$ mice (Fig. 4E,F). We therefore next tested $Jag1^{Ndr/Ndr}$ T cell effects on liver fibrosis. We investigated $Jag1^{Ndr/Ndr}$ contribution of lymphocytes to fibrosis using bile duct ligation (BDL), a surgically induced obstructive cholestasis model involving neutrophils and T cells (Licata et al, 2013). The transplanted $Rag1^{-/-}$ recipients were subjected to BDL surgery 4 weeks after $Jag1^{+/+}$ or $Jag1^{Ndr/Ndr}$

lymphocyte transfer (Fig. 6A). BDL treatment increased total and conjugated bilirubin, alkaline phosphatase (ALP), aspartate transaminase (AST) and alanine transaminase (ALT), reflecting ongoing liver damage, irrespective of the transplantation (physiological levels in gray, Fig. 6B).

To test immune cell impact on the extent of fibrosis, we analyzed collagen deposition with SR staining. Fibrosis levels in BDL-treated $Jag1^{Ndr/Ndr} \rightarrow Rag1^{-/-}$ livers were significantly lower than in BDL-treated $Jag1^{+/+} \rightarrow Rag1^{-/-}$ livers (Fig. 6C,D). Importantly, the impact on fibrosis was region-specific: pericentral fibrosis, with pericellular and perisinusoidal fibrosis, was similar in $Jag1^{+/+} \rightarrow Rag1^{-/-}$ and $Jag1^{Ndr/Ndr} \rightarrow Rag1^{-/-}$ mice (Figs. 6E and EV5A), while periportal fibrosis was reduced by threefold in $Jag1^{Ndr/Ndr} \rightarrow Rag1^{-/-}$ livers (Fig. 6C,F). To determine whether $Jag1^{Ndr/Ndr}$ lymphocytes affect fibrotic response to BDL, we analyzed the periportal expression of cytokeratin 19 (CK19, a marker of cholangiocytes), alpha smooth muscle actin (aSMA, myofibroblast, and HSC marker), and collagen1 (an acute fibrotic marker expressed by HSCs (Iwaisako et al, 2014)). The CK19$^+$ ductular reaction area was similar in $Jag1^{+/+} \rightarrow Rag1^{-/-}$ and $Jag1^{Ndr/Ndr} \rightarrow Rag1^{-/-}$ mice subjected to BDL. However, there was a tendency towards less collagen1, and there was significantly less aSMA+ signal in $Jag1^{Ndr/Ndr} \rightarrow Rag1^{-/-}$ livers subjected to BDL (Fig. 6G,H).

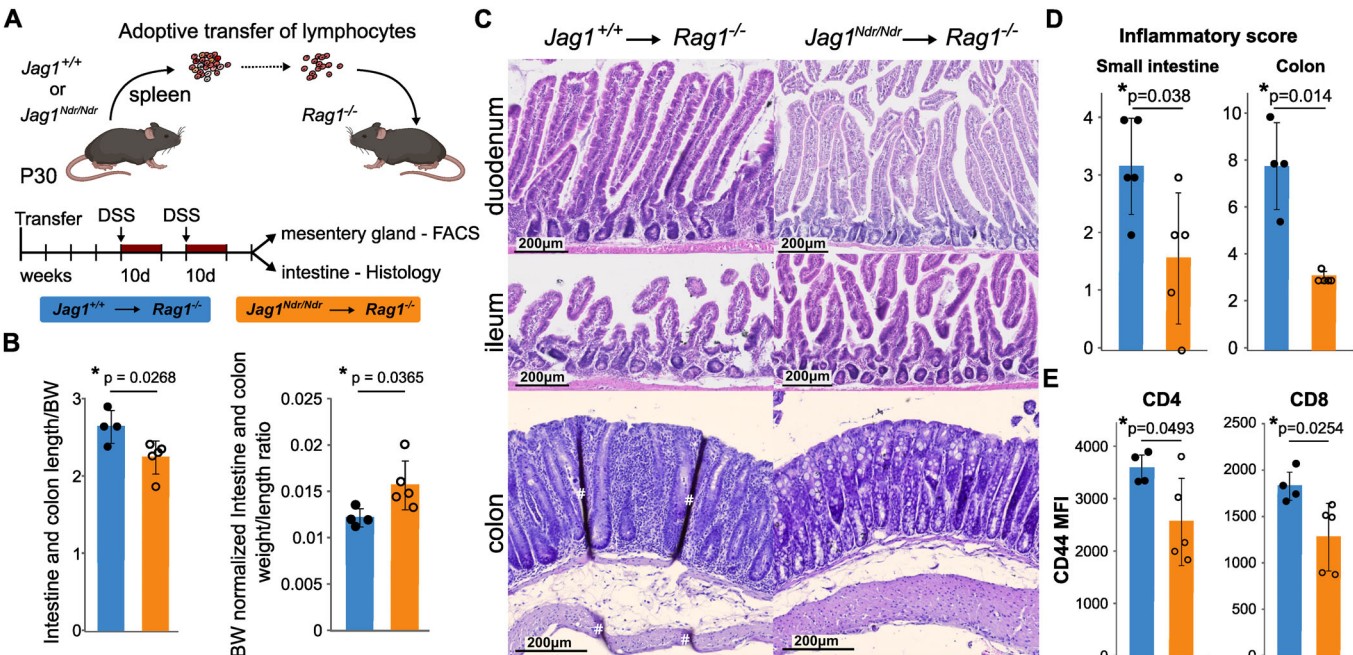

**Figure 5.** *Jag1^{Ndr/Ndr}* T cells do not mount an adequate response to DSS-induced colitis.

(A) Scheme of the Dextran sodium sulfate (DSS)-induced colitis. (B) Length, and weight/length ratios of the intestines and colons from DSS-treated *Jag1^{+/+}→ Rag1^{−/−}* (n = 4) and *Jag1^{Ndr/Ndr}→Rag1^{−/−}* (n = 5) mice, normalized to body weight. (C, D) Representative H&E-stained intestinal (top), and colonic (bottom) sections from *Rag1^{−/−}* mice transplanted with *Jag1^{+/+}* or *Jag1^{Ndr/Ndr}* T cells, upon DSS treatment (C), and their inflammatory scores (D). (E) Flow cytometry analysis of the mean fluorescence intensity (MFI) of CD44 staining of CD4^+ and CD8^+ T cells from DSS-treated *Jag1^{+/+}→Rag1^{−/−}* and *Jag1^{Ndr/Ndr}→Rag1^{−/−}* mesenteric lymph nodes. Each data point represents a biological replicate. mean ± SD, Unpaired, two-tailed Student's t-test, *p ≤ 0.05. # - tissue folding artifact.

Next, we assessed T cell liver infiltration in response to BDL-induced cholestatic liver injury with western blot and immunostaining for LCK. There were no differences in the capacity of *Jag1^{Ndr/Ndr}* or *Jag1^{+/+}* T cells to migrate into the liver upon BDL-induced liver injury (Fig. EV5B,C), with numerous LCK^+ T cells present in BDL ductular reaction loci (Fig. EV5D,E). However, there was a 2-fold increase in LCK^+/FOXP3^+ double positive Treg cells in the periportal area of *Jag1^{Ndr/Ndr} → Rag1^{−/−}* livers subjected to BDL (Fig. 6I,J).

To determine whether there could be a relationship between lymphocytic composition and fibrosis in patients with ALGS, we again analyzed previously published bulk RNAseq data ((Andersson et al, 2018), Fig. EV5F). A biliary stricture gene signature, reflecting cholestasis, was mildly upregulated in ALGS samples, while both immature T cell and Treg (FOXP3-driven) gene signatures were significantly enriched in livers from patients with ALGS (Fig. EV5G). Liver damage in non-ALGS liver disease (using liver injury marker *LGALS3BP)* (B Yang et al, 2021), was positively correlated with recruitment of lymphocytes (including *CD8A^+*, and *FOXP3^+* populations of T cells), as well as the extent of fibrosis (*COL1A1* abundance) (Fig. EV5H). However, in ALGS, the extent of liver damage, lymphocyte recruitment and fibrosis were unlinked (Fig. EV5H). These data are in line with the observation that liver stiffness (a proxy for fibrosis) in ALGS is independent of biomarkers of liver disease (Leung et al, 2023). While Treg infiltration in ALGS was independent of liver damage, it exhibited a tendency towards a negative correlation with fibrosis (Fig. EV5H), corroborating that elevated levels of Tregs may limit fibrosis in

ALGS. Altogether, these data suggest that the liver and lymphocytes may be differentially affected in different patients with ALGS, a disorder that is well known for its heterogeneous presentation. In conclusion, both *Jag1^{+/+}* and *Jag1^{Ndr/Ndr}* lymphocytes can home to a cholestatic liver, but *Jag1^{Ndr/Ndr}* and ALGS Tregs may modify the periportal fibrotic response.

# Discussion

Inflammation and fibrosis impair liver function, harm hepatocytes, and can initiate a cascade of events leading to portal hypertension, and ultimately liver cancer (Henderson et al, 2020). Notch signaling regulates liver development, and its disruption results in ALGS and bile duct paucity (Kohut et al, 2021). Whether Notch disruption in ALGS affects immune system development, and thus alters the course of fibrotic liver disease, was not known. Here, we dissected the onset of fibrosis using unbiased single-cell techniques, and transcriptomic cross-comparisons between *Jag1^{Ndr/Ndr}* and ALGS patient datasets. We show that cholestatic *Jag1^{Ndr/Ndr}* mice recapitulate ALGS-like pericellular fibrosis (Fabris et al, 2007) and develop splenomegaly, a common consequence of portal hypertension. Our unbiased omics analysis of the neonatal cholestatic liver revealed that *Jag1^{Ndr/Ndr}* hepatocytes exhibit an immature phenotype, with a limited capacity to transform into the activated pro-inflammatory state. Further, we showed that T cell development and function is affected in *Jag1^{Ndr/Ndr}* mice. We tested the competence of *Jag1^{Ndr/Ndr}* lymphocytes in a series of

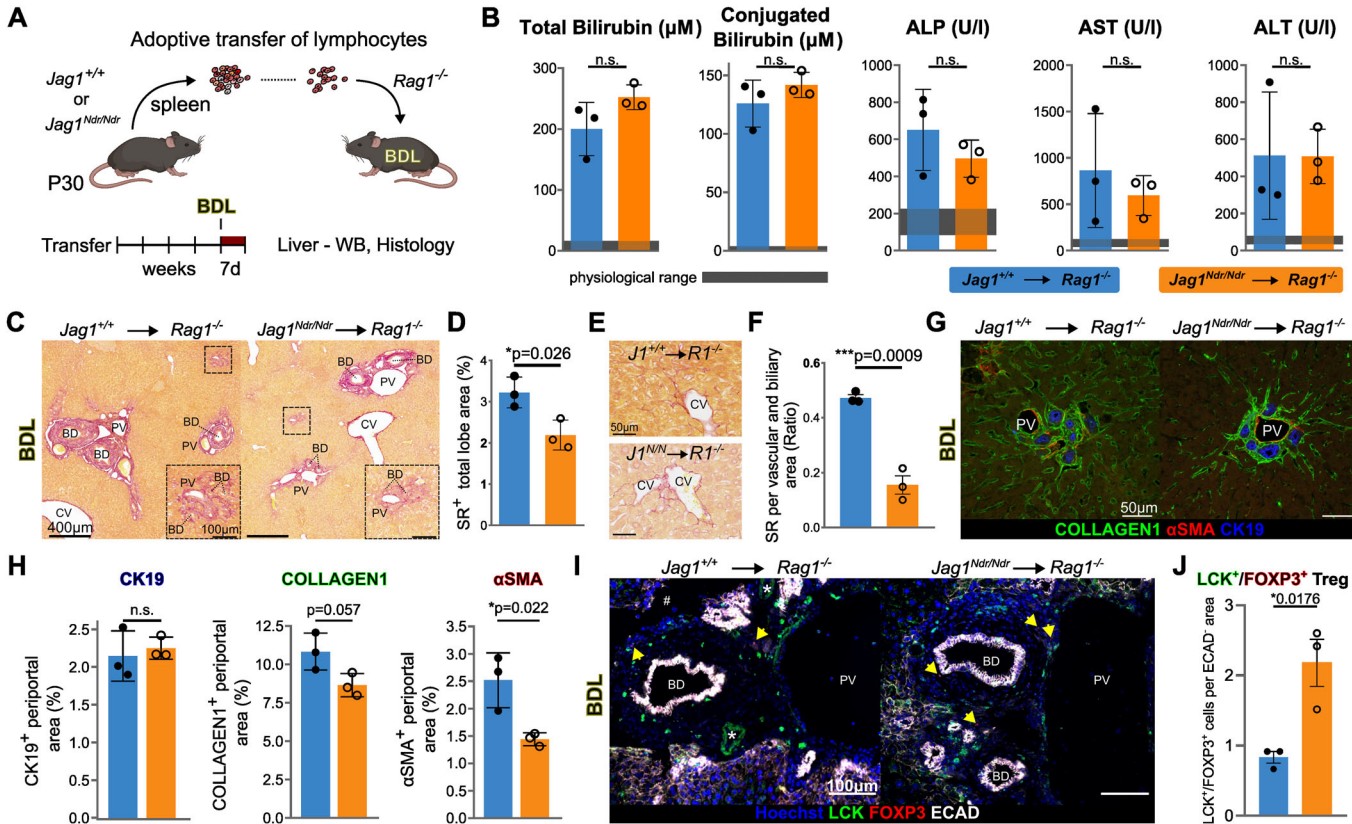

**Figure 6. *Jag1^Ndr/Ndr* Tregs limit the extent of bile duct ligation-induced periportal fibrosis.**

(**A**) Scheme of the BDL experimental model. (**B**) Liver biochemistry of the *Jag1^+/+^→Rag1^−/−^* and *Jag1^Ndr/Ndr^→Rag1^−/−^* mice after BDL (*n* = 3 each). Gray area indicates physiological ranges. (**C, D**) Representative images of Sirius red (SR) staining in *Jag1^+/+^→Rag1^−/−^* (left) and *Jag1^Ndr/Ndr^→Rag1^−/−^* (right) liver after BDL (**C**), and corresponding quantification (**D**), (*n* = 3 each). (**E**) Representative pericentral SR staining in *Jag1^+/+^→Rag1^−/−^* and *Jag1^Ndr/Ndr^→Rag1^−/−^* mice after BDL. (**F**) Quantification of SR staining in periportal areas of *Jag1^+/+^→Rag1^−/−^* and *Jag1^Ndr/Ndr^→Rag1^−/−^* livers after BDL, (*n* = 3 each). (**G, H**) CK19, aSMA, and Collagen1 staining of *Jag1^+/+^→Rag1^−/−^* and *Jag1^Ndr/Ndr^→Rag1^−/−^* livers after BDL (**G**), and quantification in periportal areas (**H**), (*n* = 3 each). (**I, J**) Representative immunofluorescent images of liver from *Jag1^+/+^→Rag1^−/−^* (left) and *Jag1^Ndr/Ndr^→Rag1^−/−^* (right) mice after BDL, stained for LCK, FOXP3, ECAD and nuclei (Hoechst). Yellow arrows mark LCK^+^/FOXP3^+^ Tregs (**I**), with LCK^+^/FOXP3^+^ Treg quantification in ECAD^-^ periportal area (**J**), (*n* = 3 each). Graphs represent means ± SD, unpaired, two-tailed Student's t-test); # - tissue-folding artifact; * in (**I**) denotes staining artifact of the LCK antibody. Scalebar lengths are specified for each panel and are identical within panels. BD bile duct, BDL bile duct ligation, CV central vein, HA hepatic artery, PV portal vein.

transplantation experiments, demonstrating that *Jag1^Ndr/Ndr^* lymphocytes' activation and contribution to periportal but not parenchymal fibrosis is limited, likely via Treg activity. Jag1 thus regulates hepatocyte injury response and thymocyte development and competence, with implications for the course of liver disease in ALGS.

*Jag1^Ndr^* mice are a suitable model to explore the consequences of systemic Jag1 dysregulation, which is relevant in ALGS (Andersson et al, 2018; Hankeova et al, 2021, 2022; Iqbal et al, 2024; Mašek and Andersson, 2024). As such, the immature hepatocytes in *Jag1^Ndr/Ndr^* mice could arise due to differentiation defects, or in response to liver injury, or both. Adult hepatocytes revert to an "immature hepatocyte phenotype" upon injury (Ben-Moshe et al, 2022; Iwai et al, 2000; Nakano et al, 2017), and in the context of CCL4 injury, JAG1 expressed by activated HSCs induces NOTCH2-dependent dedifferentiation of mature hepatocytes into Afp-expressing immature hepatocytes (Nakano et al, 2017). Whether an immature hepatocyte phenotype can be triggered by injury even in a Jag1 insufficiency context is thus an intriguing question. While the

Jag1^Ndr^ protein cannot bind or activate NOTCH1 (Hansson et al, 2010), it retains some capacity to bind and activate NOTCH2 (Andersson et al, 2018), which may be sufficient to induce dedifferentiation upon injury. In patients with ALGS, who have a single mutation in either *JAG1* or *NOTCH2*, the remnant healthy allele(s) could be expected to mediate signaling. However, some *JAG1* mutations exhibit dominant-negative effects (Guan et al, 2023; Ponio et al, 2007; Xiao et al, 2013), which could entail further repression of JAG1/NOTCH2 signaling. In this context, it is important to note that the *Jag1^Ndr/Ndr^* mice are homozygous for the missense mutation but retain some JAG1 activity (Andersson et al, 2018; Hansson et al, 2010), and it is not clear to which degree this mimics JAG1 heterozygosity in humans. It would be of interest to test whether Jag1 potency affects hepatoblast differentiation or injury-induced reversion of hepatocytes in patients, as a function of their specific *JAG1* mutation.

Despite acute cholestasis at P3 and P10, there was only modest or no induction of distress molecules or chemokines that are otherwise robustly upregulated in activated hepatocytes (Allen et al,

2011). *Egr1* is an early stress response gene (Gashler and Sukhatme, 1995), and its absence strongly attenuates inflammation in the BDL model of cholestasis (Allen et al, 2011). The reduced expression of *Egr1* in E16.5 *Jag1^Ndr/Ndr* hepatocytes is consistent with its decrease in hematopoietic stem cells in *Pofut1* KO mice. In this model, the reduction in *Egr1* is attributed not to Notch activation status but to impaired adhesion of Notch receptor-expressing hematopoietic stem cells to ligand-expressing neighboring cells, which is mediated by fucose-dependent Notch-receptor-ligand engagement (W. Wang et al, 2015). *Egr1* may thus be Jag1/Notch-regulated, limiting the capacity of hepatocytes to become activated in the *Jag1^Ndr/Ndr* environment with reduced Jag1 activity (ST2, Fig. 2O).

A limitation of this study is the underrepresentation of the hepatoblast/cyte parenchymal cells in the scRNA-seq dataset (Fig. 2A–D), which constituted ~6.5% of analyzed cells at E16.5, and ~7.5% of cells at P3 (Fig. EV1D). This parenchymal proportion is lower than in vivo but is consistent with scRNA seq datasets obtained with ex vivo liver digest (Guilliams et al, 2022). One risk is that cell stress as a result of dissociation could result in further loss of injured *Jag1^Ndr/Ndr* hepatocytes, impacting the interpretation of cell type abundance. Nuclear scRNAseq can overcome cell type-dependent dissociation sensitivity bias (Guilliams et al, 2022), and could provide further insights into *Jag1^Ndr/Ndr* livers at the single-cell level. Nonetheless, both bulk RNA seq deconvolution and histological analyses confirmed that patients and *Jag1^Ndr/Ndr* mice exhibit hepatoblast enrichment and less differentiated hepatocytes (Fig. 2E–N).

While the focus of this work is fibrosis, the reduced CD4^+CD8^+ DP cells and the CD4^+ Treg enrichment in *Jag1^Ndr/Ndr* mice is intriguing given the prominent roles of Notch in immune cell development (Brandstadter and Maillard, 2019). A comprehensive assessment of the immune system in patients with ALGS in a large, controlled cohort of patients has not been reported, although current data suggest the immune system may be compromised, with frequent (re)infections in ~25% of patients (Tilib Shamoun et al, 2015). In vitro co-culture experiments revealed that low Notch activation by JAG1 exhibits stage-specific functions in DN thymocyte development and induces CD4^+ T cell differentiation rather than DP differentiation (Lehar et al, 2005). In vivo, *Jag1* overexpression in T-lymphocyte progenitors (Beverly et al, 2006) or conditional *RBPJκ* silencing in TECs (García-León et al, 2022) induces TEC apoptosis and drives premature thymic involution. In *Jag1^Ndr/Ndr* mice a small-for-size thymus was evident at P0 and P10 (Fig. 4B,C) which normalized by P30.

Whether the thymus size normalization at P30 (Fig. 4C) is due to inhibited TEC apoptosis, and whether this impacts thymic inκvolution at later stages, is an interesting question that warrants further investigation. The small *Jag1^Ndr/Ndr* thymus and hypomorphic TEC-mediated JAG1 signaling (Lehar et al, 2005) could explain the dearth of CD4^+ T cells at P3 in *Jag1^Ndr/Ndr* livers (Fig. 3C–F). Interestingly, JAG1 can induce differentiation of T cells into Tregs (Hoyne et al, 2000; Vigouroux et al, 2003), which implies Tregs should be reduced in *Jag1^Ndr/Ndr* mice. Instead, Tregs were more frequent in both *Jag1^Ndr/Ndr* thymus and spleen at P30 (Fig. 4E,F). The enrichment in Tregs in *Jag1^Ndr/Ndr* mice is, however, in line with the previously reported Treg expansion upon *Notch1* or *RBPJκ* inactivation (Charbonnier et al, 2015). A Notch-independent Jag1-CD46 interaction was proposed to instruct Th1 responses in T cells, a process that was inhibited in patients with

ALGS (Le Friec et al, 2012). Currently, it is unknown if any analogous mechanism could also impact T cell development. Further immunological study of patients with ALGS and conditional abrogation of *Jag1* in vivo, would help further delineate the temporal and spatial requirements for *Jag1* in Treg development in ALGS.

The transplantation of *Jag1^Ndr/Ndr* lymphocytes resulted in reduced periportal fibrosis in response to BDL, compared to transplantation of *Jag1^+/+* immune cells (Fig. 6). In this experiment, the recipient *Rag1^−/−* mice have normal hepatic Notch signaling, and only Notch-defective lymphocytes cells are transplanted. The comparable numbers of LCK^+ T cells in the livers of transplanted *Rag1^−/−* mice (Fig. EV5B-E) demonstrate that *Jag1^Ndr/Ndr* T cells can respond to chemoattraction by activated hepatocytes, corroborating our interpretation that the *Jag1^Ndr/Ndr* hepatocytes are unable to adequately attract lymphocytes due to their low/absent inflammatory signature (Figs. 2O and 3C–F). The *Jag1^Ndr/Ndr* immune system presents three T cell phenotypes which all could mitigate liver fibrosis: fewer and less activated CD8^+ T cells which would otherwise promote liver fibrosis (Shivakumar et al, 2007) (Fig. 4F) and an increase in Tregs, which restrict liver fibrosis (Roh et al, 2014; Zhang and Zhang, 2020) (Fig. 4E,F). Finally, our experimental setup does not exclude an additional contribution by other lymphocytes (B cells or innate lymphoid cells) to the altered BDL-induced fibrosis, and selective testing of the individual subpopulations would be an intriguing follow-up to this study.

Importantly, while hepatic stellate cells are major drivers of fibrosis, portal fibroblasts appear to play a more significant role in BDL-induced fibrosis (Iwaisako et al, 2014; Koyama et al, 2017; Lua et al, 2016). The fact that *Jag1^Ndr/Ndr* lymphocytes limit periportal fibrosis after BDL suggests that portal fibroblast-induced fibrosis may be limited by *Jag1*-deficient T cell populations. Resolving whether HSCs and/or portal fibroblasts are key drivers of fibrosis in ALGS would be important for further studies on the interaction between the immune system and myofibroblasts in ALGS.

The pericellular fibrosis in *Jag1^Ndr/Ndr* mice recapitulates the fibrotic phenotype of patients with ALGS and suggests that *Jag1* is required for mounting a robust or rapid periportal fibrotic response, as seen in biliary atresia (Fabris et al, 2007). Fibrotic liver repair in ALGS is characterized by an expansion of hepatobiliary cells but no overt increase in reactive ductular cells, while the highly fibrotic biliary atresia is characterized by an increase in reactive ductular and hepatic progenitor cells (Fabris et al, 2007). Reactive ductular cells are pro-fibrotic (Banales et al, 2019), and their absence in ALGS may thus limit bridging or periportal fibrosis (Fabris et al, 2007). The transplantation experiments described here show that *Jag1*-compromised lymphocytes also contribute to attenuated periportal fibrosis. However, Notch signaling modulates fibrosis via multiple other cell types in the liver, which may also play a role in ALGS. Notch activation in hepatocytes drives fibrosis in the context of NAFLD, and its inhibition attenuates NAFLD-associated fibrosis (Schwabe et al, 2020). Hepatocytic Jag1 also contributes directly to liver fibrosis and is required for NASH-induced liver fibrosis (J Yu et al, 2021) and is required for NASH-induced liver fibrosis (J Yu et al, 2021). Notch activation in HSCs (Bansal et al, 2015), or liver sinusoidal endothelial cells can also aggravate hepatic fibrosis (Duan et al, 2018). Intriguingly, Notch signaling is not required for methionine-choline deficient (MCD) diet-induced fibrosis (Zhu et al, 2018), demonstrating that fibrosis can be Notch-dependent or -independent, based on the nature of the hepatic

insult. Thus, while $Jag1^{Ndr/Ndr}$ lymphocytes are sufficient to mitigate periportal liver fibrosis in a model of biliary injury with otherwise normal Notch signaling (Fig. 6), it is likely that multiple additional Notch-regulated mechanisms contribute to the fibrotic process in ALGS and remain to be investigated in terms of their relative contribution.

Splenomegaly has been described as a consequence of portal hypertension in ALGS (Kamath et al, 2020), but could also be attributed to immune-related pathology (McKenzie et al, 2018). $Jag1^{Ndr/Ndr}$ mice exhibit splenomegaly as early as P10, which is exacerbated at P30 (Fig. 1F,G). Patients with other liver diseases display portal hypertension and cirrhosis, with both splenomegaly and hypersplenism associated with a high $CD4^+/CD8^+$ ratio, but a low $Treg^+/CD4^+$ ratio (Nomura et al, 2014). However, $Jag1^{Ndr/Ndr}$ mice present with splenomegaly but not hypersplenism. An overactive spleen (hypersplenism) would remove red blood cells which are instead enriched in $Jag1^{Ndr/Ndr}$ mice, and Tregs were enriched in $Jag1^{Ndr/Ndr}$ mice, not depleted as seen in cirrhosis/hypersplenism. These data are thus consistent with portal hypertension-induced splenomegaly rather than hypersplenism.

In sum, we described and experimentally tested the function of the immune system in a model of ALGS. Our results demonstrate that *Jag1* mutation concurrently impacts hepatocyte maturity and adaptive immunity, and thus modulates liver fibrosis via multiple axes. We suggest that the course of liver disease in ALGS may be determined by the interaction between hepatic and immune system developmental defects. These results provide new insights into how Jag1 controls fibrosis in ALGS via interaction between hepatocytes and T cells.

# Methods

### Reagents and tools table

| Reagent/Resource | Reference or Source | | Identifier or Catalog Number |
|---|---|---|---|
| **Experimental Models** | | | |
| Jag1^Ndr (C3HeB/FeJ-Jag1Ndr/Ieg) | (Andersson et al, 2018) | | EMMA ID - EM:13207 |
| Rag1^−/− | Jackson Laboratories | | Jax: 002216 |
| **Recombinant DNA** | | | |
| n.a. | n.a. | | n.a. |
| **Antibodies** | | | |
| **Antibodies used in immunohistochemical or western blot analysis** | | | |
| Anti-mouse-HRP | Abcam | WB 1:20,000 | #ab205719 |
| Anti-rabbit-HRP | Abcam | WB 1:20,000 | #ab205718 |
| Donkey anti-Rat IgG (H + L) Highly Cross-Adsorbed (also Ms) Secondary Antibody, Alexa Fluor™ 647 | ThermoFisher | IF 1:250 (1:500 for the FOXP3, ECAD, LCK staining) | A78947 |
| Anti-rabbit-AlexaFluor-488 | Jackson ImmunoResearch | IF 1:250 | 711-545-152 |
| Anti-mouse-RhodamineRedX | Jackson ImmunoResearch | IF 1:250 | 715-295-150 |

| Reagent/Resource | Reference or Source | | Identifier or Catalog Number |
|---|---|---|---|
| Anti-β-actin | Santa Cruz Biotechnology | WB 1:2000 | #sc-47778 |
| Lck | Kind gift from Dominik Filipp | WB/IF 1:1000 | (Veillette et al, 1988) |
| Tyr505Lck | Cell Signaling | WB 1:1000 | #2751 |
| aSMA - paraffin sections | Dako | IF 1:50 | M0851 |
| aSMA - cryosections | Sigma-Aldrich | IF 1:500 | A2547 |
| Collagen I - paraffine sections | Abcam | IF 1:100 | ab21286 |
| ECAD - cryosections | BD Biosciences | IF 1:100 | 610181 |
| FOXP3 - paraffine sections | Thermo Fisher | IF 1:500 | FJK-16s - # 14-5773-80 |
| FOXP3 - cryosections | BD | IF 1:500 | 562996 |
| CK19 - paraffine sections | DSHB | IF 1:250 | TROMA-III RRID: AB_2133570 |
| **Antibodies used in E9.5 flow cytometry.** | | | |
| CD41 | Biolegend | 1:100 | #133903; MWReg30 |
| CD117 | Invitrogen | 1:50 | #17-1172-83; ACK2 |
| Viability dye | Thermo Fisher | 1:1000 | #L34957 |
| TER119 | Invitrogen | 1:100 | #35-5921-82; TER-119 |
| FcRy | Biolegend | 1:50 | #101317; 93 |
| **Antibodies used in 25-color flow cytometry.** | | | |
| CD45 | BD Biosciences | 1:100 | 30-F11 |
| CD31 | BD Biosciences | 1:100 | 390 |
| CD49a | BD Biosciences | 1:400 | Ha31/8 |
| TCR-beta | BD Biosciences | 1:100 | H57-597 |
| TER119 | BD Biosciences | 1:200 | TER-119 |
| CD11b | BD Biosciences | 1:200 | M1/70 |
| EpCAM | BD Biosciences | 1:100 | G8.8 |
| Ly6C | BD Biosciences | 1:200 | AL-21 |
| CD34 | Biolegend | 1:50 | MEC14.7 |
| Streptavidin | BD Biosciences | 1:400 | NA |
| Tim-4 | BD Biosciences | 1:100 | RMT4-54 |
| Jag1 | Biolegend | 1:100 | HMJ1-29 |
| CD3 | Biolegend | 1:200 | 17A2 |
| MHC-II | Biolegend | 1:400 | I-A/I-E |
| CD117 | Biolegend | 1:50 | 2B8 |
| Viability dye | Thermo Fisher | 1:100 | NA |
| CD4 | Biolegend | 1:400 | RM4-5 |
| Ly6G | BD Biosciences | 1:200 | 1A8 |
| CD19 | Biolegend | 1:200 | 6D5 |
| Siglec-F | BD Biosciences | 1:200 | E50-2440 |

| Reagent/Resource | Reference or Source | | Identifier or Catalog Number |
| --- | --- | --- | --- |
| CD71 | BD Biosciences | 1:100 | C2 |
| CD64 | BD Biosciences | 1:100 | X54-5/71 |
| Nestin | Thermo Fisher | 1:400 | 307501 |
| NKp46 | Biolegend | 1:50 | 29A1.4 |
| F4/80 | Biolegend | 1:100 | BM8 |
| CD8 | Thermo Fisher | 1:400 | 53-6.7 |
| CD11c | Biolegend | 1:200 | N418 |
| TruStain FcX (CD16/32) | Biolegend | 1:100 | 93 |
| Brilliant Stain Buffer Plus | BD Biosciences | 1:5 | N/A |
| **Antibodies used P30 stage and transfer experiments.** | | | |
| CD4 | Biolegend | 1:400 | BV785 |
| CD44 | Biolegend | 1:800 | BV711 |
| CD8 | Biolegend | 1:400 | Percp-Cy5.5 |
| TCR-B | Biolegend | 1:200 | APC-Cy7 |
| Viability dye 506 | Thermo Scientific | 1:1000 | # 65-0866-14 |
| Foxp3 | eBioscience | 1:400 | BV421 |
| **Oligonucleotides and other sequence-based reagents** | | | |
| n.a. | n.a. | | n.a. |
| **Chemicals, Enzymes and other reagents** | | | |
| RPMI | Sigma-Aldrich | | #R8758 |
| DSS | Sigma-Aldrich | | #D8906 |
| PFA | VWR | | #20909.290 |
| TrueView | Vector Laboratories | | SP-8500-15 |
| Fluorescent mounting medium | Dako | | S3023 |
| RIPA Lysis and Extraction Buffer | Thermo Scientific | | #89900 |
| Protease and phosphatase inhibitor | Thermo Scientific | | #A32961 |
| Pierce™ BCA Protein Assay Kit | Thermo Scientific | | #23225 |
| Amersham™ Hybond® PVDF membranes | Sigma-Aldrich | | # GE10600021 |
| Pierce™ ECL Western Blotting Substrate | Thermo Scientific | | #32209 |
| HBSS | Gibco | | #14175095 |
| Dispase | Gibco | | #17105041 |
| FCS | Gibco | | #10270106 |
| EDTA | Thermo Scientific | | #AM9262 |
| fixation buffer from eBioscience™ FOXP3/ Transcription Factor Staining Buffer Set | eBioscience | | #00-5523-00 |

| Reagent/Resource | Reference or Source | Identifier or Catalog Number |
| --- | --- | --- |
| Dead Cell Removal Kit | Miltenyi | #130-090-101 |
| DPBS -Mg$^{2+}$,-Ca$^{2+}$ | Gibco | #D8537 |
| Liver digest medium | Gibco | #17703034 |
| TrypLE Express | Gibco | #12604013 |
| 0.5M EDTA pH 8.0 | Thermo Scientific | #R1021 |
| BSA | Sigma-Aldrich | #A7906 |
| collagenase D | Roche | #11088858001 |
| dispase II | Gibco | #17105041 |
| DNase I | Sigma-Aldrich | #4716728001 |
| Percoll | Sigma-Aldrich | #GE17-0891-01 |
| Chromium™ Single Cell 3′ Library and Gel Bead Kit v2 | 10X Genomics | PN-120237 |
| Bioanalyzer High Sensitivity DNA kit | Agilent | #5067-4626 |
| **Software** | | |
| Zen | Carl Zeiss | V3.4 |
| Fiji | https://github.com/fiji/fiji | v2.9.0 |
| FlowJo software | BD | V10.7.1 |
| **Other** | | |
| 40 µm cell strainer | BD | #352350 |
| 96-well V-bottom plates | Sigma-Aldrich | #BR781601 |
| 2 ml Eppendorf DNA LoBind® tubes | Merck | #EP0030108078 |
| Millex-GV Filter 0.22 µm | Merck-Millipore | #SLGVR33RS |
| EASYstrainer | Greiner | #542040 |
| 70 µm filter cap | Falcon | #38030 |

## Methods and protocols

### Mouse maintenance and breeding

All animal experiments were performed in accordance with ARRIVE guidelines, local rules and regulations and all experiments were approved by Stockholm's Norra Djurförsöksetiska nämnd (Stockholm animal research ethics board, ethics approval numbers: N50/14, N61/16, N5253/19, N2987/20) or Czech Academy of Sciences ethics board (approval number: AVCR 8362-2021). The *Jag1$^{Ndr/Ndr}$* mice were maintained in a mixed C3H/C57bl6 genetic background as reported previously (Andersson et al, 2018) and are deposited in the European Mouse Mutant Archive EMMA: https://www.infrafrontier.eu/search?keyword=EM:13207. *Rag1*-deficient animals were purchased from Jax (Jax: 002216) and maintained in house, in compliance with the FELASA guidance in individually ventilated cages. Animals were maintained with standard day/night cycles, provided with food and water ad libitum, and were housed in cages with enrichment. Males and females were bred overnight

and noon of day of plug was considered embryonic day (E) 0.5. Nodder mice were genotyped by the Transnetyx® (USA) automated qPCR genotyping company or by TaqMan qPCR assay. Both males and females were used in experiments as indicated in Source Data Table S10 (ST10). In certain cases (E9.5stage), sex was unfortunately not determined.

### Adoptive transfer of lymphocytes and DSS-induced colitis mouse model

Adult Rag1-deficient animals (Jax: 002216) were used as the acceptors of lymphocytes isolated from $Jag1^{+/+}$ or $Jag1^{Ndr/Ndr}$ mice. Spleens from $Jag1^{+/+}$ or $Jag1^{Ndr/Ndr}$ mice (littermates) were passed through 40-μm nylon mesh to obtain a single cell suspension, which was then incubated overnight in 10% complete RPMI (Sigma-Aldrich, #R8758) to remove adherent cells. After the incubation, the floating fraction containing lymphocytes was collected. One million lymphocytes were injected into the recipient via tail vein, in PBS. For the induction of colitis by DSS (Sigma-Aldrich, # D8906) based on (Chassaing et al, 2014), mice were given 2.5% DSS in drinking water for 1 week, followed by 10 days of recovery and subsequently one more week of DSS treatment and 7 days of recovery prior sacrifice and tissue collection. Scoring of the intestinal inflammatory status was performed on formalin-fixed, paraffin-embedded liver sections (4 μm) stained with hematoxylin and eosin (H&E) was assessed as described in (Erben et al, 2014). The whole intestinal section from each animal was quantified in a blinded way by three experimenters (Fig. EV4 and Fig. 5).

### Bile duct ligation model of cholestasis

The common bile duct ligation (BDL)-induced cholestasis in mice was performed as in (Tag et al, 2015) with the following modifications. Mice were anesthetized with ketamine (80 mg/kg) and xylazine (10 mg/kg) and the extrahepatic common bile duct was cut in between the sutures. Mice were sacrificed after 7 days for further analyses. Three animals were excluded from the BDL cohort as there was a ligation leakage (two $Jag1^{Ndr/Ndr} \rightarrow Rag1^{-/-}$, and one $Jag1^{+/+} \rightarrow Rag1^{-/-}$) after surgery, these criteria were pre-established.

### Immunohistochemical procedures

For antibodies see Reagents and Tools Table.

### Immunohistochemistry, hematoxylin, and Sirius red staining

Formalin-fixed, paraffin-embedded liver sections (4 μm) were stained with hematoxylin and eosin (H&E) or Sirius red (SR) (Chalupský et al, 2013). For immunofluorescence, paraffin sections were subjected to heat-induced antigen retrieval in citrate (pH 6) buffer, and incubated with primary antibodies overnight at 4 °C or 1 h/37 °C. Slides were incubated with secondary antibodies for 120 min at 22 °C.

### Liver section image acquisition

Mice > P10 were euthanized with $CO_2$ and mice < P10 were decapitated. The liver was immediately dissected out, and the left lateral lobe was cut in halves, one half was embedded in OCT and frozen on dry ice, and the other was fixed O/N in 4% PFA (VWR, #20909.290) in PBS, and processed for paraffin sectioning. Immuno-fluorescent staining was carried out on cryosections (14 μm) or paraffine sections (5 μm) as previously described (Andersson et al,

2018; Hankeova et al, 2021). The liver was imaged using the LSM 980 confocal microscope or Axioscan (Carl Zeiss).

Fresh frozen cryosections from the left lateral lobe were postfixed in 4% formaldehyde for 10 min, washed 3× in PBS, blocked and permeabilized in PBS containing 10% BSA and 0.3% Triton X-100 for 1 h at room temperature (RT). Tissue sections were then incubated with primary antibodies in a blocking solution overnight at 4 °C. Next, the sections were washed 3× with PBS, and incubated with fluorochrome-conjugated secondary antibody for 1 h at RT. Following the incubation, the sections were washed 3× with PBS, incubated with Hoechst 3342 for 10 min at RT, and washed again 3× with PBS and mounted in MOWIOL and imaged with the Axioscan using 20× objective (Carl Zeiss). For the quantification of LCK/FOXP3 immunofluorescence, 6–9 ROIs from each animal were quantified in a blinded way by three (EV4), or two (Fig. 6) experimenters.

SR-positive collagen deposits were measured from scanned whole liver sections (scanned using slide scanner Axio Scan.Z1, Carl Zeiss MicroImaging, Jena, Germany) and quantified by Fiji software. Keratin 19-, aSMA- and collagen-positive areas were measured from portal field (PF) view obtained with 40× objective (HCX PL APO 40x/0.75) from liver sections and quantified in Fiji software using IJ_IsoData (collagen), RenyiEntropy (aSMA), or IJ-IsoData (K19) thresholds.

Alfa-fetoprotein and Cyp1A2 staining on paraffin sections was executed as described in (Hankeova et al, 2021). For AFP immunostaining, the slides were additionally treated with the quenching agent TrueView (SP-8500-15, Vector Laboratories) to remove non-specific background signals. Slides were mounted with Dako Fluorescent mounting medium (S3023) and imaged on a Nikon/CrEST X-Light V3 Spinning Disk confocal microscope.

### Western blotting

Mouse liver tissue was weighed and homogenized in RIPA Lysis and Extraction Buffer (Thermo Scientific, #89900) with the Protease and phosphatase inhibitor (Thermo Scientific, #A32961), using 15–25 mg of tissue/200 μl. Tissue samples were disrupted using a chilled glass-pestle homogenizer. Samples were then agitated for 2 h at 4 °C, and subsequently centrifuged at 16,000 rcf for 20 min at 4 °C. The supernatant was collected, and the protein concentrations determined using the Pierce™ BCA Protein Assay Kit (Thermo Scientific, #23225). For western blotting, aliquots of 25 μg were denatured by boiling in Laemmli Sample Buffer (2×), separated by SDS-PAGE, and transferred onto PVDF membranes (Amersham™ Hybond®, # GE10600021) by electro-blotting. The membrane was cut, blocked by 5% non-fat milk in 1X TBST (Tris-buffered saline, 0.1% Tween 20) for 1 h/RT, and incubated ON/4 °C with a Rabbit anti-Lck, and Mouse anti-β-actin antibodies. The following day, membranes were washed 4×/10 min in 1X TBST and incubated with corresponding HRP-conjugated secondary antibodies for 1 h/RT, followed by 4×/10 min wash in 1X TBST. Chemiluminescence was detected with Pierce™ ECL Western Blotting Substrate (Thermo Scientific, #32209) on ChemiDoc machine (Bio-Rad).

### Sample processing and flow cytometry analysis of embryo proper and yolk sac at E9.5

Primary antibody mixes for E9.5 analysis are in Reagents and Tools Table.

For dissociation and flow cytometry, we followed a protocol described by Balounová and colleagues (Balounová et al, 2019). In brief, E9.5 embryo proper (EP) and yolk sac (YS) were dissected and dissociated separately. After briefly washing in cold Hank's balanced saline solution (HBSS) (Gibco, #14175095), EP and YS were incubated with 1 mg/mL Dispase (Gibco, #17105041) in HBSS for (~10 min for EP, ~17 min for YS) in 1.5 mL Eppendorf tubes in a water bath at 37 °C, and occasionally mixed by gentle pipetting. The reaction was stopped by washing in HBSS with 2% FCS (Gibco, #10270106). Suspensions were passed through a 40 µm cell strainer (BD, #352350) and centrifuged at $350 \times g$/7 min/4 °C. Pelleted cells were resuspended in 200 µl flow cytometry wash (DPBS with 2% FBS, 2 mM EDTA, Thermo Scientific, #AM9262) and transferred to 96-well V-bottom plates (Sigma-Aldrich, #BR781601) for further flow cytometric analysis. Primary antibody mixes (Reagents and Tools Table) were added and incubated for 30 min at 4 °C, followed by LIVE/DEAD Fixable Aqua Dead Cell Stain (Thermo Fisher, #L34957) incubation for additional 30 min at 4 °C. Cells were fixed for 15 min/RT with fixation buffer from eBioscience™ FOXP3/Transcription Factor Staining Buffer Set (eBioscience, #00-5523-00). Acquisition was performed on LSRII flow cytometer (BD Biosciences). All data were analyzed using FlowJo software (FlowJo, V10.7.1).

### E16.5 and P3 sample preparation for 10X sequencing and flow cytometry

Reagents for the single-cell isolation: Tabletop centrifuge with cooling; QuadroMACS separator (Miltenyi, #130-091-051); Dead Cell Removal Kit (Miltenyi, #130-090-101); sterile 5 ml Transfer Pipettes (Merc, #HS206371C-500EA); DPBS -$Mg^{2+}$,-$Ca^{2+}$ (Gibco, #D8537); HBSS -$Mg^{2+}$,-$Ca^{2+}$ (Gibco, #14175095); 2 ml Eppendorf DNA LoBind® tubes (Merck, #EP0030108078), Liver digest medium (LDM) (Gibco; #17703034) - unfreeze in 4C, protect from light, avoid freeze-thaw; TrypLE Express (Gibco, #12604013); Millex-GV Filter 0.22 µm (Merck-Millipore, #SLGVR33RS); 40 µm strainers (BD, #352350) or EASYstrainer (Greiner, #542040); 5 ml flow cytometry tubes with 70 µm filter cap (Falcon, #38030); FCS (Gibco, #10270106); 0.5M EDTA pH 8.0 (Thermo Scientific, #R1021); BSA (Sigma-Aldrich, #A7906).

Buffers:

E16.5—Wash Buffer (WB16-E) 1% BSA in HBSS, filter sterile (0.22 µm), Wash Buffer (WB16) 1% BSA, 1 mM EDTA, HBSS, filter sterile (0.22 µm).

P3—Wash Buffer (WBP3-E) 1% FCS in HBSS, filter sterile (0.22 µm), Wash Buffer (WBP3) 1% FCS, 1 mM EDTA, HBSS, filter sterile (0.22 µm).

Embryo stage was determined based on the presence of the vaginal plug the morning after breeding initiation, and the noon of the day of plug was considered as E0.5. All embryos at E16.5 and P3 pups from the *Jag1^{Ndr/+}* x *Jag1^{Ndr/+}* breeding were sacrificed by decapitation, their livers (and spleens at P3) dissected out and placed in 12- or 24-well plates with ice-cold DPBS on ice. The sex of P3 pups was determined based on scrotum pigmentation (Wolterink-Donselaar et al, 2009). Material for genotyping was collected (tail or limb tissue) and processed for genotyping using TaqMan probe (Thermo Fisher) yielding results in 3 h (P3 *Jag1^{Ndr/Ndr}* pups were identified based on the presence of jaundice and random *Jag1^{Ndr/+}* and *Jag1^{+/+}* controls genotypes were confirmed afterward). The tissue was kept in ice-cold DPBS for 30 min before being transferred into 5 ml flow cytometry tubes with

1.5 ml LDM + 0.5 ml of TrypLE Express (E16.5) or 2 ml LDM (P3 stage), pipetted 5× through sterile wide-bore blue tip (E16.5) or transfer pipettes (P3) and incubated at 37 °C/5 min in a water bath (inverting each tube after 2–3 min). After the 5 min incubation, the tissue was disrupted further by gentle pipetting 10× using a filtered 1 ml tip, followed by an additional round of incubation at 37 °C/5 min in a water bath (inverting each tube after 2–3 min). The enzymatic digestion was terminated by the addition of 1.5 ml of WB16-E/WBP3-E and additional mechanical disruption of the remaining tissue by pipetting each sample 20× using filtered 1 ml wide-bore blue pipette tip. The suspension was passed through 70 µm filters into flow cytometry tubes, the filter was rinsed once with 0.5 ml WB16/WBP3, and the filtrate was passed through a 40 µm strainer into two 2 ml Eppi tubes.

The duplicates of E16.5/P3 samples were centrifuged at $100 \times g$ for 5 min at 5 °C. Each supernatant was transferred to two new tubes, while the pellets were resuspended in 200 µl of Dead Cell Removal MicroBead suspension in Binding buffer and the resuspended pellets were further incubated for 5 min at RT. In the meantime, the supernatant from the previous step was centrifuged at $250 \times g$ for 5 min at 5 °C. The new supernatant was discarded, and the corresponding pellets were mixed with 200 µl of Dead Cell Removal MicroBeads with the previous cell suspension in Binding buffer. The mixture was incubated for 10 min at RT. After the incubation 300 µl 1× Binding Buffer was added to the cell suspension to obtain a final volume of 500 µl, suitable for magnetic separation with MS columns. The separation was done based on the manufacturer's recommendations. Briefly, we rinsed the columns with 500 µl of 1× Binding Buffer, applied the cell suspension onto the column, and collected the flowthrough containing unlabeled cells (living cells). Columns were further washed three more times with 500 µl of 1× Binding Buffer, flowthrough was collected and 20 µl filter sterile 10% BSA was added to the purified cells. Resulting samples were divided in two, whereof 250 µl was used for scRNAseq, following the 10x Genomics® Single Cell Protocol, and the remainder was used for the flow cytometry as described below.

The duplicate P3 samples were centrifuged at $100 \times g$ for 5 min at 5 °C, and supernatants were transferred to new tubes. The pelleted cells from each duplicate were pooled and resuspended in 1.5 ml of the WBP3-E buffer and transferred to a new 2 ml tube. In the meantime, the supernatants from the previous step were centrifuged at $250 \times g$ for 5 min at 5 °C. The supernatants were discarded, and the corresponding pelleted cells were resuspended in 0.5 ml of the WBP3-E buffer and added to the resuspended cells obtained from the first centrifugation. The full sample cell suspensions (now in 2 ml of WBP3-E buffer) were then transferred into a new protein lo-bind tube through 40 µl cell strainer to remove any remaining cell debris or large clumps. We then determined the cell concentration using a Countess® II Quantification to calculate the appropriate volume for the subsequent resuspension to obtain the target concentrations of ~1200 cells/µl as recommended by the 10x Genomics® Single Cell Protocol. The whole procedure including tissue collection, genotyping and dissociation took 3 h, before initiation of processing with 10x microfluidics chromium and/or for flow cytometry.

### Flow cytometry E16.5, P3—staining

Primary antibody mixes for E16.5 and P3 analysis are in Reagents and Tools Table.

The flow cytometry was performed based on (Filipovic et al, 2019). Single cell suspensions (cells) from liver and spleen were washed twice and resuspended in flow cytometry buffer (PBS with 2 mM EDTA and 2% FBS), filtered through a 100 μm strainer (BD Falcon) and stained in 96-well V-bottom plates. Unless otherwise stated, staining steps were performed with antibodies diluted according to the Reagents and Tools Table in 50 μl of the flow cytometry buffer, at 4 °C in the dark, and washing steps were performed by resuspending the sample in 150 μl of the flow cytometry buffer and centrifuging plates for 5 min at $500 \times g$ at 4 °C. The BD Horizon Brilliant Stain Buffer Plus was added to the antibody mix in every step when the BD Horizon Brilliant dyes were used. Cells were pre-incubated with TruStain FcX (anti-CD16/32) to block Fc receptors for 10 min at room temperature. Cells were first stained with anti-CD64 antibody for 45 min. Next, cells were stained with antibodies against other surface antigens (Reagents and Tools Table) for 30 min, followed by two washes in flow cytometry buffer. Cells were then stained with the LIVE/DEAD Fixable Aqua Dead Cell Stain (Thermo Fisher) and fluorescently conjugated streptavidin for 30 min. This was followed by two washes in flow cytometry buffer. Next, cells were fixed for 45 min in 100 μl fixation/permeabilization working solution from eBioscience Foxp3/Transcription Factor Staining Buffer kit (Thermo Fisher) at room temperature. After this step, cells were washed in 1x permeabilization buffer from the same kit. Finally, cells were stained with antibodies against intracellular antigens diluted in 1x permeabilization buffer from the same kit for 30 min at room temperature. Samples were then washed twice in 1x permeabilization buffer and resuspended in flow cytometry buffer for acquisition. For single-stained compensation controls, Ultra-Comp eBeads Compensation Beads (Thermo Fisher) were used according to manufacturer's instructions. FACSymphony A5 flow cytometer (BD Biosciences) was used for acquisition. The instrument was equipped with the following lasers: UV (355 nm), violet (405 nm), blue (488 nm), yellow/green (561 nm), and red (637 nm).

### Flow cytometry E16.5, P3—analysis

FCS3.0 files were exported from the FACSDiva and imported into FlowJo v.10.7.1 for analysis. The following plugin versions (from FlowJo Exchange) were used: FlowAI (2.1), DownSample (3.2), UMAP (3.1), PhenoGraph (3.0). The data were pre-processed using FlowAI (parameters selected: all checks, second fraction FR = 0.1, alpha FR = 0.01, maximum changepoints = 3, changepoint penalty = 500, dynamic range check side = both) to remove any anomalies present in FCS files. A compensation matrix was generated using AutoSpill in FlowJo. Events were downsampled from the live cell gate from all samples using DownSample, and categorical values were added to the downsampled populations (e.g., genotype, tissue) prior to concatenation so that groups of interest could be deconvoluted during the analysis. UMAP and PhenoGraph were run using all parameters from the panel except BV510. As over- and under-represented input groups are similarly weighted in the PhenoGraph output clusters, PhenoGraph results were normalized to account for the total number of cells from each input group. Some figures were generated in RStudio (version 1.3.959) using RColorBrewer (v1.1-2), ggplot2 (v3.2.1 and v3.3.0), tidyr (v.1.0.2), reshape2 (v.1.4.3), and pheatmap (v.10.12).

### Flow cytometry analysis of thymic epithelial cells

Thymi from 4-week-old $Jag1^{+/+}$ or $Jag1^{Ndr/Ndr}$ mice were minced by scissors into small pieces and dissociated by enzymatic digestion for 30 min at 37 °C using 0.3 mg ml$^{-1}$ collagenase D (Roche), 1 mg ml$^{-1}$ dispase II (Gibco), and 10 ng ml$^{-1}$ DNase I (Sigma-Aldrich) in RPMI supplemented with 2% FCS. Percoll density gradient centrifugation was performed to enrich for thymic epithelial cells. Cells were resuspended in 2 ml of 1.115 g ml$^{-1}$ isotonic Percoll (Sigma-Aldrich) and placed at the bottom of a tube. Subsequently, 1 ml of isotonic 1.065 g ml$^{-1}$ Percoll and then 1 ml of PBS were layered on top. The Percoll gradient was run at 2700 rpm ($1451 \times g$), at 4 °C, with no acceleration or brake for 30 min in the Eppendorf 5804 R centrifuge with S–4–72 rotor. The thymic epithelial cells were collected and subjected to flow cytometric analysis using the following set of antibodies for extracellular staining: CD45 APC-Cy7, EpCAM, APC, Ly6d PB, MHCII PE (all Biolegend). Intracellular staining of Aire AF488 (Invitrogen) was done after fixation and permeabilization of cells by the Foxp3/Transcription factor fixation buffer set (Thermo Scientific) according to manufacturer's instructions. Extracellular cell staining for flow cytometry analysis was performed at 4 °C, in the dark, on ice, for 30 min, while the fixation and intracellular staining was performed at room temperature, in the dark for 40 min (each step). Dead cells were excluded by addition of Viability dye 506 (Thermo Scientific) prior to fixation. Samples were analyzed using a BD LSRII flow cytometer and FlowJO software version 10.7 (BD).

### Flow cytometry analysis of the T cell compartment

Spleens and thymi of $Jag1^{+/+}$ or $Jag1^{Ndr/Ndr}$ mice were passed through 40 um nylon mesh to obtain a single-cell suspension. The following antibodies were used for T cell analysis: CD4 BV785, CD44 BV 711, CD8 Percp-Cy5.5, TCR-B APC-Cy7 (all Biolegend) (Reagents and Tools Table). Extracellular cell staining for flow cytometry analysis was performed at 4 °C, in the dark, on ice, for 30 min. Dead cells were excluded from analysis by addition of Viability dye 506 (Thermo Scientific) prior to the fixation. The fixation was performed at room-temperature in dark for 40 min using Foxp3/Transcription factor fixation buffer set (Thermo Scientific). Foxp3 BV421 (eBioscience) staining was done at room temperature in dark for 40 min. Samples were analyzed using BD LSRII flow cytometer and FlowJO software version 10.7 (BD).

### Library preparation and sequencing

The E16.5 and P3 samples were processed on a Chromium microfluidics platform (10X Genomics) using the Chromium™ Single Cell 3' Library and Gel Bead Kit v2 (10X Genomics, # PN-120237) and the Chromium™ Single Cell A Chip Kit (10X Genomics, #PN-120236) following the manufacturer's instructions. For the E16.5 library preparation, validation with Bioanalyzer High Sensitivity DNA kit (Agilent, #5067-4626), and sequencing with NovaSeq (100 cycles) platform was performed by the SciLifeLab, Stockholm. For the P3 samples, library preparation, validation with Bioanalyzer High Sensitivity DNA kit (Agilent, # 5067-4626), and sequencing with NextSeq550 (75 cycles) or S4 Novaseq platform was performed by the *Bioinformatics and Expression analysis core facility* (BEA), Huddinge, Stockholm. All scRNAseq data are available online (GSE236483).

### Analysis of the scRNA seq using Seurat packages

Demultiplexing, quality control, and gene expression data counts were performed using CellRanger (Chromium), by the BEA and SciLife core facilities. Matrices were further processed and analyzed with R (version 4.1.2) in R Studio. Prior to transformation into Seurat Objects, matrices were precleared using SoupX (Young and Behjati, 2020) with default settings of setContaminationFraction parameter (sc) for the P3, and (sc, 0.13) for the E16.5 stages. All bioinformatic analyses were performed using Seurat, and doublet removal was executed with scDblFinder (Germain et al, 2022), QC cutoffs for E16.5 datasets were nFeature_RNA > 500 & nFeature_RNA < 7000 & percent.mt < 10, for P3 a range of nFeature_RNA > 300–500 & nFeature_RNA < 4500–6000 & percent.mt < 10 were used, followed by SCT based integration of the E16.5 and P3 $Jag1^{+/+}$, $Jag1^{Ndr/+}$, and $Jag1^{Ndr/Ndr}$ datasets (Butler et al, 2018). The high prevalence of erythrocytes and erythroblasts in embryonic/neonatal liver may have contributed ambient RNA to the droplet-based scRNA-seq: despite in silico free-mRNA decontamination and doublets removal, erythrocyte signatures persisted in a subset of hepatoblasts/hepatocytes and Kupffer cells. In total, 183,542 cells passed through the cross-contamination adjustment, quality control filtering and doublet removal in each of the duplicates per genotype and stage, analyzed in Seurat (Butler et al, 2018) (Figs. 2A,B and EV1A–C). A mean of 1520 genes, 6703 reads, and 3.0% of mitochondrial mRNA content were detected per cell. For comparative transcriptomic analysis, logNorm and scaled RNA data were used (Guilliams et al, 2022). The persistent erythroid signature in the Hepatoblast/cyte I and Hepatoblast/cyte II cells was removed by subtracting the top 40 mRNA markers of Erythroblasts, Erythrocytes and B cells (Source Data Table ST9). For the hepatoblasts/hepatocyte DGE analysis (Source Data Tables S1–5) we analyzed each cluster separately using the Seurat Find Markers function.

### Deconvolution of bulk RNA seq using MuSiC packages

To estimate cell type proportions in $Jag1^{Ndr/Ndr}$ livers compared to $Jag1^{+/+}$ livers at P10, we performed cell type deconvolution of our previously generated bulk RNA-seq datasets GSE104875 (Andersson et al, 2018) employing the MuSiC package for Multi-subject Single-cell Deconvolution (X Wang et al, 2019) that uses single-cell RNA-seq data as a reference for deconvolution.

Deconvolution using an annotated scRNAseq dataset available under GEO accession number GSE171993 (Liang et al, 2022): The dataset was imported into R 4.1.0 package Seurat 4.0.5 (Hao et al, 2021) and stages P1, P3, and P7 were subset. The original annotation accounted for 30 cell types in liver scRNA-seq data (Liang et al, 2022). We filtered for high-quality cells containing <5% of mitochondrial genes expressed per cell and 200–4000 genes per cell, and for genes that are expressed in at least 3 cells, resulting in a dataset of 32,967 cells and 23,170 genes. The Seurat object was converted to an Expression set, required as input in MuSiC using a function SeuratToExpressionSet from BisqueRNA 1.0.5 R package (Jew et al, 2020). MuSiC was performed using music_prop function. To analyze the differentiation signature of hepatocytes, we further subset neonatal hepatocytes (P1, P3, P7) from scRNA-seq dataset (1281 cells), converted to Expression set and used this as a scRNA-seq reference for MuSiC on the P10 bulk RNA-seq dataset.

Deconvolution using a time-course mSTRT scRNA-seq dataset on isolated albumin-positive cells (GSE209749) (L Yang et al, 2023): The analysis was done in R version 4.3.0 with updated MuSiC v1.0.0 that supports SingleCellExperiment class as single cell reference. The scRNA-seq count matrix "GSE209749_readcount.mSTRT-seq.csv.gz"

was downloaded from GEO and loaded into R. The time-course dataset was subset (removing P60_2N, P60_4N, and P60_8N cells belonging to a different experiment) and basic quality control was performed with SingleCellExperiment (Amezquita et al, 2020), scater (McCarthy et al, 2017) and scran packages (Lun et al, 2016). Cells with low library size and low number of detected genes were filtered out, leaving 1783 cells in the time-course dataset stored as SingleCellExperiment object. Estimated proportions of different stages (E17.5–P60) were generated with music_prop function applied to the P10 bulk RNAseq matrix.

### Gene set enrichment analysis (GSEA) for hepatoblast and hepatocyte signature

To derive hepatoblast and hepatocyte gene expression signatures we used GEO bulk RNA-dataset dataset GSE176069. This dataset comprises primary $Dlk1^+$ hepatoblasts from E14.5 livers and primary hepatocytes from adult C57BL/6JOlaHsd livers (Harlan Laboratories/Envigo) or C57BL6/JRj (Janvier Labs) mice (Belicova et al, 2021). The provided raw counts matrix was analyzed in R (version 4.1.0) with DESeq2 (Love et al, 2014) (version 1.34.0) package to identify differentially expressed genes using Wald test. The genes were sorted by p-adjusted value (default Benjamini and Hochberg method) and log2 fold change. The top 500 most significant genes were considered markers of hepatoblasts and hepatocytes, respectively.

The top 500 markers were provided as gene lists to GSEA desktop tool (version 4.3.2, Broad Institute (Subramanian et al, 2005)) to analyze whether there is an enrichment of heptoblasts or hepatocyte signature in our previously generated bulk RNA-seq datasets GSE104875, and GSE104873) (Andersson et al, 2018) on P10 $Jag1^{+/+}$ and $Jag1^{Ndr/Ndr}$ livers, and control and ALGS patients, respectively. These datasets were processed with DESeq2 from raw counts to size factor normalized counts as input for the GSEA desktop tool. The genes were ranked using Signal2Noise metric and permutation test procedure ($n = 1000$) was done using the "gene_set" parameter. The results were replotted using ggplot2 (version 3.4.0) package in R. The top 50 enriched genes per signature identified by GSEA were visualized using heatmaps and DESeq2 processed datasets.

### Bulk RNA sequencing

Bulk RNA-seq datasets (GSE104875) and (GSE104873) were analyzed (Andersson et al, 2018), and scRNAseq dataset GSE171993 and GSE209749 (Liang et al, 2022; L. Yang et al, 2023) were used as references for cell type deconvolution. Files from GSE104875 and GSE104875 series were downloaded and converted to fastq files using SRA Toolkit (NCBI) and aligned using STAR aligner tool (Dobin et al, 2013) to the genomes of *Mus musculus* (GRCm38.p6) and *Homo sapiens* (GRCh38.p13), respectively (downloaded from NCBI). Reads aligning to gene exons were counted using featureCounts program from Subread package (Liao et al, 2014) Out of these, lists of differentially expressed genes (DEGs) between groups as well as normalized read counts were obtained using DESeq2 package (Love et al, 2014). Considering the study design, all genes showing *p*-value (adjusted by Benjamini–Hochberg method) <0.05 were further examined for the enrichment of functional processes. To identify differentially expressed pathways, gene list of DEGs were queried by clusterProfiler package (G. Yu et al, 2012) against Disease Ontology and

**The paper explained**

**Problem**

Patients with Alagille syndrome (ALGS) develop dispersed, pericellular liver fibrosis rather than bridging fibrosis, which is a common response to bile duct disease and cholestasis. Understanding this phenomenon could provide important clues for novel treatment options for liver fibrosis.

**Results**

First, we show that intrahepatic ALGS-like pericellular fibrosis is recapitulated in *Jag1^Ndr/Ndr* mice. Combining single-cell transcriptomics and flow cytometry, we identify dysregulation of maturing hepatocytes and T cells at the onset and propagation of fibrosis. Neonatal *Jag1^Ndr/Ndr* hepatocytes and patient liver samples exhibited an enriched hepatoblast signature, and repressed mature hepatocyte signature, suggesting that hepatocyte maturation and function is disrupted in patients and *Jag1^Ndr/Ndr* mice. The hepatic phenotype was accompanied by an overabundance of regulatory T cells, which could limit the extent of periportal fibrosis as demonstrated by transplantation of *Jag1^Ndr/Ndr* immune cells into immunodeficient mice, followed by surgically induced cholestasis.

**Impact**

These data demonstrate that Jag1 hypomorphism results in both developmental hepatic and immune defects, which interact to determine the fibrotic process in ALGS. These insights reveal potential cellular targets which could be modulated to alter the course of liver fibrosis in ALGS or other liver diseases.

MSigDB databases using overrepresentation DisGeNet and GSEA analysis.

## Statistical analysis

Statistical analyses of differences between genotypes were evaluated using two-sided unequal variance t-test or, when more than two conditions were compared, a one-way ANOVA was used, as appropriate and as described in figure legends and Source data. Pearson correlation was used for correlation analysis. Transcriptomic data was analyzed as described in the sections above. $P$-value (or adjusted $p$ value in the case of transcriptomic data) was considered statistically significant if $P < 0.05$ (*$P < 0.05$, **$P < 0.01$, ***$P < 0.0001$), unless otherwise specified in figure legend. Specific $P$-values are listed in figure legends and Source data. Sample size was calculated using the Resource equation method. indicating that 6 biological replicates would be suitable per condition when comparing two genotypes, and 4–6 mice per condition are suitable when comparing three genotypes. The majority of experiments include 5 or more biological replicates per condition. Due to high mortality of *Jag1^Ndr/Ndr* mice, and exclusion criteria being met, Fig. 6 includes fewer mice despite initial inclusion of satisfactory numbers of mice per condition.

### Graphics

Figure schematics and synopsis graphics were created with BioRender.com and assembled in Inkscape.

## Data availability

Primary datasets to main Figures, Source Data Checklist, and Source Data Tables S1–10 are available here: https://www.ebi.ac.uk/biostudies/studies/S-BSST1446, scRNAseq data can be found at GEO (GSM7548683) using the following link: https://www.ncbi.nlm.nih.gov/geo/query/acc.cgi?acc=GSM7548683.

The source data of this paper are collected in the following database record: biostudies:S-SCDT-10_1038-S44321-024-00145-8.

## Peer review information

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

## Acknowledgements

The data handling and initial computations were enabled by resources in projects [SNIC 2018/8-200 and SNIC 2020-16-189] provided by the Swedish National Infrastructure for Computing (SNIC) at UPPMAX, partially funded by the Swedish Research Council through grant agreement no. 2018-05973. We thank the core facility at Novum, BEA, Bioinformatics and Expression Analysis, which is supported by the board of research at the Karolinska Institute and the research committee at the Karolinska Hospital. We thank Vinicna Microscopy Core facility at CUNI for an excellent microscopy support. We thank Raoul Kuiper and the FENO core facility for the immunohistochemical staining of JAG1 in adult mice, and for discussions. We thank Stefaan Verhulst and Leo Van Grunsven for fruitful discussions of HSC biology and analyses which are not included in this version of the manuscript. We thank Katarina Kováčová for excellent technical support, Sandra De Haan for assistance with collection of biological material, and Roman Hillje for his easy-to-use templates for visualization in R https://romanhaa.github.io/projects/scrnaseq_workflow/. This work was supported by the following grant agencies: Czech Science Foundation (24-10622S), PRIMUS/21/SCI/006 project funded by Charles University Grant Agency, MSCA Fellowships CZ - Charles University

CZ.02.01.01/00/22_010/0002902, Wenner-Gren Foundation to JM, The European Association for the Study of the Liver (EASL: The Daniel Alagille Award to ERA and the Sheila Sherlock Post Doc fellowship to JM), funding from Karolinska Institutet (2-560/2015-280, 2-2110/2019-7, 2-195/2021), KI/SLL Center for Innovative Medicine (CIMED, 2-538/2014-29), the Swedish Research Council / Vetenskapsrådet (2019-01350) to ERA, Alex & Eva Wallström Foundation to ERA, Czech Science Foundation (21-21736S), and ID Project No. LX22NPO5102 - Funded by the European Union - Next Generation EU to MG. Cancerfonden fellowship for IF. LB was supported by Horizon Europe research and innovation program under the Marie Skłodowska-Curie Actions, grant agreement ID: 101057846. Charles University Grant Agency Junior Fund fellowship to DVO. DVO, AF, TB, and JD acknowledge support from Talking microbes - understanding microbial interactions within One Health framework (CZ.02.01.01/00/22_008/0004597). Finally, JD and TB are kindly supported by the Czech Science Foundation JUNIOR STAR grant (No. 21-22435M), Czech Science Foundation grant (No. 22-30879S), Charles University Grant Agency (No. PRIMUS/21/MED/003) and from Ministry of Education, Youth and Sports grant ERC CZ (LL2315).

## Author contributions

**Jan Mašek**: Conceptualization; Resources; Data curation; Software; Formal analysis; Supervision; Funding acquisition; Validation; Investigation; Visualization; Methodology; Writing—original draft; Project administration; Writing—review and editing. **Iva Filipovic**: Conceptualization; Data curation; Software; Formal analysis; Investigation; Methodology; Writing—review and editing. **Noémi Van Hul**: Resources; Data curation; Formal analysis; Supervision; Validation; Investigation; Writing—review and editing. **Lenka Belicová**: Data curation; Software; Formal analysis; Investigation; Visualization; Methodology; Writing—review and editing. **Markéta Jiroušková**: Data curation; Formal analysis; Investigation; Visualization; Methodology; Writing—review and editing. **Daniel V Oliveira**: Data curation; Formal analysis; Supervision; Validation; Investigation. **Anna Maria Frontino**: Data curation; Formal analysis; Validation; Investigation. **Simona Hankeova**: Resources; Formal analysis; Investigation; Writing—review and editing. **Jingyan He**: Resources; Formal analysis; Investigation. **Fabio Turetti**: Formal analysis; Investigation. **Afshan Iqbal**: Formal analysis; Investigation. **Igor Červenka**: Data curation; Software; Formal analysis. **Lenka Sarnová**: Methodology. **Elisabeth Verboven**: Resources. **Tomáš Brabec**: Formal analysis; Writing—review and editing. **Niklas K Björkström**: Conceptualization; Supervision; Funding acquisition; Writing—review and editing. **Martin Gregor**: Conceptualization; Supervision; Funding acquisition; Writing—review and editing. **Jan Dobeš**: Formal analysis; Funding acquisition; Investigation; Methodology; Writing—review and editing. **Emma R Andersson**: Conceptualization; Data curation; Formal analysis; Supervision; Funding acquisition; Validation; Visualization; Methodology; Writing—original draft; Project administration; Writing—review and editing.

Source data underlying figure panels in this paper may have individual authorship assigned. Where available, figure panel/source data authorship is listed in the following database record: biostudies:S-SCDT-10_1038-S44321-024-00145-8.

## Funding

## Disclosure and competing interests statement

The authors declare no competing interests. ERA has formerly collaborated with Travere, and Moderna, with no personal remuneration and no conflict of interest.

# Expanded View Figures

**Figure EV1.  Single-cell profiling of the developing and postnatal liver reveals a cell population shift from embryonic erythropoiesis to postnatal immune surveillance in *Jag1*$^{+/+}$, *Jag1*$^{Ndr/+}$, and *Jag1*$^{Ndr/Ndr}$ mice.**

(A) Violin plots of read counts (nCount_RNA), unique gene counts (nFeature_RNA), and percentage of mitochondrial mRNA (percent-mt) across the cell types identified in the *Jag1*$^{+/+}$, *Jag1*$^{Ndr/+}$, and *Jag1*$^{Ndr/Ndr}$ livers at E16.5 and P3 ($n = 2$ each). (B) Genotype, stage, or organ contribution to the composition of the scRNAseq dataset of E16.5 and P3 *Jag1*$^{+/+}$, *Jag1*$^{Ndr/+}$, and *Jag1*$^{Ndr/Ndr}$ mice. (C) Dot plot of the SCT-normalized mRNA marker expression of 78 DEGs characteristic for the individual cell types identified in (B). (D) Bar graph representing the relative proportion of parenchymal and Kupffer cells (clusters 1–6) versus non-parenchymal cells (clusters 7–20) in dissociated *Jag1*$^{+/+}$, *Jag1*$^{Ndr/+}$, and *Jag1*$^{Ndr/Ndr}$ livers at E16.5 and P3 ($n = 2$ each). Graph represents mean ± SD. (E) Graph depicting relative cell type contribution in dissociated *Jag1*$^{+/+}$, *Jag1*$^{Ndr/+}$, and *Jag1*$^{Ndr/Ndr}$ livers at E16.5 and P3 normalized to the total sample cell count.

▶

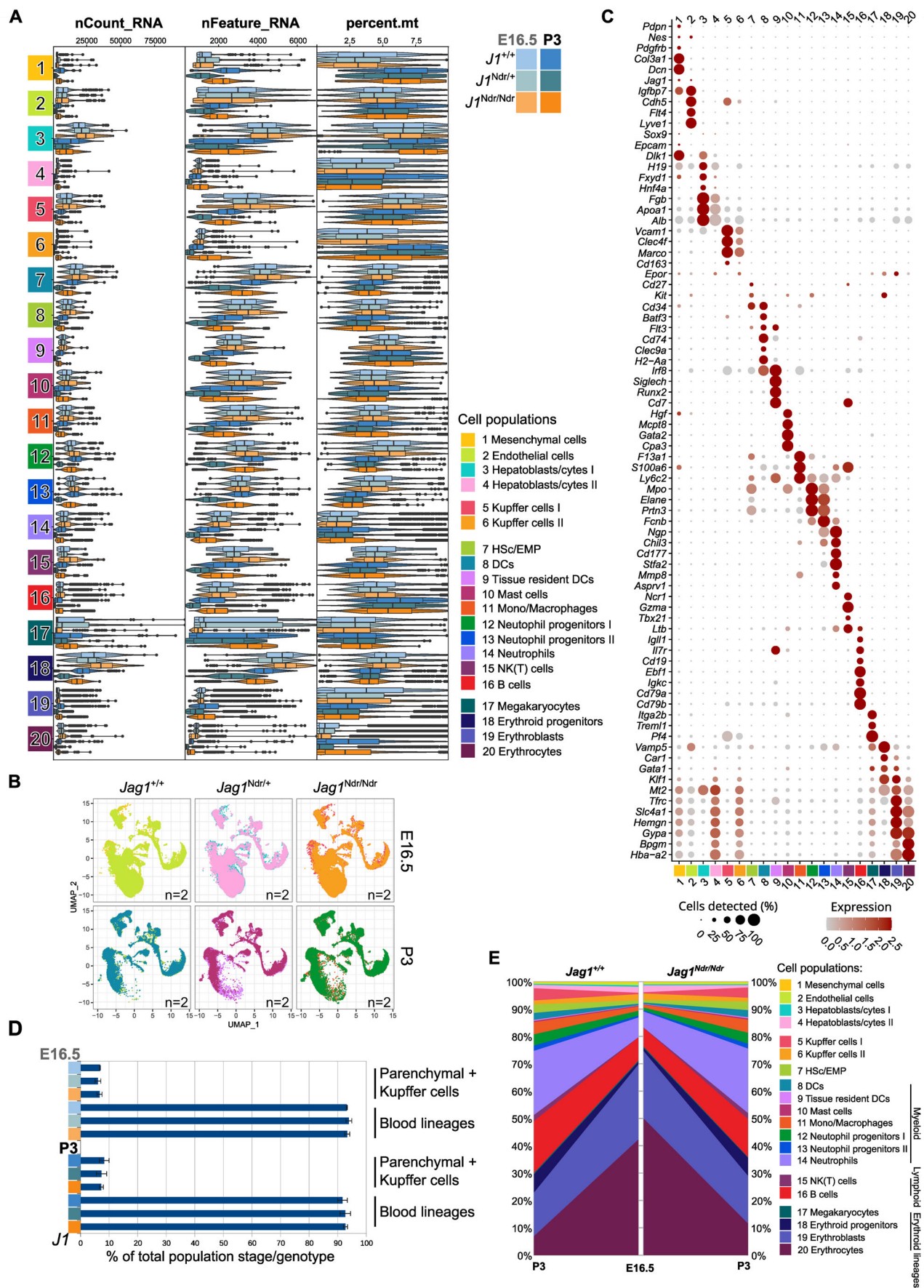

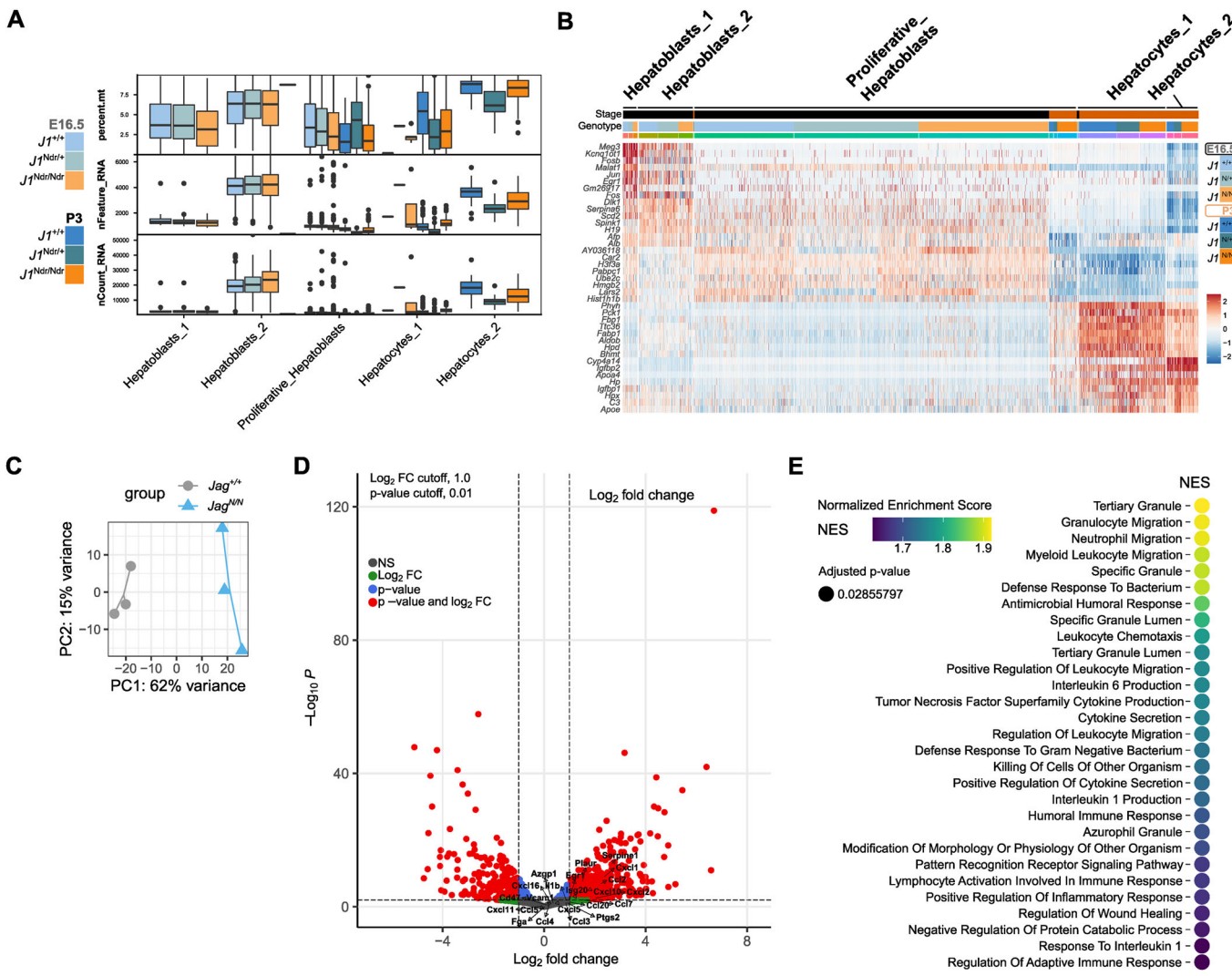

**Figure EV2. Altered gene expression profile of hepatocytes and whole livers in *Jag1^{Ndr/Ndr}* animals.**

(A) Box and whiskers plot of the percentage mitochondrial mRNA (percent.mt), unique gene counts (nFeature_RNA), and read counts (nCount_RNA) across the hepatocyte-like cell types identified in the *Jag1^{+/+}*, *Jag1^{Ndr/+}*, and *Jag1^{Ndr/Ndr}* livers by scRNA seq at E16.5 and P3 ($n = 2$ each). Center line indicates Median (50th percentile). The lower and upper bound of box indicate Q1, and Q3 (25th and 75th percentiles), respectively. Lower and upper whisker indicate min and max values within 1.5 * IQR below Q1 and above Q3, respectively. (B) Heatmap of the top 8 mRNA markers for each cluster. (C) PCA distribution of the bulk RNAseq samples from Andersson et al, 2018, re-analyzed in Fig. 2 and Fig. EV2. (D) Volcano plot of the DEGs of *Jag1^{Ndr/Ndr}* vs. *Jag1^{+/+}* liver at P10. Pro-inflammatory markers of activated hepatocytes are highlighted. Significance was confirmed via Benjamini–Hochberg $P$ value correction. (E) Top29 GO pathways enriched in the *Jag1^{Ndr/Ndr}* livers, NES (normalized enrichment score) indicates overlap of genes across the pathways. Significance was confirmed via false-discovery rate (FDR) statistic with 25% cutoff.

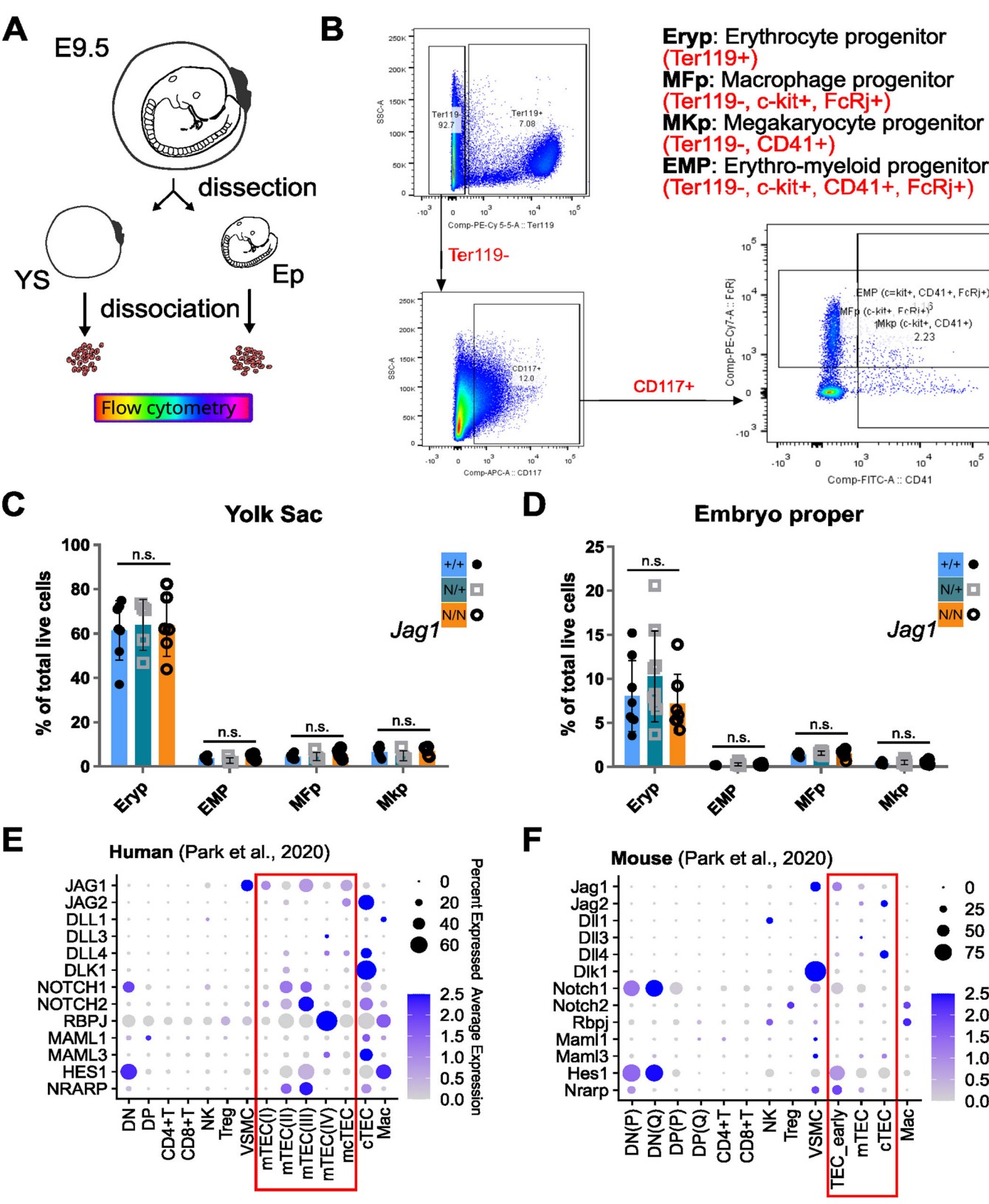

**Figure EV3.  The proportions of early embryonic hematopoietic progenitor populations are not altered in *Jag1^{Ndr/Ndr}* embryo proper (EP) and yolk sac (YS).**

(A) Schematic of the experiment. E9.5 embryos were harvested and yolk sac (YS) and embryo proper (EP) were processed separately for flow cytometry analysis. (B) Gating strategy for identification of the Erythrocyte progenitors (Eryp), Macrophage progenitors (MFp), Megakaryocyte progenitors (MKp), and Erythro-myeloid progenitors (EMP) after dead cell exclusion. (C, D) Relative proportion of live Eryp, EMP, MFp, and MKp cells from *Jag1^{Ndr/Ndr}* ($n = 7$), *Jag1^{Ndr/+}* ($n = 9$), and *Jag1^{+/+}* ($n = 8$) yolk sac (C) and whole E9.5 embryos (D). Graph represents mean ± SD. (E, F) Dot plot of the re-analyzed median scaled ln-normalized mRNA expression of Notch signaling components in human (E) and mouse (F') thymic cell populations from Park et al, 2020. One-way ANOVA, multiple comparison with Bonferroni method; n.s., no significant difference.

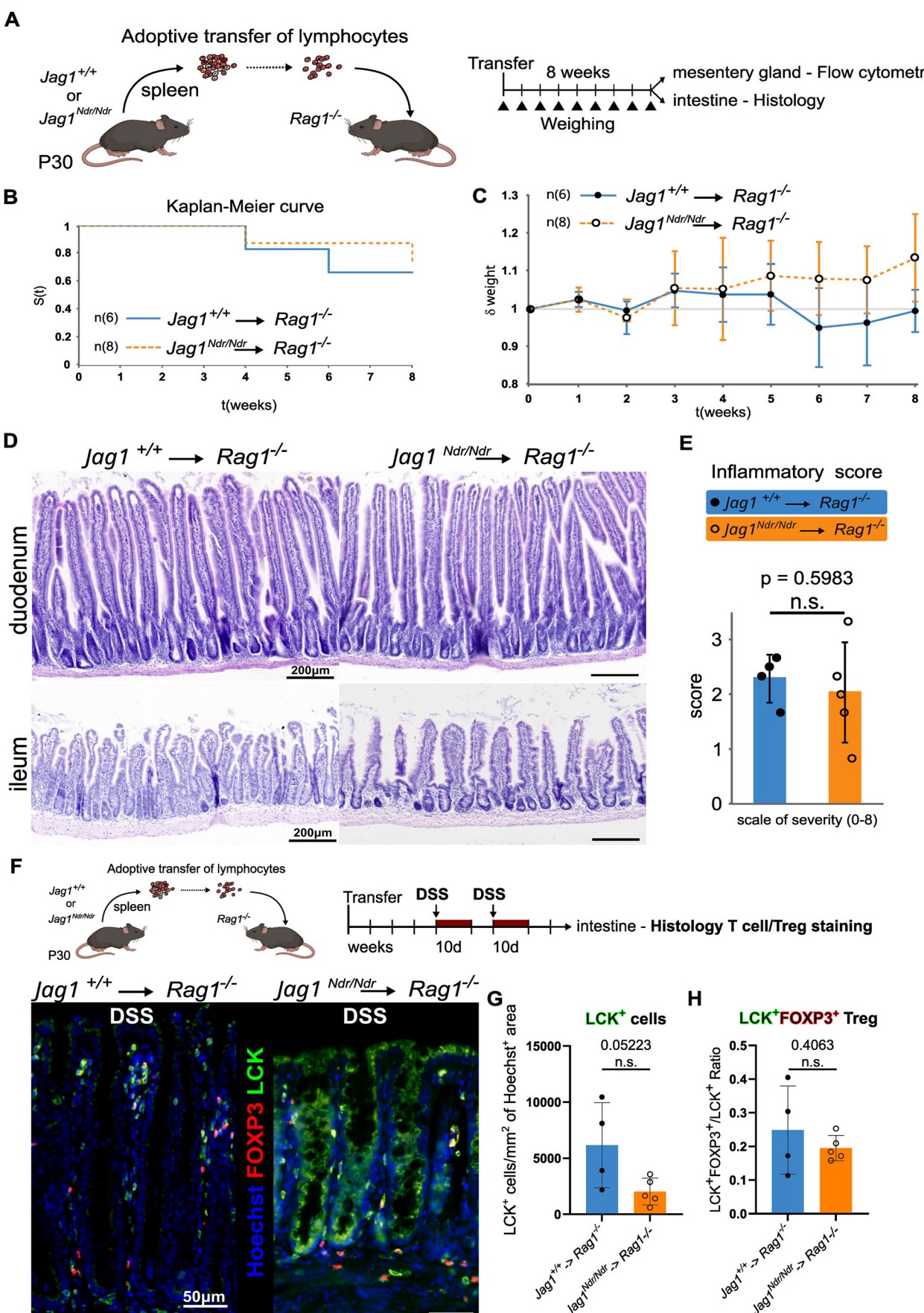

◀ **Figure EV4.** *Jag1^{Ndr/Ndr}* **T cells are not autoimmune as shown by the transfer to** *Rag1^{−/−}* **hosts.**

(A) Schematic of the experiment. (B) Survival curve of the *Rag1^{−/−}* animals over the course of 8 weeks following transfer with *Jag1^{+/+}* ($n = 6$) or *Jag1^{Ndr/Ndr}* ($n = 8$) T cells. (C) Relative quantification of *Jag1^{+/+}→Rag1^{−/−}* and *Jag1^{Ndr/Ndr}→Rag1^{−/−}* mouse weight normalized to its original value on day 0 after T cell transfer over 8 weeks ($1 = 100\%$ of original weight, mean ± SD, $n = 6$-10 mice). (D) Representative H&E staining of intestinal sections (duodenum—top, ileum—bottom) performed 8 weeks after T cell transfer. (E) Comparison of the intestinal inflammatory score calculated using the H&E-stained slides of intestinal sections from the *Jag1^{+/+}→Rag1^{−/−}* ($n = 4$) and *Jag1^{Ndr/Ndr}→Rag1^{−/−}* ($n = 5$) mice. (F–H) Representative immunofluorescent images of sections from *Jag1^{+/+}→Rag1^{−/−}* (left) and *Jag1^{Ndr/Ndr}→Rag1^{−/−}* (right) mice intestine after DSS, stained for LCK and FOXP3, counterstained with Hoechst (F). Quantification of average LCK^+ T cell count/Hoechst area (G), and LCK^+/FOXP3^+ Treg/LCK^+ ratio (H) Means ± SD, Statistical analysis was performed by unpaired, two-tailed Student's t-test.

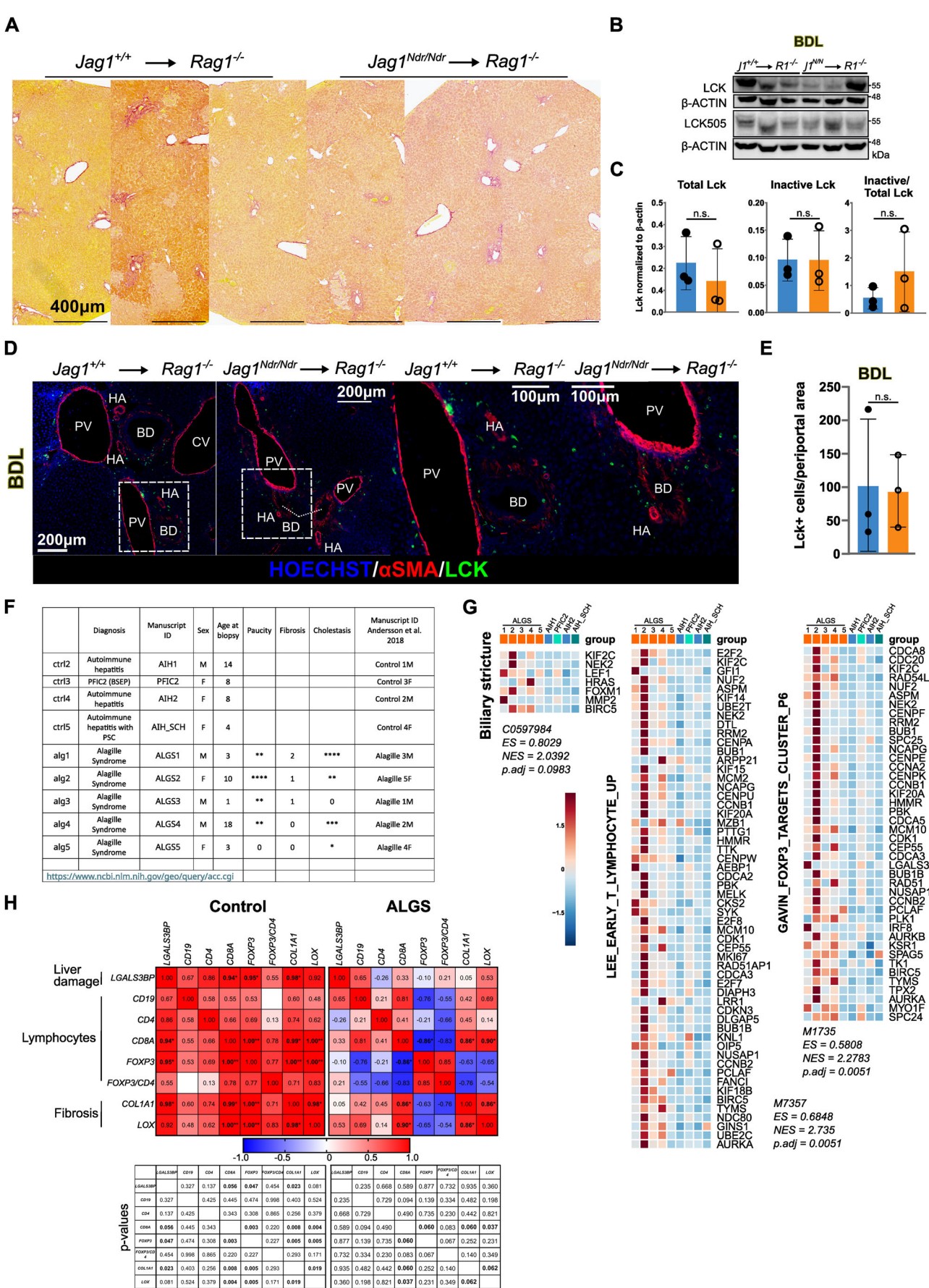

◀ **Figure EV5.** **T cells, present in the periportal area after bile duct ligation of *Jag1^Ndr/Ndr^* transplanted mice, are enriched in patients with ALGS and anti-correlate with fibrosis.**

(A) Representative Sirius red staining in Zone 3 of *Jag1^+/+^→Rag1^−/−^* and *Jag1^Ndr/Ndr^→Rag1^−/−^* mice after BDL. (B, C) Western blot of total LCK, inactive Tyr505 LCK and β-ACTIN levels in LLL lysates from *Jag1^+/+^→Rag1^−/−^* and *Jag1^Ndr/Ndr^→Rag1^−/−^* mice (*n* = 3 each) after BDL (B) and respective quantification (C). Graph represents mean ± SD. (D, E) Representative immunofluorescent images of cryosections from the left lateral lobe of *Jag1^+/+^→Rag1^−/−^* (left) and *Jag1^Ndr/Ndr^→Rag1^−/−^* (right) mice after BDL treatment, stained with antibodies against Lck and SMA (D), and respective quantification of Lck^+ in the periportal area (E), (*n* = 3 each). Graph represents mean ± SD, Statistical analysis was performed by unpaired, two-tailed Student's t-test. (F) List of patient samples from Andersson et al, 2018, re-analyzed in this study. Asterisk indicates severity not significance. (G) Heatmap of gene sets over- and under-represented in ALGS and control patients. Significance was confirmed via false-discovery rate (FDR) statistic with 25% cutoff. (H) Pearson correlation of liver damage marker *LGALS3BP*, markers of lymphocytes *CD19, CD4, CD8A, FOXP3* and markers of fibrosis *COL1A1, LOX* mRNA expression in patients with ALGS and control patients (*p* values *≤0.06; **≤0.005, are indicated below the heatmap). CV central vein, PV portal vein, BD bile duct, HA hepatic artery, AIH autoimmune hepatitis, SCH sclerosing cholangitis, PFIC2 Progressive familial intrahepatic cholestasis type 2, (N)ES (normalized) enrichment score. Fisher Exact test was used for the gene set enrichment analysis; LLL left lateral lobe, CV central vein, PV portal vein, BD bile duct, HA hepatic artery.

