## [Peer Review File · EMBO Molecular Medicine]

Jag1 Insufficiency Alters Liver Fibrosis via T Cell and Hepatocyte Differentiation Defects

Jan Mašek, Iva Filipovic, Noémi van Hul, Lenka Belicova, Marketa Jirouskova, Daniel Oliveira, Anna Frontino, Simona Hankeova, Jingyan He, Fabio Turetti, Afshan Iqbal, Igor Cervenka, Lenka Sarnova, Elisabeth Verboven, Tomáš Brabec, Niklas Björkström, Martin Gregor, Jan Dobes, and Emma Andersson

Corresponding authors: Emma Andersson (Emma.Andersson@ki.se) , Jan Mašek (masekja1@natur.cuni.cz)

Review Timeline:

Transferred from Review Commons:	15th May 24
Editorial Decision:	21st May 24
Revision Received:	15th Jul 24
Editorial Decision:	13th Aug 24
Revision Received:	4th Sep 24
Accepted:	9th Sep 24

Editor: Lise Roth

Transaction Report:

Review
COMMONS

This article was transferred to EMBO Molecular Medicine following peer review at Review Commons.

Review #**1. Evidence, reproducibility and clarity:****Evidence, reproducibility and clarity (Required)**

This is an interesting study that examines defects in the Jag1^{ndr/ndr} mouse model of Alagille syndrome. The novel aspects of this manuscript are the comparisons, at many levels, between the mouse model and ALG patient samples, including an examination of immune profiles. The conclusions that the Jag1^{ndr/ndr} mouse model is an accurate representation of the human ALG syndrome appear valid. However the reported differences in immune profiles, particularly in the Jag1^{ndr/ndr} mouse model are difficult to understand. The data presented indicate a reduction in CD4⁺ cells in the Jag1^{ndr/ndr} mouse at day P3 in both liver and spleen. Additionally, the authors report differences between the the Jag1^{ndr/ndr} mouse and controls at day P30 in the relative percentages of DN, DP and SP CD4 and CD8 cells in the thymus. When examining the peripheral lymphoid system, CD4⁺ numbers are the same in both the Jag1^{ndr/ndr} animals and controls however CD8⁺ numbers are reduced and FoxP3/CD4⁺ cells are increased in both the spleen and the thymus. FoxP3/CD4⁺ T cells are usually assumed to be regulatory T cells that dampen the inflammatory responses of T cells. Therefore, the increase in this population in an animal model of what is assumed to be an inflammatory disease is confusing and confounding. The authors do not present a clear analysis of how they feel an increase of Tregs would lead to this disease. One possibility is that this population is not functioning as conventional Tregs and rather are promoting inflammation but this conclusion would require a functional analysis of this population of cells, at the very least in an in vitro analysis of T cell suppression. From an immunologist's point of view, their data are antithetical to what one would expect to find in an inflammatory disease. Perhaps this reviewer is missing an important point but if I am missing it, then other who read this manuscript also may be confused.

Minor points that should be addressed include:

- The source cells used in the transfer experiments reported in Figure 5 is unclear. Are they using total spleen cells with T, B and myeloid cells or are they using purified T cells. And if it is the latter, have they assessed the ratio of CD4⁺ versus FoxP3/CD4⁺ cells in the transferred cells?
- In the DSS experiments in Figure 5, there does not appear to be a no DSS control. What does the architecture look like without DSS?
- Again, in Figure 5, were FoxP3/CD4⁺ cells enumerated?
- The authors noted that splenomegaly was observed in the Jag1^{ndr/ndr} mouse model. Again this is antithetical to what one would expect when one sees an increase in FoxP3/CD4⁺ T regs.

2. Significance:**Significance (Required)**

The strengths of this paper are the careful comparisons between the mouse model and the human ALG syndrome. These comparisons are valuable and worth publication.

Weaknesses are stated above. Needs a clearer explanation for their immune analysis.

3. How much time do you estimate the authors will need to complete the suggested revisions:

Estimated time to Complete Revisions (Required)

(Decision Recommendation)

Between 1 and 3 months

4. Review Commons values the work of reviewers and encourages them to get credit for their work. Select 'Yes' below to register your reviewing activity at Web of Science Reviewer Recognition Service (formerly Publons); note that the content of your review will not be visible on Web of Science.

Yes

Review #2

1. Evidence, reproducibility and clarity:

Evidence, reproducibility and clarity (Required)

****Summary:****

Masek and colleagues use multi-pronged studies on the Jag1[Ndr/Ndr] mouse model of Alagille syndrome (ALGS) combined with transcriptomic analysis on livers from patients with ALGS to elucidate the potential mechanisms regulating liver fibrosis in this disease. The authors first show that Jag1[Ndr/Ndr] animals develop pericellular and perisinusoidal fibrosis and exhibit evidence for portal hypertension, similar to patients with ALGS. Single-cell RNA-sequencing indicated more hepatoblasts and less hepatocytes, relatively speaking, in Jag1[Ndr/Ndr] P3 livers, which suggested hampering of hepatoblast differentiation to hepatocytes. Deconvolution of previously generated bulk RNA-seq data from Jag1[Ndr/Ndr] P10 livers and GESA on RNAseq data from livers of these mice and patients with ALGS confirmed the P3 scRNA-seq observations and indicated mild pro-inflammatory activation of immature hepatocytes in ALGS livers. GESA also suggested an inability of Jag1[Ndr/Ndr] livers to attract T cells upon cholestatic injury. Indeed, 25-color flow cytometry on liver and spleen from mutant and control mice indicated a defect in T cell response to cholestasis in this model. The authors then examined the effects of the Ndr mutation on T-cell development and function. They found that the Ndr/Ndr thymi were significantly smaller than control thymi. Moreover, Ndr/Ndr thymi showed an increase in CD4+ T-cells and Tregs at the expense of double-positive T-cells. The authors then performed lymphocyte transplantation studies and concluded that Ndr/Ndr T-cells fail to mount an adequate response to inflammation in a DSS model of ulcerative colitis. The authors tested the contribution of Ndr/Ndr immune cells to liver fibrosis in a model of experimentally induced cholestasis (bile duct ligation; BDL). Ndr/Ndr T-cells did not show any defects in migrating into the liver upon BDL. However, the periportal fibrosis observed in BDL model was reduced in animals receiving Ndr/Ndr immune cells compared to those receiving Jag1+/+ immune cells. This was accompanied by significantly less aSMA staining in these livers. Finally, reanalysis of bulk RNAseq data from liver samples from ALGS and other liver diseases suggested that the presence of FOXP3+ T-reg cells in the liver is associated with higher liver fibrosis in non-ALGS liver diseases but lower liver fibrosis in ALGS livers. The authors have used an impressive combination of single-cell RNA-sequencing, reanalysis of previous bulk RNA-sequencing data from their group and others, 25-color FACS analysis, and adoptive immune transfer experiments in this manuscript, and systematically provide quantification and statistical analysis for their

data. Overall, this is an interesting and important study. Prior studies are referenced appropriately. The text and figures are clear and accurate. I don't think any additional experiments are essential. However, the issues listed under Major comments should be discussed and clarified in the manuscript, especially the first item.

****Major comments:****

- Only a small fraction of the cells in scRNA-seq experiments have been assigned to hepatocytes/hepatoblast clusters, with the majority of these cells allocated to Hepato-Ery cluster. This suggests that many hepatocytes and potentially hepatoblasts have been lost during sample preparation. The authors should discuss this issue and its potential implications on the interpretation of the cell ratios and gene expression conclusions of scRNA-seq data.
- The Jag1[Ndr/Ndr] strain is an excellent model for various aspects of ALGS phenotypes. However, when it comes to linking the effects of this mutation to the function of a specific cell type, it is worth considering that Jag1[Ndr/Ndr] might not recapitulate the effects of loss of one copy of JAG1 observed in most patients with ALGS. This is especially important given the sensitivity of various cellular and organ-level processes to the degree of Notch pathway activation. In the context of the present manuscript, it is possible that what the authors have observed in Jag1[Ndr/Ndr] lymphocytes does not mirror how a JAG1-heterozygous human lymphocyte behaves. This is not a major concern, but it is worth considering.
- The basis for the opposite type of correlation between COL1A1 expression and POXP3 level in ALGS versus non-ALGS liver disease is not clear.

****Minor comments:****

- Page 2, last paragraph of Introduction, Page 12 last sentence, and Supplementary Methods: Please use "adoptive immune transfer" instead of "adaptive immune transfer".
- Pages 3 and 4: Reference is made to Figures 3E-O, which appears to be Figure 2E-O.
- Figure 3 legend: "Analysis in (E) is one-way ANOVA with Dunnett's multiple comparison test". Panel E compares two means, so ANOVA is not the appropriate statistical analysis for these data. Is this sentence related to panel D?
- Page 9: Please correct misspelling: "response to intestinal insult (Fig. 5). W therefore".
- The Science Translation Medicine references lack page number.

****Referees cross-commenting****

To my knowledge, ALGS is not considered to be an inflammatory disorder. Furthermore, the splenomegaly observed in the mouse model could be due to portal hypertension rather than a primary immune disturbance. Having said that, I agree with the other reviewers that the manuscript will benefit from further discussion and clarification on the immune-related observations.

2. Significance:

Significance (Required)

Despite severe cholestasis, ALGS patients do not show as much fibrosis as other cholestatic diseases, including biliary atresia (BA). A previous study had suggested that this phenomenon could be due to the difference in the nature of reactive hepatobiliary cells in ALGS compared to BA (Fabris et al, 2007). Moreover, a number of studies have suggested a role for Notch pathway activation in several cell types in the liver in the development of liver fibrosis (for example, Sawitza et al, Hepatology, 2009; Chen et al, Plos One, 2012; Duan et al, Hepatology, 2018; Yu et al, Science Translational Medicine, 2021). However, although a role for Notch signaling in T-cells is well established, it was not known whether impaired T-cell development/function contributes to reduced fibrosis in ALGS liver disease. Accordingly, the current manuscript provides novel insight into the mechanism of fibrosis in this disease. Moreover, the observation that Jag1-mutant T-cells do not confer as much protection as control T-cells to immunodeficient mice subjected to DSS-induced ulcerative colitis provides strong evidence for impaired T-cell immunity in this ALGS model and might help explain other aspects of ALGS phenotypes.

The manuscript will be of interest to broad audience (Notch signaling, cholestatic liver disease, mechanisms of liver fibrosis, T-cell development).

I have expertise in Notch signaling and in using animal models of human developmental disorders.

3. How much time do you estimate the authors will need to complete the suggested revisions:

**Estimated time to Complete Revisions (Required)
(Decision Recommendation)**

Less than 1 month

4. Review Commons values the work of reviewers and encourages them to get credit for their work. Select 'Yes' below to register your reviewing activity at Web of Science Reviewer Recognition Service (formerly Publons); note that the content of your review will not be visible on Web of Science.

Yes

Review #3 -

1. Evidence, reproducibility and clarity:

Evidence, reproducibility and clarity (Required)

The article entitled "Jag1 Insufficiency Disrupts Neonatal T Cell Differentiation and Impairs Hepatocyte Maturation, Leading to Altered Liver Fibrosis" by Mašek et al described the role of Notch ligand JAGGED1 (JAG1) in the T-cell differentiation contributing to liver fibrosis and immune system development in ALGS. This article is well written and has important preliminary findings that could establish Jag1 and its downstream signaling pathways as potential therapeutic targets to attenuate liver fibrosis.

1. ****Minor comments:**** In page 4, they mentioned that "the hepatoblast marker alpha fetoprotein (AFP) was 3.1-fold enriched (Fig. 3J,K), while the mature hepatocyte marker CYP1A2 protein was 1.7-fold less expressed (Fig. 3L-M)", the figure numbers should be

changed to 2J, K, L-M etc.

2. In liver fibrosis the Th17 cells play crucial roles. Please show the level of IL17A mRNA level in the liver in the Jag1Ndr/Ndr mice compared to the Jag1+/+ mice.

3. Also, please show the expression level of pro-inflammatory molecules, for example, TNF α , IL1 β , MCP1 etc and the level of MMPs (especially MMP2, MMP8, MMP9) in the livers of the mice models used.

4. Authors have shown significant alterations in the Treg population in their Jag1Ndr/Ndr mice of ALGS. Please also show the expression of IL10 and TGF β in the liver and whether they are correlated with the level of Treg populations.

5. It would be interesting to know whether the IFN γ mRNA expression in the livers were altered in the Jag1Ndr/Ndr mice with altered populations of CD8 T cells.

2. Significance:

Significance (Required)

Strength: This article is well written and has important preliminary findings that could establish Jag1 and its downstream signaling pathways as potential therapeutic targets to attenuate liver fibrosis.

Limitations: This study lacked the detailed molecular pathways which could explain how the Jag1 altered the T-cell recruitment, development and hepatocyte maturation in the development of liver fibrosis in the ALGS model.

3. How much time do you estimate the authors will need to complete the suggested revisions:

Estimated time to Complete Revisions (Required)

(Decision Recommendation)

Less than 1 month

4. Review Commons values the work of reviewers and encourages them to get credit for their work. Select 'Yes' below to register your reviewing activity at Web of Science Reviewer Recognition Service (formerly Publons); note that the content of your review will not be visible on Web of Science.

Yes

Revision Plan

Manuscript number: RC-2024-02445

Corresponding author(s): Jan Masek, Emma R. Andersson

1. General Statements

We thank the reviewers, for their careful reading of our manuscript, for pointing out its strengths (including relevance to, and analysis of, human patients), and for providing constructive feedback to improve the clarity of the manuscript.

2. Description of the planned revisions

Reviewer #1 (Evidence, reproducibility and clarity (Required)):

- Again, in Figure 5, were FoxP3/CD4+ cells enumerated?

Author Response: *Fig 5 showed that the inflammatory score, and activation of CD4 and CD8 cells, were lower in the intestine of DSS-treated mice transplanted with Jag1^{Ndr/Ndr} lymphocytes than in those transplanted with Jag1^{+/+} lymphocytes. However, in Figure 5 we had not quantified the number of FoxP3/CD4+ cells (Tregs). We agree that it would be interesting to know whether the dampened intestinal inflammation (in response to a classical inflammatory disease model (DSS-treatment)) is also mediated by excess Tregs. We will therefore now quantify Foxp3+ cells on the intestinal sections of experimental animals used for acquisition of data in Fig 5.*

3. Description of the revisions that have already been incorporated in the transferred manuscript.

Reviewer #1 (Evidence, reproducibility and clarity (Required)):

Reviewer 1 comment: This is an interesting study that examines defects in the Jag1^{ndr/ndr} mouse model of Alagille syndrome. The novel aspects of this manuscript are the comparisons, at many levels, between the mouse model and ALG patient samples, including an examination of immune profiles. The conclusions that the Jag1^{ndr/ndr} mouse model is an accurate representation of the human ALG syndrome appear valid. However the reported differences in immune profiles, particularly in the Jag1^{ndr/ndr} mouse model are difficult to understand. The data presented indicate a reduction in CD4+ cells in the Jag1^{ndr/ndr} mouse at day P3 in both liver and spleen. Additionally, the authors report differences between the the Jag1^{ndr/ndr} mouse and controls at day P30 in the relative percentages of DN, DP and SP CD4 and CD8 cells in the thymus. When examining the peripheral lymphoid system, CD4+ numbers are the same in both the Jag1^{ndr/ndr} animals and controls however CD8+ numbers are reduced and FoxP3/CD4+ cells are increased in both the spleen and the thymus. FoxP3/CD4+ T cells are usually assumed to be regulatory T

Revision Plan

cells that dampen the inflammatory responses of T cells. Therefore, the increase in this population in an animal model of what is assumed to be an inflammatory disease is confusing and confounding. The authors do not present a clear analysis of how they feel an increase of Tregs would lead to this disease. One possibility is that this population is not functioning as conventional Tregs and rather are promoting inflammation but this conclusion would require a functional analysis of this population of cells, at the very least in an in vitro analysis of T cell suppression. From an immunologist's point of view, their data are antithetical to what one would expect to find in an inflammatory disease. Perhaps this reviewer is missing an important point but if I am missing it, then other who read this manuscript also may be confused.

Author Response: *We thank the reviewer for carefully assessing our work, and for noting which aspects of the immune analyses should be more thoroughly explained. We apologize for any confusion, which a clearer introduction will help to avoid.*

Alagille syndrome is not thought of as an inflammatory disorder, it is a congenital disorder affecting bile duct development (Kohut et al 2021, Semin Liver Dis). During normal bile duct development, JAG1+ portal fibroblasts signal to NOTCH2+ hepatoblasts to instruct bile duct development. In the context of low JAG1 signaling, hepatoblasts either fail to adopt a cholangiocyte fate, or fail to undergo bile duct morphogenesis, resulting in bile duct paucity and cholestasis. This cholestasis should activate inflammatory processes leading to fibrosis, which is the subject of this study.

We agree with the reviewer that Tregs would be expected to suppress inflammation, and our data are consistent with Treg suppression of inflammation. We show, for the first time, that Tregs are enriched in Jag1^{Ndr/Ndr} mice (Fig 4) and present evidence that they suppress inflammation (Fig 5) and fibrosis (Fig 6), which could explain the atypical fibrosis seen in patients with ALGS.

To clarify that ALGS is a genetic liver disease affecting bile duct formation, we:

- 1. Modified and extended the following text in the Introduction (Page 2, lines 14-17):
“ALGS is mainly caused by mutations in the Notch ligand JAGGED1 (JAG1, 94%) (Mašek & Andersson, 2017; Oda et al, 1997), **affecting bile duct development and morphogenesis**, resulting in bile duct paucity and cholestasis. Immune dysregulation has also been described (Tilib Shamoun et al, 2015), **but how this might interact with liver disease in ALGS to affect fibrosis is not known.**”*
- 2. Introduce the disease, the animal model, and the scientific question in a schematic in new Fig 1A.*

Reviewer 1 comment: Minor points that should be addressed include:
• The source cells used in the transfer experiments reported in Figure 5 is unclear. Are they using total spleen cells with T, B and myeloid cells or are they using purified T cells. And if it

Revision Plan

is the latter, have they assessed the ratio of CD4+ versus FoxP3/CD4+ cells in the transferred cells?

Author Response: *Total spleen cells including all lymphocytes were transplanted, as described in Materials and Methods. The constituent T-cell populations are characterized and shown in Fig 4F. To clarify this, we:*

1. *added the text “Adoptive transfer of lymphocytes” to the schematic in Fig 5A, FigS5A, and Fig 6A, and*
2. *modified the opening paragraph related to results presented in Fig.5 and FigS5 in the following way (page 8, line 209): “To investigate Jag1^{Ndr/Ndr} T cell function, we performed adoptive transfer of the splenic lymphocytes into Rag1^{-/-} mice, which lack mature B- and T cell populations, but provide a host environment with normal Jag1 (Mombaerts et al, 1992).”*

To acknowledge that B-cells and innate lymphoid cells might contribute to the observed results, we include a following sentence in the Discussion:

(page 12, lines 369-371) “Finally, our experimental setup does not exclude an additional contribution of other lymphocytes (B-cells or innate lymphoid cells) to the BDL-induced fibrosis, and selective testing of the individual subpopulations would be an intriguing follow up to this study.”

Reviewer 1 comment: In the DSS experiments in Figure 5, there does not appear to be a no DSS control. What does the architecture look like without DSS?

Author Response: *The intestinal architecture and phenotype of mice transplanted with Jag1^{+/+} or Jag1^{Ndr/Ndr} lymphocytes, not treated with DSS, are presented in Supplementary Figure 5. In the absence of DSS, Jag1^{+/+} or Jag1^{Ndr/Ndr} -transplanted mice exhibit no overt differences in survival or weight gain/loss. The intestinal inflammatory score was not different in the two conditions and was 2.29 +/-0.44 and 2.03 +/-0.92 for Jag1^{+/+} or Jag1^{Ndr/Ndr} -transplanted mice, respectively.*

To compare the results with and without DSS, we added the following text to the results section, when describing the DSS results (Page 9, lines 223-226):

“As expected, histological scoring of intestinal and colonic inflammation revealed elevated inflammation in Jag1^{+/+}→Rag1^{-/-} mice treated with DSS (Fig. 5C,D) compared to Jag1^{+/+}→Rag1^{-/-} mice not treated with DSS (Fig. S5). However, there was significantly less inflammation in Jag1^{Ndr/Ndr}→Rag1^{-/-} mice than in Jag1^{+/+}→Rag1^{-/-} mice (Fig. 5C,D).”

Reviewer 1 comment: The authors noted that splenomegaly was observed in the Jag1^{ndr/ndr} mouse model. Again this is antithetical to what one would expect when one sees an increase in FoxP3/CD4+ T regs.

Revision Plan

Author Response: We thank the reviewer for pointing at a possible discrepancy, related to Fig1 in which we report the presence of splenomegaly. Although there can be multiple causes of splenomegaly, it is one of the hallmarks of portal hypertension (as also corroborated by Reviewer 2), tightly connected with liver fibrosis, present in patients with ALGS and we report it as such in the manuscript. To clarify this, we added the following text sections:

1. Results (page 2, lines 37,38) “Liver fibrosis compresses blood vessels and reduces their blood flow, leading to portal hypertension, a serious consequence of liver disease which can manifest as splenomegaly.”
2. Discussion (page 13, line 394-401): “Splenomegaly has been described as a consequence of portal hypertension in ALGS (Kamath et al, 2020), but could also be attributed to immune-related pathology. *Jag1^{Ndr/Ndr}* mice exhibit splenomegaly as early as P10, and is exacerbated at P30 (Fig. 1E,F). Patients with other liver diseases display portal hypertension and cirrhosis, with both splenomegaly and hypersplenism associated with a high CD4+/CD8+ ratio, but a low Treg+/CD4+ ratio (Nomura et al, 2014). However, *Jag1^{Ndr/Ndr}* mice present with splenomegaly but not hypersplenism. An overactive spleen (hypersplenism) would remove red blood cells which are instead enriched in *Jag1^{Ndr/Ndr}* mice, and Tregs were enriched in *Jag1^{Ndr/Ndr}* mice, not depleted as seen in cirrhosis/hypersplenism. These data are thus consistent with portal hypertension-induced splenomegaly rather than hypersplenism.”

Reviewer #1 (Significance (Required)):

Reviewer 1 comment: The strengths of this paper are the careful comparisons between the mouse model and the human ALG syndrome. These comparisons are valuable and worth publication.

Author Response: We thank the reviewer for these comments.

Reviewer 1 comment: Weaknesses are stated above. Needs a clearer explanation for their immune analysis.

Author Response: We thank the reviewers for highlighting points requiring clarification and hope the proposed text changes and additional data presented in response to the comments of all three reviewers lead to a significant clarification of the immunological aspect of our study.

Reviewer #2 (Evidence, reproducibility and clarity (Required)):

Reviewer 2 comment:

Revision Plan

Summary:

Masek and colleagues use multi-pronged studies on the Jag1[Ndr/Ndr] mouse model of Alagille syndrome (ALGS) combined with transcriptomic analysis on livers from patients with ALGS to elucidate the potential mechanisms regulating liver fibrosis in this disease. The authors first show that Jag1[Ndr/Ndr] animals develop pericellular and perisinusoidal fibrosis and exhibit evidence for portal hypertension, similar to patients with ALGS. Single-cell RNA-sequencing indicated more hepatoblasts and less hepatocytes, relatively speaking, in Jag1[Ndr/Ndr] P3 livers, which suggested hampering of hepatoblast differentiation to hepatocytes. Deconvolution of previously generated bulk RNA-seq data from Jag1[Ndr/Ndr] P10 livers and GESA on RNAseq data from livers of these mice and patients with ALGS confirmed the P3 scRNA-seq observations and indicated mild pro-inflammatory activation of immature hepatocytes in ALGS livers. GESA also suggested an inability of Jag1[Ndr/Ndr] livers to attract T cells upon cholestatic injury. Indeed, 25-color flow cytometry on liver and spleen from mutant and control mice indicated a defect in T cell response to cholestasis in this model. The authors then examined the effects of the Ndr mutation on T-cell development and function. They found that the Ndr/Ndr thymi were significantly smaller than control thymi. Moreover, Ndr/Ndr thymi showed an increase in CD4+ T-cells and Tregs at the expense of double-positive T-cells. The authors then performed lymphocyte transplantation studies and concluded that Ndr/Ndr T-cells fail to mount an adequate response to inflammation in a DSS model of ulcerative colitis. The authors tested the contribution of Ndr/Ndr immune cells to liver fibrosis in a model of experimentally induced cholestasis (bile duct ligation; BDL). Ndr/Ndr T-cells did not show any defects in migrating into the liver upon BDL. However, the periportal fibrosis observed in BDL model was reduced in animals receiving Ndr/Ndr immune cells compared to those receiving Jag1+/+ immune cells. This was accompanied by significantly less aSMA staining in these livers. Finally, reanalysis of bulk RNAseq data from liver samples from ALGS and other liver diseases suggested that the presence of FOXP3+ T-reg cells in the liver is associated with higher liver fibrosis in non-ALGS liver diseases but lower liver fibrosis in ALGS livers. The authors have used an impressive combination of single-cell RNA-sequencing, reanalysis of previous bulk RNA-sequencing data from their group and others, 25-color FACS analysis, and adoptive immune transfer experiments in this manuscript, and systematically provide quantification and statistical analysis for their data. Overall, this is an interesting and important study. Prior studies are referenced appropriately. The text and figures are clear and accurate. I don't think any additional experiments are essential. However, the issues listed under Major comments should be discussed and clarified in the manuscript, especially the first item.

Author Response: *We sincerely thank the reviewer for the comprehensive and insightful assessment of our manuscript. We are particularly gratified to note your acknowledgment of the thoroughness of our experimental approach and the clarity of our presentation. We are pleased that no further experiments would be required, and will address the points raised under Major comments which enhance our study's quality and accessibility.*

Revision Plan

Reviewer 2 comment:

Major comments:

- Only a small fraction of the cells in scRNA-seq experiments have been assigned to hepatocytes/hepatoblast clusters, with the majority of these cells allocated to Hepato-Ery cluster. This suggests that many hepatocytes and potentially hepatoblasts have been lost during sample preparation. The authors should discuss this issue and its potential implications on the interpretation of the cell ratios and gene expression conclusions of scRNA-seq data.

Author Response: *We agree with the reviewer regarding this aspect of our study. We mentioned this limitation in the supplementary methods section: "Liver parenchymal cells constituted ~6.5% of cells at E16.5, and ~7.5% of cells at P3 and included mesenchymal cells, endothelial cells, hepatoblasts and hepatocytes (Fig. S1D), this parenchymal proportion is lower than in vivo, but consistent with ex vivo liver digest (Guilliams et al, 2022)." We recognize it may be too inaccessible there, and we thus added the following text to the Discussion section of the manuscript: (Pages 11-12, lines 330-337) "A limitation of this study is the underrepresentation of the hepatoblast/cyte parenchymal cells in the scRNA-seq dataset (Fig. 2A-D), which constituted ~6.5% of analyzed cells at E16.5, and ~7.5% of cells at P3 (Fig. S1D). This parenchymal proportion is lower than in vivo, but is consistent with scRNA seq datasets obtained with ex vivo liver digest (Guilliams et al, 2022). One risk is that cell stress as a result of dissociation could result in further loss of injured Jag1^{Ndr/Ndr} hepatocytes, impacting the interpretation of cell type abundance. Nuclear scRNAseq can overcome cell type-dependent dissociation sensitivity bias (Guilliams et al, 2022), and could provide further insights into Jag1^{Ndr/Ndr} livers at the single cell level. Nonetheless, both bulk RNA seq deconvolution and histological analyses confirmed that patients and Jag1^{Ndr/Ndr} mice exhibit hepatoblast enrichment and less differentiated hepatocytes."*

Reviewer 2 comment: The Jag1[Ndr/Ndr] strain is an excellent model for various aspects of ALGS phenotypes. However, when it comes to linking the effects of this mutation to the function of a specific cell type, it is worth considering that Jag1[Ndr/Ndr] might not recapitulate the effects of loss of one copy of JAG1 observed in most patients with ALGS. This is especially important given the sensitivity of various cellular and organ-level processes to the degree of Notch pathway activation. In the context of the present manuscript, it is possible that what the authors have observed in Jag1[Ndr/Ndr] lymphocytes does not mirror how a JAG1-heterozygous human lymphocyte behaves. This is not a major concern, but it is worth considering.

Revision Plan

Author Response: *We agree and thus added the following discussion paragraph (page 11, lines 315-321) “In patients with ALGS, who have a single mutation in either JAG1 or NOTCH2, the remnant healthy allele(s) could be expected to mediate signaling. However, some JAG1 mutations exhibit dominant negative effects (Ponio et al, 2007; Xiao et al, 2013; Guan et al, 2023), which could entail further repression of JAG1/NOTCH2 signaling. In this context, it is important to note that the Jag1^{Ndr/Ndr} mice are homozygous for the missense mutation, but retain some JAG1 activity, and it is not clear to which degree this mimics JAG1 heterozygosity in humans. It would be of interest to test whether Jag1 potency affects hepatoblast differentiation or injury-induced reversion of hepatocytes in patients as a function of their genotype.”*

Reviewer 2 comment: •The basis for the opposite type of correlation between COL1A1 expression and POXP3 level in ALGS versus non-ALGS liver disease is not clear.

Author Response: *We thank the reviewer for pointing out the unclear interpretation of the patient data. In patients with ALGS, the extent of fibrosis is likely to be highly multifactorial, involving (as we show) hepatocyte immaturity, dampened inflammation, and immune system dysregulation (possibly involving more than T-cells). Since human patients ARE so heterogeneous, teasing apart the relative contribution of each is currently outside the scope of our study, but will be an important area of future research. Nonetheless we thought it was important and interesting to show these patterns in supplementary Fig 6, now extended with further data, and analyses, and described in the following manner:*

Results section: (page 10, lines 267-275) “Liver damage in non-ALGS liver disease (using liver injury marker LGALS3BP) (Yang et al, 2021), was positively correlated with recruitment of lymphocytes (including CD8A+, and FOXP3+ populations of T cells), as well as the extent of fibrosis (COL1A1 abundance) (Fig. S6G). However, in ALGS, the extent of liver damage, lymphocyte recruitment and fibrosis were unlinked (Fig. S6G). These data are in line with the observation that liver stiffness (a proxy for fibrosis) in ALGS is independent of biomarkers of liver disease (Leung et al, 2023). While Treg infiltration in ALGS was independent of liver damage, it exhibited a tendency towards a negative correlation with fibrosis (Fig. S6G), corroborating that elevated levels of Tregs may limit fibrosis in ALGS. Altogether, these data suggest that the liver and lymphocytes may be differentially affected in different patients with ALGS, a disorder that is well known for its heterogenous presentation.”

Minor comments:

- Page 2, last paragraph of Introduction, Page 12 last sentence, and Supplementary Methods: Please use "adoptive immune transfer" instead of "adaptive immune transfer".
- Pages 3 and 4: Reference is made to Figures 3E-O, which appears to be Figure 2E-O.
- Figure 3 legend: "Analysis in (E) is one-way ANOVA with Dunnett's multiple comparison test". Panel E compares two means, so ANOVA is not the appropriate statistical analysis for these data. Is this sentence related to panel D?

Revision Plan

- Page 9: Please correct misspelling: "response to intestinal insult (Fig. 5). W therefore".
- The Science Translation Medicine references lack page number.

Author Response: *We thank the reviewer deeply for taking the time to meticulously note and convey these errors, helping us to correct these. The suggested corrections have been implemented. Science Transl Med is an online journal and does not have page numbers – we have added an issue number to facilitate retrieval of these references.*

Additionally, we noticed that the image of a consecutive liver section with CYP1A2 staining from Jag1^{Ndr/Ndr} liver in Fig 2 L was accidentally flipped along the horizontal axis, which we have now corrected. We also changed the scRNAseq cell cluster naming from Hepatoblasts/cytes, Hepato_Ery, and Kupffer cells, Kuffer cells_Ery to Hepatoblasts/cytes I, and II, and Kupffer cells I and II, respectively, to match the Neutrophil progenitors I and II naming convention. Names were subsequently also changed in Fig S1 and methods.

****Referees cross-commenting****

To my knowledge, ALGS is not considered to be an inflammatory disorder. Furthermore, the splenomagaly observed in the mouse model could be due to portal hypertension rather than a primary immune disturbance. Having said that, I agree with the other reviewers that the manuscript will benefit from further discussion and clarification on the immune-related observations.

Author Response: *We thank Reviewer 2 for indicating to Reviewer 1 that ALGS is not considered an inflammatory disorder, which we agree with. It was not our intention to convey this idea. To avoid confusion, we now:*

1. *Added a schematic in Fig 1A.*
2. *Modified and extended the following text in the Introduction: (Page 2, lines 14-17): "ALGS is mainly caused by mutations in the Notch ligand JAGGED1 (JAG1, 94%) (Mašek & Andersson, 2017; Oda et al, 1997), affecting bile duct development and morphogenesis, resulting in bile duct paucity and cholestasis. Immune dysregulation has also been described (Tilib Shamoun et al, 2015), but how this might interact with liver disease in ALGS to affect fibrosis is not known."*

Furthermore, we have addressed or will address all comments from reviewer 1 to clarify the immune-related observations.

Reviewer #2 (Significance (Required)):

Despite severe cholestasis, ALGS patients do not show as much fibrosis as other cholestatic diseases, including biliary atresia (BA). A previous study had suggested that this phenomenon could be due to the difference in the nature of reactive hepatobiliary cells in ALGS compared to BA (Fabris et al, 2007). Moreover, a number of studies have suggested a role for Notch pathway activation in several cell types in the liver in the development of liver

Revision Plan

fibrosis (for example, Sawitza et al, Hepatology, 2009; Chen et al, Plos One, 2012; Duan et al, Hepatology, 2018; Yu et al, Science Translational Medicine, 2021). However, although a role for Notch signaling in T-cells is well established, it was not known whether impaired T-cell development/function contributes to reduced fibrosis in ALGS liver disease. Accordingly, the current manuscript provides novel insight into the mechanism of fibrosis in this disease. Moreover, the observation that Jag1-mutant T-cells do not confer as much protection as control T-cells to immunodeficient mice subjected to DSS-induced ulcerative colitis provides strong evidence for impaired T-cell immunity in this ALGS model and might help explain other aspects of ALGS phenotypes.

The manuscript will be of interest to broad audience (Notch signaling, cholestatic liver disease, mechanisms of liver fibrosis, T-cell development).

I have expertise in Notch signaling and in using animal models of human developmental disorders.

Author Response: We thank the reviewer for the balanced assessment of our manuscript in light of the current knowledge, and for highlighting its importance in the context of not only Notch and ALGS, but also other cholestatic and fibrotic liver diseases.

Reviewer #3 (Evidence, reproducibility and clarity (Required)):

The article entitled "Jag1 Insufficiency Disrupts Neonatal T Cell Differentiation and Impairs Hepatocyte Maturation, Leading to Altered Liver Fibrosis" by Mašek et al described the role of Notch ligand JAGGED1 (JAG1) in the T-cell differentiation contributing to liver fibrosis and immune system development in ALGS. This article is well written and has important preliminary findings that could establish Jag1 and its downstream signaling pathways as potential therapeutic targets to attenuate liver fibrosis.

Author Response: *We thank the reviewer for recognizing our work and pointing out the therapeutical implications of our findings.*

Reviewer 3 comment 1: Minor comments: In page 4, they mentioned that "the hepatoblast marker alpha fetoprotein (AFP) was 3.1-fold enriched (Fig. 3J,K), while the mature hepatocyte marker CYP1A2 protein was 1.7-fold less expressed (Fig. 3L-M)", the figure numbers should be changed to 2J, K, L-M etc.

Author Response: *We thank the reviewer for identifying these errors. The suggested corrections have been implemented.*

Reviewer 3 comment 2: In liver fibrosis the Th17 cells play crucial roles. Please show the level of IL17A mRNA level in the liver in the Jag1Ndr/Ndr mice compared to the Jag1+/+ mice.

Revision Plan

Author Response: We thank the reviewer for the insightful comments. We indeed investigated the Th17 vs Treg immune response, however we detect neither Th17-expressed Il17, Il17a, Il17f, nor Il21 and Il22 mRNA in the bulk RNA data, suggesting their expression is either masked or they are not present in significant numbers within the liver tissue at P10, preventing us from drawing any conclusions about this cell population.

Reviewer 3 comment 3: Also, please show the expression level of pro-inflammatory molecules, for example, TNF α , IL1 β , MCP1 etc and the level of MMPs (especially MMP2, MMP8, MMP9) in the livers of the mice models used.

Author Response: The expression of Il10, Il1b, Mcp1(Ccl2), was presented in the manuscript Fig. 2O, and we attach here a full list together with the expression levels of Mmp2/8/9, Tnfa, Ifng, Il17 receptor family and Tgfb1-3. Out of these, Mmp8 (0.9 Log2fold change = 1.9-fold), Ccl2 (2.2 Log2fold change = 4.7-fold), and Il17rb (1.1 Log2fold change = 2.1-fold) were significantly upregulated, but do not indicate any specific leukocyte population's response. This is in line with data in Fig S2E, demonstrating a dominance of myeloid over adaptive immune response in the GSEA of the immune KEGGs.

Since lymphocytes are underrepresented in the bulk transcriptomics, and individual genes might report activity of many different cell types, we chose to focus on the list of genes shown to be markers of activated hepatocytes, to avoid over interpretation of the RNA sequencing data. Instead, the immune analyses were based on flow cytometry data, which we expect should accurately report cell type abundance across organ systems.

Reviewer 3 comment 4. Authors have shown significant alterations in the Treg population in their Jag1Ndr/Ndr mice of ALGS. Please also show the expression of IL10 and TGF β in the liver and whether they are correlated with the level of Treg populations.

Author response: IL10 and Tgfb mRNA levels in liver are shown in the heatmap above, and were not significantly different between genotypes at P10. They were also not correlated with

Revision Plan

Foxp3 levels, as shown in the correlation matrices below (Pearson's *R* values in top row, significance values in bottom row).

Cntrl

Pearsons R

p value

	Il10	Tgfb1	Tgfb2	Tgfb3	Foxp3
Il10		0.2547	0.1881	0.5027	0.9975
Tgfb1	0.2547		0.0666	0.248	0.7477
Tgfb2	0.1881	0.0666		0.3146	0.8144
Tgfb3	0.5027	0.248	0.3146		0.4997
Foxp3	0.9975	0.7477	0.8144	0.4997	

Ndr

	Il10	Tgfb1	Tgfb2	Tgfb3	Foxp3
Il10		0.6952	0.2723	0.7667	0.3969
Tgfb1	0.6952		0.423	0.0715	0.9078
Tgfb2	0.2723	0.423		0.4945	0.6692
Tgfb3	0.7667	0.0715	0.4945		0.8363
Foxp3	0.3969	0.9078	0.6692	0.8363	

Reviewer 3 comment 5. It would be interesting to know whether the IFN γ mRNA expression in the livers were altered in the Jag1Ndr/Ndr mice with altered populations of CD8 T cells.

Author Response: *There was no significant difference in IFN γ mRNA expression levels between Jag1^{+/+} and Jag1^{Ndr/Ndr} livers at P10 (please see the heatmap in response to comment no.3, above).*

Reviewer #3 (Significance (Required)): Strength: This article is well written and has important preliminary findings that could establish Jag1 and its downstream signaling pathways as potential therapeutic targets to attenuate liver fibrosis.

Author Response: *Thank you for these comments and pointing out the wider implications of our findings.*

Revision Plan

Reviewer 3 Limitations: This study lacked the detailed molecular pathways which could explain how the Jag1 altered the T-cell recruitment, development and hepatocyte maturation in the development of liver fibrosis in the ALGS model.

Author Response: *We agree that this study does not focus on molecular pathways. The intention of this study was to identify which cell populations contribute to atypical neonatal fibrosis in ALGS. Because we expected this process to be multifactorial, Jag1^{Ndr/Ndr} mice, carrying a systemic mutation, present both advantages (Jag1 abrogation in all cells  ALGS-like organ interactions) and limitations (inability to identify contributions of individual cell types). However, by identifying maturing hepatocytes and Tregs as dysregulated, and demonstrating that Jag1^{Ndr/Ndr} lymphocytes behave abnormally and suppress inflammation and fibrosis in Rag1^{-/-} mice (with normal Jag1 expression), we establish a biological framework that can now be further investigated with conditional genetic tools and in vitro systems, to elucidate specific molecular pathways, that were beyond the scope of the current study.*

4 Description of analyses that authors prefer not to carry out

Not applicable

21st May 2024

Dear Dr. Andersson,

Thank you for the submission of your manuscript to our editorial offices. I have now had the opportunity to read it, together with the referees' reports and your rebuttal letter, and to discuss them with the other members of our editorial team.

We agree that the study fits the scope of the journal, and we appreciate that you have addressed most of the referees' concerns, and are willing to address the rest. We thus encourage you to submit a revised version of your manuscript, including the modifications and revisions described in your point-by-point letter.

Acceptance of the manuscript will entail a second round of review. EMBO Molecular Medicine encourages a single round of revision only and therefore, acceptance or rejection of the manuscript will depend on the completeness of your responses included in the next, final version of the manuscript. For this reason, and to save you from any frustrations in the end, I would strongly advise against returning an incomplete revision.

When submitting your revised manuscript, please carefully review the instructions that follow below. Failure to include requested items will delay the evaluation of your revision:

We require:

- 1) A .docx formatted version of the manuscript text (including legends for main figures, EV figures and tables). Please make sure that the changes are highlighted to be clearly visible.
- 2) Individual production quality figure files as .eps, .tif, .jpg (one file per figure). For guidance, download the 'Figure Guide PDF' (<https://www.embopress.org/page/journal/17574684/authorguide#figureformat>).
- 3) At EMBO Press we ask authors to provide source data for the main figures. Our source data coordinator will contact you to discuss which figure panels we would need source data for and will also provide you with helpful tips on how to upload and organize the files.
- 4) A .docx formatted letter INCLUDING the reviewers' reports and your detailed point-by-point responses to their comments. As part of the EMBO Press transparent editorial process, the point-by-point response is part of the Review Process File (RPF), which will be published alongside your paper.
- 5) A complete author checklist, which you can download from our author guidelines (<https://www.embopress.org/page/journal/17574684/authorguide#submissionofrevisions>). Please insert information in the checklist that is also reflected in the manuscript. The completed author checklist will also be part of the RPF.
- 6) It is mandatory to include a 'Data Availability' section after the Materials and Methods. Before submitting your revision, primary datasets produced in this study need to be deposited in an appropriate public database, and the accession numbers and database listed under 'Data Availability'. Please remember to provide a reviewer password if the datasets are not yet public (see <https://www.embopress.org/page/journal/17574684/authorguide#dataavailability>). In case you have no data that requires deposition in a public database, please state so in this section. Note that the Data Availability Section is restricted to new primary data that are part of this study.
- 7) For data quantification: please specify the name of the statistical test used to generate error bars and P values, the number (n) of independent experiments (specify technical or biological replicates) underlying each data point and the test used to calculate p-values in each figure legend. The figure legends should contain a basic description of n, P and the test applied. Graphs must include a description of the bars and the error bars (s.d., s.e.m.). Please provide exact p values.
- 8) Our journal encourages inclusion of *data citations in the reference list* to directly cite datasets that were re-used and obtained from public databases. Data citations in the article text are distinct from normal bibliographical citations and should directly link to the database records from which the data can be accessed. In the main text, data citations are formatted as follows: "Data ref: Smith et al, 2001" or "Data ref: NCBI Sequence Read Archive PRJNA342805, 2017". In the Reference list, data citations must be labeled with "[DATASET]". A data reference must provide the database name, accession

number/identifiers and a resolvable link to the landing page from which the data can be accessed at the end of the reference. Further instructions are available at .

9) We replaced Supplementary Information with Expanded View (EV) Figures and Tables that are collapsible/expandable online. A maximum of 5 EV Figures can be typeset. EV Figures should be cited as 'Figure EV1, Figure EV2' etc... in the text and their respective legends should be included in the main text after the legends of regular figures.

10) The paper explained: EMBO Molecular Medicine articles are accompanied by a summary of the articles to emphasize the major findings in the paper and their medical implications for the non-specialist reader. Please provide a draft summary of your article highlighting

11) For more information: There is space at the end of each article to list relevant web links for further consultation by our readers. Could you identify some relevant ones and provide such information as well? Some examples are patient associations, relevant databases, OMIM/proteins/genes links, author's websites, etc...

12) Author contributions: CRediT has replaced the traditional author contributions section because it offers a systematic machine readable author contributions format that allows for more effective research assessment. Please remove the Authors Contributions from the manuscript and use the free text boxes beneath each contributing author's name in our system to add specific details on the author's contribution. More information is available in our guide to authors.

13) Disclosure statement and competing interests: We updated our journal's competing interests policy in January 2022 and request authors to consider both actual and perceived competing interests. Please review the policy <https://www.embopress.org/competing-interests> and update your competing interests if necessary.

14) Every published paper now includes a 'Synopsis' to further enhance discoverability. Synopses are displayed on the journal webpage and are freely accessible to all readers. They include a short stand first (maximum of 300 characters, including space) as well as 2-5 one-sentences bullet points that summarizes the paper. Please write the bullet points to summarize the key NEW findings. They should be designed to be complementary to the abstract - i.e. not repeat the same text. We encourage inclusion of key acronyms and quantitative information (maximum of 30 words / bullet point). Please use the passive voice. Please attach these in a separate file or send them by email, we will incorporate them accordingly.

15) As part of the EMBO Publications transparent editorial process initiative (see our Editorial at <http://embomolmed.embopress.org/content/2/9/329>), EMBO Molecular Medicine will publish online a Review Process File (RPF) to accompany accepted manuscripts.

In the event of acceptance, this file will be published in conjunction with your paper and will include the anonymous referee reports, your point-by-point response and all pertinent correspondence relating to the manuscript. Let us know whether you agree with the publication of the RPF and as here, if you want to remove or not any figures from it prior to publication. Please note that the Authors checklist will be published at the end of the RPF.

I look forward to receiving your revised manuscript.

Yours sincerely,

Lise Roth

Rev_Com_number: RC-2024-02445

New_manu_number: EMM-2024-19997-T

Corr_author: Andersson

Title: Jag1 Insufficiency Disrupts Neonatal T Cell Differentiation and Impairs Hepatocyte Maturation, Leading to Altered Liver Fibrosis

Response to reviewers

Manuscript number: RC-2024-02445, now EMM-2024-19997-T

Corresponding author(s): Jan Masek, Emma R. Andersson

1. General Statements

We thank the reviewers for their careful reading of our manuscript, for pointing out its strengths (including relevance to, and analysis of, human patients), and for providing constructive feedback to improve the clarity of *the manuscript*.

To accommodate the EMBO Molecular Medicine publishing format, we transformed Supplementary Figures 1,2,4,5,6 into Extended View Figures 1-5, and converted the former Supplementary Figure 3 into an Appendix File. The Supplementary tables are now Source Data Tables, and are available together with the rest of the raw data online, with a link provided in the Data Availability section of the manuscript. Thanks to the absence of a page limit, we also moved all the Materials and Methods from the Supplementary Materials and Methods file to the main manuscript.

Below is our point-by-point response and description of the associated revisions, please note the page and line numbers correspond to the pdf file “EMM-2024-19997-T_Manuscript.pdf” in this file, only insertions and new text are labeled in red:

Reviewer #1 (Evidence, reproducibility and clarity (Required)):

Reviewer 1 comment: This is an interesting study that examines defects in the Jag1^{ndr/ndr} mouse model of Alagille syndrome. The novel aspects of this manuscript are the comparisons, at many levels, between the mouse model and ALG patient samples, including an examination of immune profiles. The conclusions that the Jag1^{ndr/ndr} mouse model is an accurate representation of the human ALG syndrome appear valid. However the reported differences in immune profiles, particularly in the Jag1^{ndr/ndr} mouse model are difficult to understand. The data presented indicate a reduction in CD4⁺ cells in the Jag1^{ndr/ndr} mouse at day P3 in both liver and spleen. Additionally, the authors report differences between the the Jag1^{ndr/ndr} mouse and controls at day P30 in the relative percentages of DN, DP and SP CD4 and CD8 cells in the thymus. When examining the peripheral lymphoid system, CD4⁺ numbers are the same in both the Jag1^{ndr/ndr} animals and controls however CD8⁺ numbers are reduced and FoxP3/CD4⁺ cells are increased in both the spleen and the thymus. FoxP3/CD4⁺ T cells are usually assumed to be regulatory T cells that dampen the inflammatory responses of T cells. Therefore, the increase in this population in an animal model of what is assumed to be an inflammatory disease is confusing and confounding. The authors do not present a clear analysis of how they feel an increase of Tregs would lead to this disease. One possibility is that this population is not

functioning as conventional Tregs and rather are promoting inflammation but this conclusion would require a functional analysis of this population of cells, at the very least in an in vitro analysis of T cell suppression. From an immunologist's point of view, their data are antithetical to what one would expect to find in an inflammatory disease. Perhaps this reviewer is missing an important point but if I am missing it, then other who read this manuscript also may be confused.

Author Response: We thank the reviewer for carefully assessing our work, and for noting which aspects of the immune analyses should be more thoroughly explained. We apologize for any confusion, which a clearer introduction will help to avoid.

Alagille syndrome is not thought of as an inflammatory disorder, it is a congenital disorder affecting bile duct development (Kohut et al 2021, Semin Liver Dis). During normal bile duct development, JAG1+ portal fibroblasts signal to NOTCH2+ hepatoblasts to instruct bile duct development. In the context of low JAG1 signaling, hepatoblasts either fail to adopt a cholangiocyte fate, or fail to undergo bile duct morphogenesis, resulting in bile duct paucity and cholestasis. This cholestasis should activate inflammatory processes leading to fibrosis, which is the subject of this study.

We agree with the reviewer that Tregs would be expected to suppress inflammation, and our data are consistent with Treg suppression of inflammation. We show, for the first time, that Tregs are enriched in Jag1^{Ndr/Ndr} mice (Fig 4) and present evidence that Jag1^{Ndr/Ndr} immune cells suppress inflammation (Fig 5) and fibrosis (Fig 6), which could explain the atypical fibrosis seen in patients with ALGS.

To clarify that ALGS is a genetic liver disease affecting bile duct formation, we:

- 1. Modified and extended the following text in the Introduction (Page 2, lines 14-18): "ALGS is mainly caused by mutations in the Notch ligand JAGGED1 (JAG1, 94%) (Mašek & Andersson, 2017; Oda et al, 1997), affecting bile duct development and morphogenesis, resulting in bile duct paucity and cholestasis. Immune dysregulation has also been described (Tilib Shamoun et al, 2015), but how this might interact with liver disease in ALGS to affect fibrosis is not known."*
- 2. Introduce the disease, the animal model, and the scientific question in a schematic in new Fig 1A.*

Reviewer 1 comment: Again, in Figure 5, were FoxP3/CD4+ cells enumerated?

Author Response: Fig 5 showed that the inflammatory score, and activation of CD4 and CD8 cells, were lower in the intestine of DSS-treated mice transplanted with Jag1^{Ndr/Ndr} lymphocytes than in those transplanted with Jag1^{+/+} lymphocytes. However, in Figure 5 we had not quantified the number of FoxP3/CD4+ cells (Tregs). We agree that it would be interesting to know whether the dampened intestinal inflammation (in response to a classical inflammatory disease model (DSS treatment)) is also mediated by excess Tregs.

We therefore quantified Foxp3+ cells in colon sections from experimental animals used for the acquisition of data in Fig 5. The analysis, now presented in Extended View (EV) figure **EV 4F-H**, revealed no difference in the proportion of Treg/T cells (LCK+FOXP3+/LCK+) between the Jag1^{+/+}→Rag1^{-/-} and Jag1^{Ndr/Ndr}→Rag1^{-/-} animals. There was a small reduction, albeit not significant ($p=0.05223$), in the total number of LCK+ T cells in Jag1^{Ndr/Ndr}→Rag1^{-/-} animals which might explain the lower inflammatory response following the DSS treatment.

Based on this new data we added the following sentence to the manuscript (Page 5, lines 175,176): "While there was a tendency towards fewer LCK+ T cells in the colon, the ratio of FOXP3+ Tregs to LCK+ T cells was similar between the two experimental groups (Fig.EV4F-H)."

Reviewer 1 comment: Minor points that should be addressed include:

- The source cells used in the transfer experiments reported in Figure 5 is unclear. Are they using total spleen cells with T, B and myeloid cells or are they using purified T cells. And if it is the latter, have they assessed the ratio of CD4+ versus FoxP3/CD4+ cells in the transferred cells?

Author Response: Total spleen cells including all lymphocytes were transplanted, as described in Materials and Methods. The constituent T-cell populations are characterized and shown in Fig 4F. To clarify this, we:

1. added the text "Adoptive transfer of lymphocytes" to the schematic in Fig 5A, Fig EV4A, and Fig 6A, and
2. modified the opening paragraph related to results presented in Fig.5 and Fig EV4 in the following way (page 5, line 158): "To investigate Jag1^{Ndr/Ndr} T cell **function**, we performed **adoptive transfer of splenic lymphocytes** into Rag1^{-/-} mice, which lack mature B- and T cell populations, but provide a host environment with normal Jag1 (Mombaerts et al, 1992)."

To acknowledge that B-cells and innate lymphoid cells might contribute to the observed results, we include a following sentence in the Discussion:

(page 7, lines 296-298) "Finally, our experimental setup does not exclude an additional contribution by other lymphocytes (B-cells or innate lymphoid cells) to the altered BDL-

induced fibrosis, and selective testing of the individual subpopulations would be an intriguing follow up to this study.”

Reviewer 1 comment: In the DSS experiments in Figure 5, there does not appear to be a no DSS control. What does the architecture look like without DSS?

Author Response: The intestinal architecture and phenotype of mice transplanted with $Jag1^{+/+}$ or $Jag1^{Ndr/Ndr}$ lymphocytes, not treated with DSS, are presented in Figure EV4. In the absence of DSS, $Jag1^{+/+}$ - or $Jag1^{Ndr/Ndr}$ -transplanted mice exhibit no overt differences in survival or weight. The intestinal inflammatory score was not different in the two conditions and was 2.29 ± 0.44 and 2.03 ± 0.92 for $Jag1^{+/+}$ - or $Jag1^{Ndr/Ndr}$ -transplanted mice, respectively.

To compare the results with and without DSS, we added the following text to the results section, when describing the DSS results (Page 5, lines 172-175):

“As expected, histological scoring of intestinal inflammation revealed elevated inflammation in $Jag1^{+/+} \rightarrow Rag1^{-/-}$ mice treated with DSS (Fig. 5C,D) compared to $Jag1^{+/+} \rightarrow Rag1^{-/-}$ mice not treated with DSS (Fig. EV4D,E). However, there was significantly less inflammation in $Jag1^{Ndr/Ndr} \rightarrow Rag1^{-/-}$ mice than in $Jag1^{+/+} \rightarrow Rag1^{-/-}$ mice (Fig. 5C,D).”

Reviewer 1 comment: The authors noted that splenomegaly was observed in the

Jag1^{ndr/ndr} mouse model. Again this is antithetical to what one would expect when one sees an increase in FoxP3/CD4⁺ T regs.

Author Response: We thank the reviewer for pointing at a possible discrepancy, related to Fig1 in which we report the presence of splenomegaly. Although there can be multiple causes of splenomegaly, it is one of the hallmarks of portal hypertension (as also corroborated by Reviewer 2), tightly connected with liver fibrosis, present in patients with ALGS and we report it as such in the manuscript. To clarify this, we added the following text sections:

in the beginning of the results section:

1. (page 2, line 38,39) *“Liver fibrosis compresses blood vessels and reduces their blood flow, leading to portal hypertension, a serious consequence of liver disease which can manifest as splenomegaly.”*
2. and Discussion (page 8, starting at line 321): *“Splenomegaly has been described as a consequence of portal hypertension in ALGS (Kamath et al, 2020), but could also be attributed to immune-related pathology (McKenzie et al, 2018). Jag1^{Ndr/Ndr} mice exhibit splenomegaly as early as P10, which is exacerbated at P30 (Fig. 1E,F). Patients with other liver diseases display portal hypertension and cirrhosis, with both splenomegaly and hypersplenism associated with a high CD4⁺/CD8⁺ ratio, but a low Treg⁺/CD4⁺ ratio (Nomura et al, 2014). However, Jag1^{Ndr/Ndr} mice present with splenomegaly but not hypersplenism. An overactive spleen (hypersplenism) would remove red blood cells which are instead enriched in Jag1^{Ndr/Ndr} mice, and Tregs were enriched in Jag1^{Ndr/Ndr} mice, not depleted as seen in cirrhosis/hypersplenism. These data are thus consistent with portal hypertension-induced splenomegaly rather than hypersplenism.”*

Reviewer #1 (Significance (Required)):

Reviewer 1 comment: The strengths of this paper are the careful comparisons between the mouse model and the human ALG syndrome. These comparisons are valuable and worth publication.

Author Response: We thank the reviewer for these comments.

Reviewer 1 comment: Weaknesses are stated above. Needs a clearer explanation for their immune analysis.

Author Response: We thank the reviewers for highlighting points requiring clarification and hope the proposed text changes and additional data presented in response to the comments of all three reviewers lead to a significant clarification of the immunological aspect of our study.

Reviewer #2 (Evidence, reproducibility and clarity (Required)):

Reviewer 2 comment:

Summary:

Masek and colleagues use multi-pronged studies on the Jag1[Ndr/Ndr] mouse model of Alagille syndrome (ALGS) combined with transcriptomic analysis on livers from patients with ALGS to elucidate the potential mechanisms regulating liver fibrosis in this disease. The authors first show that Jag1[Ndr/Ndr] animals develop pericellular and perisinusoidal fibrosis and exhibit evidence for portal hypertension, similar to patients with ALGS. Single-cell RNA-sequencing indicated more hepatoblasts and less hepatocytes, relatively speaking, in Jag1[Ndr/Ndr] P3 livers, which suggested hampering of hepatoblast differentiation to hepatocytes. Deconvolution of previously generated bulk RNA-seq data from Jag1[Ndr/Ndr] P10 livers and GESA on RNAseq data from livers of these mice and patients with ALGS confirmed the P3 scRNA-seq observations and indicated mild pro-inflammatory activation of immature hepatocytes in ALGS livers. GESA also suggested an inability of Jag1[Ndr/Ndr] livers to attract T cells upon cholestatic injury. Indeed, 25-color flow cytometry on liver and spleen from mutant and control mice indicated a defect in T cell response to cholestasis in this model. The authors then examined the effects of the Ndr mutation on T-cell development and function. They found that the Ndr/Ndr thymi were significantly smaller than control thymi. Moreover, Ndr/Ndr thymi showed an increase in CD4⁺ T-cells and Tregs at the expense of double-positive T-cells. The authors then performed lymphocyte transplantation studies and concluded that Ndr/Ndr T-cells fail to mount an adequate response to inflammation in a DSS model of ulcerative colitis. The authors tested the contribution of Ndr/Ndr immune cells to liver fibrosis in a model of experimentally induced cholestasis (bile duct ligation; BDL). Ndr/Ndr T-cells did not show any defects in migrating into the liver upon BDL. However, the periportal fibrosis observed in BDL model was reduced in animals receiving Ndr/Ndr immune cells compared to those receiving Jag1^{+/+} immune cells. This was accompanied by significantly less aSMA staining in these livers. Finally, reanalysis of bulk RNAseq data from liver samples from ALGS and other liver diseases suggested that the presence of FOXP3⁺ T-reg cells in the liver is associated with higher liver fibrosis in non-ALGS liver diseases but lower liver fibrosis in ALGS livers. The authors have used an impressive combination of single-cell RNA-sequencing, reanalysis of previous bulk RNA-sequencing data from their group and others, 25-color FACS analysis, and adoptive immune transfer experiments in this manuscript, and systematically provide quantification and statistical analysis for their data. Overall,

this is an interesting and important study. Prior studies are referenced appropriately. The text and figures are clear and accurate. I don't think any additional experiments are essential. However, the issues listed under Major comments should be discussed and clarified in the manuscript, especially the first item.

Author Response: We sincerely thank the reviewer for the comprehensive and insightful assessment of our manuscript. We are particularly gratified to note your acknowledgment of the thoroughness of our experimental approach and the clarity of our presentation. We are pleased that no further experiments would be required, and will address the points raised under Major comments which enhance our study's quality and accessibility.

Reviewer 2 comment:

Major comments:

- Only a small fraction of the cells in scRNA-seq experiments have been assigned to hepatocytes/hepatoblast clusters, with the majority of these cells allocated to Hepato-Ery cluster. This suggests that many hepatocytes and potentially hepatoblasts have been lost during sample preparation. The authors should discuss this issue and its potential implications on the interpretation of the cell ratios and gene expression conclusions of scRNA-seq data.

Author Response: We agree with the reviewer regarding this aspect of our study. We mentioned this limitation in the supplementary methods section: "Liver parenchymal cells constituted ~6.5% of cells at E16.5, and ~7.5% of cells at P3 and included mesenchymal cells, endothelial cells, hepatoblasts and hepatocytes (Fig. EV1D), this parenchymal proportion is lower than in vivo, but consistent with ex vivo liver digest (Guilliams et al, 2022)." We recognize it may be too inaccessible there, and we thus added the following text to the Discussion section of the manuscript: (Pages 7, lines 259-26) "A limitation of this study is the underrepresentation of the hepatoblast/cyte parenchymal cells in the scRNA-seq dataset (Fig. 2A-D), which constituted ~6.5% of analyzed cells at E16.5, and ~7.5% of cells at P3 (Fig. EV1D). This parenchymal proportion is lower than in vivo, but is consistent with scRNA seq datasets obtained with ex vivo liver digest (Guilliams et al, 2022). One risk is that cell stress as a result of dissociation could result in further loss of injured Jag1^{Ndr/Ndr} hepatocytes, impacting the interpretation of cell type abundance. Nuclear scRNAseq can overcome cell type-dependent dissociation sensitivity bias (Guilliams et al, 2022), and could provide further insights into Jag1^{Ndr/Ndr} livers at the single cell level. Nonetheless, both

bulk RNA seq deconvolution and histological analyses both confirmed that patients and Jag1^{Ndr/Ndr} mice exhibit hepatoblast enrichment and less differentiated hepatocytes (Fig 2E-N).”

Reviewer 2 comment: The Jag1[Ndr/Ndr] strain is an excellent model for various aspects of ALGS phenotypes. However, when it comes to linking the effects of this mutation to the function of a specific cell type, it is worth considering that Jag1[Ndr/Ndr] might not recapitulate the effects of loss of one copy of JAG1 observed in most patients with ALGS. This is especially important given the sensitivity of various cellular and organ-level processes to the degree of Notch pathway activation. In the context of the present manuscript, it is possible that what the authors have observed in Jag1[Ndr/Ndr] lymphocytes does not mirror how a JAG1-heterozygous human lymphocyte behaves. This is not a major concern, but it is worth considering.

Author Response: *We agree and thus added the following discussion paragraph (page 6-7, lines 243-250) “In patients with ALGS, who have a single mutation in either JAG1 or NOTCH2, the remnant healthy allele(s) could be expected to mediate signaling. However, some JAG1 mutations exhibit dominant negative effects (Ponio et al, 2007; Xiao et al, 2013; Guan et al, 2023), which could entail further repression of JAG1/NOTCH2 signaling. In this context, it is important to note that the Jag1^{Ndr/Ndr} mice are homozygous for the missense mutation, but retain some JAG1 activity (Andersson et al, 2018; Hansson et al, 2010), and it is not clear to which degree this mimics JAG1 heterozygosity in humans. It would be of interest to test whether Jag1 potency affects hepatoblast differentiation or injury-induced reversion of hepatocytes in patients as a function of their specific JAG1 mutation.”*

Reviewer 2 comment: •The basis for the opposite type of correlation between COL1A1 expression and POXP3 level in ALGS versus non-ALGS liver disease is not clear.

Author Response: *We thank the reviewer for pointing out the unclear interpretation of the patient data. In patients with ALGS, the extent of fibrosis is likely to be highly multifactorial, involving (as we show) hepatocyte immaturity, dampened inflammation, and immune system dysregulation (possibly involving more than T-cells). Since human patients ARE so heterogeneous, teasing apart the relative contribution of each is currently outside the scope of our study, but will be an important area of future research. Nonetheless we thought it was important and interesting to show these patterns in Extended View Fig 5, now extended with further data, and analyses, and described and hopefully clarified in the following manner:*

Results section: (page 6, lines 209-217) “Liver damage in non-ALGS liver disease (using liver injury marker LGALS3BP) (Yang et al, 2021), was positively correlated with recruitment of lymphocytes (including CD8A+, and FOXP3+ populations of T cells), as well as the extent of fibrosis (COL1A1 abundance) (Fig. EV5G). However, in ALGS, the extent of liver damage, lymphocyte recruitment and fibrosis were unlinked (Fig. EV5G). These data are in line with the observation that liver stiffness (a proxy for fibrosis) in ALGS is independent of biomarkers of liver disease (Leung et al, 2023). While Treg infiltration in ALGS was independent of liver damage, it exhibited a tendency towards a negative correlation with fibrosis (Fig. EV5G, ST10), corroborating that elevated levels of Tregs may limit fibrosis in ALGS. Altogether, these data suggest that the liver and lymphocytes may be differentially affected in different patients with ALGS, a disorder that is well known for its heterogenous presentation.”

Minor comments:

- Page 2, last paragraph of Introduction, Page 12 last sentence, and Supplementary Methods: Please use "adoptive immune transfer" instead of "adaptive immune transfer".
- Pages 3 and 4: Reference is made to Figures 3E-O, which appears to be Figure 2E-O.
- Figure 3 legend: "Analysis in (E) is one-way ANOVA with Dunnett's multiple comparison test". Panel E compares two means, so ANOVA is not the appropriate statistical analysis for these data. Is this sentence related to panel D?
- Page 9: Please correct misspelling: "response to intestinal insult (Fig. 5). W therefore".
- The Science Translation Medicine references lack page number.

Author Response: We thank the reviewer deeply for taking the time to meticulously note and convey these errors, helping us to correct these. The suggested corrections have all been implemented. Science Transl Med is an online journal and does not have page numbers – we have added an issue number to facilitate retrieval of these references.

Additionally, we noticed that the image of a consecutive liver section with CYP1A2 staining from Jag1^{Ndr/Ndr} liver in Fig 2 L was accidentally flipped along the horizontal axis, which we have now corrected. Further, we changed the scRNAseq cell cluster naming from Hepatoblasts/cytes, Hepato_Ery, and Kupffer cells, Kuffer cells_Ery to Hepatoblasts/cytes I, and II, and Kupffer cells I and II, respectively, to match the Neutrophil progenitors I and II naming convention. Names were subsequently also changed in Fig S1 and methods.

****Referees cross-commenting****

To my knowledge, ALGS is not considered to be an inflammatory disorder. Furthermore, the splenomegaly observed in the mouse model could be due to portal hypertension rather than a primary immune disturbance. Having said that, I agree with the other reviewers that the manuscript will benefit from further discussion and clarification on the immune-related observations.

Author Response: *We thank Reviewer 2 for indicating to Reviewer 1 that ALGS is not considered an inflammatory disorder, which we fully agree with. It was not our intention to convey this idea. To avoid confusion, we now:*

1. *Added a schematic in Fig 1A.*
2. *Modified and extended the following text in the Introduction: (Page 2, lines 14-18): “ALGS is mainly caused by mutations in the Notch ligand JAGGED1 (JAG1, 94%) (Mašek & Andersson, 2017; Oda et al, 1997), affecting bile duct development and morphogenesis, resulting in bile duct paucity and cholestasis. Immune dysregulation has also been described (Tilib Shamoun et al, 2015), but how this might interact with liver disease in ALGS to affect fibrosis is not known.”*

Furthermore, we have addressed or will address all comments from reviewer 1 to clarify the immune-related observations.

Reviewer #2 (Significance (Required)):

Despite severe cholestasis, ALGS patients do not show as much fibrosis as other cholestatic diseases, including biliary atresia (BA). A previous study had suggested that this phenomenon could be due to the difference in the nature of reactive hepatobiliary cells in ALGS compared to BA (Fabris et al, 2007). Moreover, a number of studies have suggested a role for Notch pathway activation in several cell types in the liver in the development of liver fibrosis (for example, Sawitza et al, Hepatology, 2009; Chen et al, Plos One, 2012; Duan et al, Hepatology, 2018; Yu et al, Science Translational Medicine, 2021). However, although a role for Notch signaling in T-cells is well established, it was not known whether impaired T-cell development/function contributes to reduced fibrosis in ALGS liver disease. Accordingly, the current manuscript provides novel insight into the mechanism of fibrosis in this disease. Moreover, the observation that Jag1-mutant T-cells do not confer as much protection as control T-cells to immunodeficient mice subjected to DSS-induced ulcerative colitis provides strong evidence for impaired T-cell immunity in this ALGS model and might help explain other aspects of ALGS phenotypes.

The manuscript will be of interest to broad audience (Notch signaling, cholestatic liver disease, mechanisms of liver fibrosis, T-cell development).

I have expertise in Notch signaling and in using animal models of human developmental disorders.

Author Response: We thank the reviewer for the balanced assessment of our manuscript in light of the current knowledge, and for highlighting its importance in the context of not only Notch and ALGS, but also other cholestatic and fibrotic liver diseases.

Reviewer #3 (Evidence, reproducibility and clarity (Required)):

The article entitled "Jag1 Insufficiency Disrupts Neonatal T Cell Differentiation and Impairs Hepatocyte Maturation, Leading to Altered Liver Fibrosis" by Mašek et al described the role of Notch ligand JAGGED1 (JAG1) in the T-cell differentiation contributing to liver fibrosis and immune system development in ALGS. This article is well written and has important preliminary findings that could establish Jag1 and its downstream signaling pathways as potential therapeutic targets to attenuate liver fibrosis.

Author Response: We thank the reviewer for these comments and pointing out the therapeutical implications of our findings.

1. Minor comments: In page 4, they mentioned that "the hepatoblast marker alpha fetoprotein (AFP) was 3.1-fold enriched (Fig. 3J,K), while the mature hepatocyte marker CYP1A2 protein was 1.7-fold less expressed (Fig. 3L-M)", the figure numbers should be changed to 2J, K, L-M etc.

Author Response: We thank the reviewer for identifying these errors. The suggested corrections have been implemented.

Reviewer 3 comment 2: In liver fibrosis the Th17 cells play crucial roles. Please show the level of IL17A mRNA level in the liver in the Jag1Ndr/Ndr mice compared to the Jag1+/+ mice.

Author Response: We thank the reviewer for the insightful comments. We indeed investigated the Th17 vs Treg immune response, however we detect neither Th17-expressed Il17, Il17a, Il17f, nor Il21 and Il22 mRNA in the bulk RNA data, suggesting their expression is either masked or they are not present in significant numbers within the liver tissue at P10, preventing us from drawing any conclusions about this cell population.

Reviewer 3 comment 3: Also, please show the expression level of pro-inflammatory molecules, for example, TNF α , IL1 β , MCP1 etc and the level of MMPs (especially MMP2, MMP8, MMP9) in the livers of the mice models used.

Author Response: The expression of *Il10*, *Il1b*, *Mcp1*(*Ccl2*), was presented in the manuscript (Fig. 2O), and we attach here a full list together with the expression levels of *Mmp2/8/9*, *Tnfa*, *Ifng*, *Il17* receptor family and *Tgfb1-3*. Out of these, *Mmp8* (0.9 Log2fold change = 1.9-fold), *Ccl2* (2.2 Log2fold change = 4.7-fold), and *Tl17rb* (1.1 Log2fold change = 2.1-fold) were significantly upregulated, but do not indicate any specific leukocyte population's response. This is in line with data in Fig EV2E, demonstrating a dominance of myeloid over adaptive immune response in the GSEA of the immune KEGGs.

Since lymphocytes are underrepresented in the bulk transcriptomics, and individual genes might report activity of many different cell types, we chose to focus on the list of genes shown to be markers of activated hepatocytes, to avoid over interpretation of the RNA sequencing data. Instead, the immune analyses were based on flow cytometry data, which we expect should accurately report cell type abundance across organ systems.

Reviewer 3 comment 4. Authors have shown significant alterations in the Treg population in their Jag1Ndr/Ndr mice of ALGS. Please also show the expression of IL10 and TGFβ in the liver and whether they are correlated with the level of Treg populations.

Author response: IL10 and Tgfb mRNA levels in liver are shown in the heatmap above, and were not significantly different between genotypes at P10. They were also not correlated with Foxp3 levels, as shown in the correlation matrices below (Pearson's R values in top row, significance values in bottom row).

Reviewer 3 comment 5. It would be interesting to know whether the IFNγ mRNA expression in the livers were altered in the Jag1Ndr/Ndr mice with altered populations of CD8 T cells.

Author Response: There was no significant difference in IFNγ mRNA expression levels between Jag1^{+/+} and Jag1^{Ndr/Ndr} livers at P10 (please see the heatmap in response to comment no.3, above).

Reviewer #3 (Significance (Required)): Strength: This article is well written and has important preliminary findings that could establish Jag1 and its downstream signaling pathways as potential therapeutic targets to attenuate liver fibrosis.

Author Response: Thank you for these comments and pointing out the wider implications of our findings.

Reviewer 3 Limitations: This study lacked the detailed molecular pathways which could explain how the Jag1 altered the T-cell recruitment, development and hepatocyte maturation in the development of liver fibrosis in the ALGS model.

Author Response: We agree that this study does not focus on molecular pathways. The aim of this study was to identify which cell populations contribute to atypical neonatal fibrosis in ALGS. Because we expected this process to be multifactorial, Jag1^{Ndr/Ndr} mice, carrying a systemic mutation, present both advantages (Jag1 abrogation in all cells  ALGS-like organ interactions) and limitations (inability to identify contributions of individual cell types). However, by identifying maturing hepatocytes and Tregs as dysregulated, and demonstrating that Jag1^{Ndr/Ndr} lymphocytes behave abnormally and suppress inflammation and fibrosis in Rag1^{-/-} mice (with normal Jag1 expression), we establish a biological framework that can now be further investigated with conditional genetic tools and in vitro systems, to elucidate specific molecular pathways, that were beyond the scope of the current study.

Description of analyses that authors prefer not to carry out

Not applicable

13th Aug 2024

Dear Dr. Andersson,

Thank you for submitting your revised study. We have now received the feedback from referees #1 and #3 who evaluated your revised manuscript (and had reviewed your initial submission at Review Commons), and as you will see below, they are satisfied with the revisions. I am therefore pleased to inform you that I will be able to accept your manuscript once the following minor comments are addressed:

1/ Manuscript text:

- Please accept previous changes and only keep in track changes mode any new modification.
- Please note that email bounced for Afshan Iqbal (afshan.iqbal@ki.seq).
- We can accommodate a maximum of 5 keywords, please adjust accordingly.
- Methods:
 - o Thank you for including a Reagents and Tools Table. Please remove it from the manuscript and upload it as a separate document using the latest format. The callouts in the manuscript need to be updated accordingly.
 - o Please indicate the gender of the mice used in the experiments.
 - BioRender information needs to be removed from the Acknowledgments section and it needs to be noted at the end of the Methods (only) as follows:
Graphics:
(some of the... OR Figure #... OR synopsis) Graphics were created with BioRender.com.
- The Data Availability section should be placed after the Methods and before the Acknowledgements.
- Acknowledgements should match the information entered in the submission system. (Currently Swedish Research Council and grant agreement no. 2018-05973 are not entered in the submission system).

2/ Figures and Appendix:

- Please make sure all figures and figure panels are referenced in the text, and in chronological order (currently, figures Fig 5L and Fig 5N are called out before Fig. 2N, but Fig 5 doesn't have L and N panels).
- The correct nomenclature for the EV figure legends is: Figure EV1, Figure EV2, etc. instead of Extended View 1, Extended View 2, etc; the heading should be Expanded View Figures instead of Extended View.
- Appendix file: the nomenclature and callout need to be Appendix Figure S1. The page number for the figure is missing in the table of content on the title page.
- Please address the queries from our data editors in the figure legends (if not yet done):
 1. Please note that the exact p values are not provided in the legends of figures 2e-f, h, o; 3d, f; EV 5g.
 2. Please indicate the statistical test used for data analysis in the legends of figures 2h, o; EV 2d-e; EV 5f-g.
 3. Please note that the p value is not represented in the figure 3f, however statistical test related information is provided in the legend of the corresponding figure. This needs to be rectified.
 4. Please note that the box plots need to be defined in terms of minima, maxima, centre, bounds of box and whiskers, and percentile in the legend of figure EV 2a.
 5. Please note that information related to n is missing in the legends of figures 2f; 6d, f, h, j; EV 1a, d; EV 2a.
 6. Please note that the error bars are not defined in the legends of figures EV 1d; EV 3c-d; EV 5c, e.
 7. Please note that the scale bar needs to be defined for figures 6g, i.
- Please clarify "not shown" in figure 2N.

3/ Checklist:

- please fill in "Experimental Animals/housing and husbandry conditions"
- please fill in "Experimental Study Design and Statistics/ randomization procedure"

4/ For more information:

We recently stopped using this section. Please remove the FMI section from your manuscript.

5/ I introduced minor changes in your synopsis. Please let me know if you agree with the changes or amend as you see fit:

"Despite severe cholestatic liver disease due to bile duct paucity, intrahepatic fibrosis in Alagille syndrome (ALGS) differs from other cholestatic liver diseases. The way cell populations are affected by ALGS and interact to influence disease progression was investigated in an ALGS mouse model.

- Intrahepatic ALGS-like pericellular fibrosis is recapitulated by Jag1Ndr/Ndr mice.
- Single-cell transcriptomics and flow cytometry identified dysregulation of maturing hepatocytes and T cells during fibrosis onset and propagation.
- Jag1Ndr/Ndr and ALGS hepatocytes express a hepatoblast-like signature, suggesting disrupted hepatocyte maturation and compromised activation.

Regulatory T cells are enriched in Jag1Ndr/Ndr mice and can limit periportal fibrosis, as demonstrated by Jag1Ndr/Ndr cell transplantations into immunodeficient mice followed by surgically induced cholestasis."

Please also resize your synopsis picture to 550 px wide x 300-600 px high, and make sure the text remains legible.

6/ As part of the EMBO Publications transparent editorial process initiative (see our Editorial at <http://embomolmed.embopress.org/content/2/9/329>), EMBO Molecular Medicine will publish online a Review Process File (RPF) to accompany accepted manuscripts.

This file will be published in conjunction with your paper and will include the anonymous referee reports, your point-by-point response and all pertinent correspondence relating to the manuscript. Let us know whether you agree with the publication of the RPF and as here, if you want to remove or not any figures from it prior to publication.

I look forward to receiving your revised manuscript.

Yours sincerely,

Lise Roth

***** Reviewer's comments *****

Referee #1 (Comments on Novelty/Model System for Author):

This manuscript presents data that more fully describe the role of Jag1 in a mouse model of Allagile Syndrome and provides valuable insights into the human disease.

Referee #1 (Remarks for Author):

The authors very clearly addressed my initial concerns with the manuscript. Many of my concerns related to my understanding of the disease and the authors added text as well as new figures which clarified the role of inflammation in their model. The addition of Figure 1A is very helpful as is the additional text added to the manuscript.

Referee #2 (Comments on Novelty/Model System for Author):

Please see my Review Commons review.

Referee #2 (Remarks for Author):

The authors have addressed the issues that I raised in my Review Commons review. I have no further comments or concerns about this manuscript.

Rev_Com_number: RC-2024-02445

New_manu_number: EMM-2024-19997-V2

Corr_author: Andersson

Title: Jag1 Insufficiency Alters Liver Fibrosis via T Cell and Hepatocyte Differentiation Defects

The authors addressed the minor editorial issues.

9th Sep 2024

Dear Dr. Andersson,

Thank you for submitting your revised files. I am pleased to inform you that your manuscript is accepted for publication and is now being sent to our publisher to be included in the next available issue of EMBO Molecular Medicine!

With kind regards,

Lise Roth
